# Differential KEAP1/NRF2 mediated signaling widens the therapeutic window of redox-targeting drugs in SCLC therapy

Jana Samarin [1,20], Hana Nůsková [1,20], Piotr Fabrowski [1], Mona Malz[1], Eberhard Amtmann[1], Minerva J. Taeubert[1], Daniel Pastor-Flores[2,3], Daniel Kazdal [4,5], Roman Kurilov[6], Nicole de Vries [1], Hannelore Pink[1], Franziska Deis [1], Johanna Hummel-Eisenbeiss[1], Lisa Renz[1], Kamini Kaushal[7], Michael Morgen [1], Tobias P. Dick [2,3], Gerhard Hamilton[8], Martina Muckenthaler [9], Moritz Mall [10,11,12], Bryce Lim [10,11,12], Taishi Kanamaru [10,11,12], Glynis Klinke[13], Martin L. Sos[14,15,16,17], Julia Frede[14,15,18], Aubry K. Miller[1,19], Hamed Alborzinia[7,21] & Nikolas Gunkel [1,19,21] ✉

Small cell lung cancer (SCLC) patients frequently experience a remarkable response to first-line therapy. Follow up maintenance treatments aim to control residual tumor cells, but generally fail due to cross-resistance, inefficient targeting of tumor vulnerabilities, or dose-limiting toxicity, resulting in relapse and disease progression. Here we show that SCLC cells, similar to their cells of origin, pulmonary neuroendocrine cells, exhibit low activity in pathways protecting against reactive oxygen species (ROS). When exposed to a thioredoxin reductase 1 (TXNRD1) inhibitor, these cells quickly exhaust their ROS-scavenging capacity, regardless of their molecular subtype or resistance to first-line therapy. Importantly, unlike non-cancerous cells, SCLC cells cannot adapt to drug-induced ROS stress due to the suppression of ROS defense mechanisms by multiple layers of gene regulation. By exploiting this difference in oxidative stress management, we safely increase the therapeutic dose of TXNRD1 inhibitors in vivo by pharmacological activation of the NRF2 stress response pathway. This results in improved tumor control without added toxicity to healthy tissues. These findings underscore the therapeutic potential of TXNRD1 inhibitors for maintenance therapy in SCLC.

Small cell lung cancer (SCLC) accounts for approximately 15% of all lung cancer cases and is considered one of the most lethal cancer types, with a 5-year survival rate of less than 7%[1]. The main challenges in SCLC include rapid tumor growth, early metastatic spread, and a tendency to relapse with profound therapy resistance, resulting in limited treatment options. Standard-of-care (SoC) treatments target the exceptionally high proliferative rate of SCLC cells and their dependency on the biosynthetic pathways required for cell replication. This is achieved through the use of DNA cross-linking agents like cisplatin or carboplatin, topoisomerase inhibitors such as etoposide or topotecan and γ-radiation, all of which target DNA synthesis, replication, and repair[2–4].

Given that 70–80% of SCLC patients respond well to chemotherapy, it seems reasonable to assume that maintenance therapy represents a promising strategy. However, the success of maintenance therapy depends on the vulnerability of the treated tumor, its inability to develop cross-resistance mechanisms during the cytotoxic assault by the drug and conversely, the presence of robust defense

mechanisms in healthy tissue to mitigate the drug's side effects. Due to this complex set of requirements, targeting cell cycle control via ATR, WEE1, and CHK1 inhibitors, as well as DNA repair pathways through PARP1 inhibitors such as Olaparib and Veliparib has failed to provide substantial benefit for patients[5]. More recently, a combination of SoC and immunotherapy has been explored in clinical trials, resulting in a transient delay of tumor relapse for a few months[6]. Another promising approach is the inhibition of LSD1, which has been implicated in SCLC lineage specification[7] and recognition by the immune system[8]. However, the heterogeneous drug responses in SCLC cells, arising from both intrinsic and acquired resistance to multiple LSD1 inhibitors, limit the potential of this strategy until biomarkers are identified to predict drug response[9]. Furthermore, recent studies have shown that chemotherapy-resistant SCLC cells exhibit elevated MYC expression and display distinct metabolic vulnerabilities. Thus, in murine models harboring MYC-driven tumors, targeted arginine depletion using pegylated arginine deiminase (ADI-PEG 20) significantly inhibited tumor progression and improved survival outcomes[10]. Inspired by the notion that epigenetic mechanisms, rather than druggable genetic drivers determine adaptation to their microenvironment and response to therapy[11], epigenetic drugs like Romidepsin or Panobinostat provided a compelling rationale for treatment but ultimately failed to demonstrate a significant benefit for patients[12,13]. In the light of these failures to translate preclinical concepts into improved disease control, SCLC remains regarded as a "graveyard of drug development"[14].

An underexploited concept in cancer therapy involves the use of redox-targeting drugs. The rationale behind this approach is that cancer cells are more vulnerable to therapy-induced reactive oxygen species (ROS) stress than non-cancerous cells, thereby creating an opportunity for selective drug targeting. We have recently identified a set of 15 genes, called "**A**nti-oxidant **C**apacity **B**iomarkers" (ACB) which accurately predict the sensitivity of cell lines across various cancer types to redox-targeting drugs such as ferroptosis inducers (GPX4 inhibitors) or thioredoxin reductase 1 (TXNRD1) inhibitors[15]. Importantly, only defined patient groups in each cancer entity express a favorable ACB profile (low-ACB expression), emphasizing the necessity of patient stratification for redox-targeting therapies.

In this work, we show that pathways protecting against ROS stress are underactive in SCLC cells. Upon treatment with a TXNRD1 inhibitor, these cells rapidly deplete their ROS-scavenging capacity, regardless of molecular subtype or cisplatin resistance status. Notably, unlike non-cancerous cells, SCLC cells are unable to compensate for TXNRD1 inhibition via the NRF2 stress response pathway. Our investigation into the biochemical and molecular basis of sustained drug sensitivity revealed a complex mechanism, potentially involving both epigenetic and transcriptional regulation. Leveraging this reduced ability to manage and adapt to oxidative stress, we conducted proof-of-concept studies in mouse models, demonstrating that TXNRD1 inhibition robustly maintained first-line therapy efficacy. Additionally, selective activation of the NRF2 stress response pathway using Bardoxolon-Methyl (CDDO-Me) allowed to increase the therapeutic dose of the TXNRD1 inhibitor, which improved tumor control without increasing toxicity in healthy tissues. These results highlight the therapeutic promise of TXNRD1 inhibitors for cancers like SCLC, characterized by diminished ROS-scavenging capacity and robust suppression of adaptive resistance mechanisms.

## Result

### SCLC cells are hypersensitive to TXNRD1 inhibition, independent of their resistance-status to cisplatin

We previously identified a set of 15 antioxidant capacity biomarkers (ACBs) involved in interconnected redox pathways. By calculating the ACB score as the average expression of these genes, we were able to accurately predict sensitivity to redox-targeting agents, such as inhibitors of TXNRD1 and GPX4[15]. Non-small cell lung cancer (NSCLC) cell lines show varying levels of ACB expression. Lines carrying loss-of-function mutations in KEAP1 or CUL3 exhibit increased ACB expression, whereas those with wild-type KEAP1 or non–loss-of-function KEAP1 mutations generally display low basal ACB levels. Likewise, in SCLC, ACB expression remains low across most cell lines, consistent with their wild-type KEAP1 status—suggesting a high susceptibility to redox-targeting therapies. (Figs. 1A, left panel, B, and S1A). Interestingly, the ACB status of SCLC is largely independent of the neuroendocrine subtype, with the exception of samples derived from PDX models (Fig. S1B). Moreover, we observed that the ACB levels are significantly lower in SCLC tumors as compared to normal tissues (Fig. 1C), suggesting that a selective tumor response to TXNRD1 inhibitors could be achieved in SCLC patients. Of note, pulmonary neuroendocrine cells (PNECs), which are the cells of origin of SCLC (reviewed in ref. 16), demonstrate similarly low-ACB scores, indicating that the ACB status of SCLC cells is inherited rather than selected during carcinogenesis.

To test whether this low-ACB status translates into high sensitivity to redox-targeting drugs, we assessed the activity of two TXNRD1 inhibitors, DKFZ-682 and DKFZ-608, in a panel of SCLC cell lines ($n = 13$) that capture ACB expression levels present in 82% of this cancer entity (Fig. 1A). The two molecules are equipotent enzymatic inhibitors of TXNRD1, show high correlation of cellular TXNRD1 inhibition and cytotoxicity and are, in contrast to auranofin, predominantly acting in the cytoplasm with limited mitochondrial ROS induction[15,17] (Fig. S1C–E). Our results show that unlike NSCLC, SCLC cells consistently exhibit sensitivity to TXNRD1 inhibition (Figs. 1A, right panel and S1F), in line with the notion that low ACB levels translate into high activity of redox-targeting drugs in cancer. Furthermore, it highlights the potential of ACBs as a highly predictive biomarker for patient stratification, offering guidance on the use of oxidative stress induction as a therapeutic strategy.

The homogenous activity profile of our TXNRD1 inhibitors is especially notable compared to (1S,3R)-RSL3 (RSL3) and the selective nitroisoxazole-GPX4 inhibitor ML210[18], all of which exhibit a more variable response pattern, independent of the cell's ASCL1/REST or NEUROD1/REST status (Fig. S2A, B). Overall, these findings suggest that TXNRD1 inhibition holds strong potential for SCLC therapy, given its consistent activity profile and independence from the neuroendocrine (NE) status of target cells.

In order to assess the efficacy of TXNRD1 inhibition in the context of platinum-based therapy, we assembled a panel of cells derived from both chemotherapy relapsed patients (PC, post chemo) and treatment-naïve patients (CN, chemo naïve) with varying resistance or sensitivity to cisplatin. PC-cells exhibit reduced expression levels of SLFN11 (Fig. S2C), a previously described biomarker for chemotherapy resistance[19]. As such, these cells are approximately 20-fold more resistant to cisplatin compared to CN-cells (Figs. 1D and S2D). Moreover, the cell line panel accurately reflects on the narrow therapeutic window of cisplatin following relapse, as we observe a ratio of only 1.7 of the average sensitivity of PC-SCLC compared to HaCaT and Beas-2B, which we used to represent non-cancerous cells. In contrast to cisplatin, the TXNRD1 inhibitors DKFZ-682, DKFZ-608 and auranofin demonstrated equal efficiency in PC and CN cells, showing that TXNRD1 inhibitors are not affected by mechanisms causing cisplatin resistance (Fig. 1D). A comparison to non-cancerous cells revealed a therapeutic window of up to 50-fold for DKFZ-608, 6-fold for DKFZ-682 and 3.8-fold for auranofin (Fig. S2D). The consistently high toxicity of TXNRD1 inhibitors across the entire SCLC cell line panel suggests an opportunity in overcoming resistance induced by first line therapy. Therefore, we examined the effect of DKFZ-682 and DKFZ-608 on cells that survived cisplatin treatment (CPR, cisplatin-resistant), either selected in cell culture or derived from relapsed xenograft tumors in mice. In both cases, the cytotoxic efficacy of TXNRD1 inhibition remained unaffected (H69$_{CPR}$, H526$_{CPR}$) or only marginally decreased

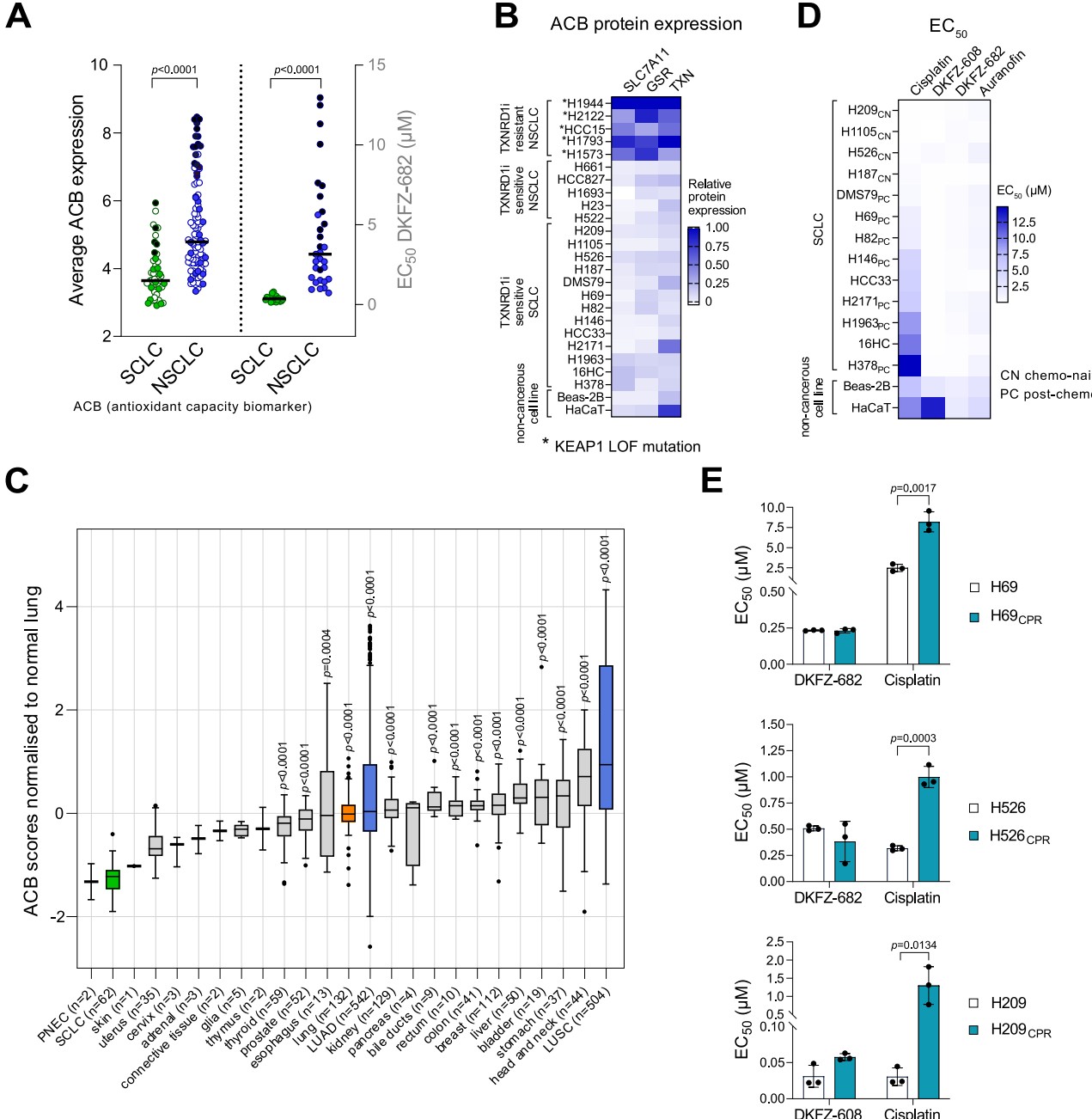

**Fig. 1 | SCLCs have a low expression of ACBs and are very sensitive to TXNRD1 inhibition, regardless of their resistance-status to cisplatin. A** Left panel: The expression of 15 ACB genes for each SCLC ($n = 50$) and NSCLC cell lines ($n = 98$). Right panel: $EC_{50}$ values of selected SCLC ($n = 13$) and NSCLC ($n = 32$) cell lines. Dots represent mean $EC_{50}$ values derived from two (H1105, HCC33, HCC15, H1573, H1792, H1437, H3122, and H1651) or at least three independent experiments for the remaining cell lines, each performed with biological triplicates. Statistical significance was assessed using a two-tailed unpaired $t$-test. The exact $p$ values are indicated in the figure. Cell lines with known loss-of-function mutations in KEAP1 or Cul3 are indicated as black dots. **B** The heatmap with the protein levels of SLC7A11, GSR, and TXN, analyzed in total cell extracts by immunoblotting. Cell lines with loss-of-function mutations (LOF) in KEAP1 or Cul3 are indicated with asterix. Protein expression in H1944 was set to 1 for each protein and for each experiment. Relative data represent mean of two (SLC7A11, GSR, TXN1 in SCLC; SLC7A11 in NSCLC) or three (GSR and TXN1 in NSCLC) independent experiments. **C** Expression values of ACB genes were normalized to normal lung tissue. Box plots display the median (center line), interquartile range (box), and whiskers extending to 1.5× the interquartile range; individual data points are overlaid. Statistical differences among groups were assessed using the Kruskal–Wallis test followed by two-sided Dunn's multiple comparisons test comparing each tumor type with SCLC. Exact $p$ values are indicated in the figure when significant (<0.05). Sample sizes for each group are indicated on the $x$-axis labels. **D, E** Data are presented as mean of $EC_{50}$ values of three independent experiments, each performed in biological replicate. **E** Statistical significance was assessed using a two-tailed unpaired $t$-test. Exact $p$ values are indicated in the figure when significant (<0.05). **D** CN: derived from chemo-naïve patients, PC: derived from patients after cisplatin plus etoposide therapy. **E** H69$_{CPR}$, H526$_{CPR}$, H209$_{CPR}$ are cisplatin-resistant sublines. Source data are provided as a Source data file.

(H209$_{CPR}$), while significant resistance to cisplatin was observed (Figs. 1E and S3A, B). In addition, we tested primary circulating tumor cells (CTC), which were collected from relapsed SCLC patients[20]. Consistent with cell lines, CTCs exhibited a heterogeneous response to cisplatin with an inverse correlation of SLFN11 expression and drug sensitivity ($r = -0.62$) (Fig. S2C) and a homogeneously high response to TXNRD1-inhibitor treatment, with DKFZ-608 being more efficient than auranofin (Fig. S3C).

A recently published single-cell data set[21] analyzing paired samples of CTCs from a naïve and relapsed patient revealed that ACB scores are only marginally increased in the cell population surviving cisplatin therapy (Fig. S3D). Additionally, ACB levels do not segregate according to epithelial-mesenchymal transition (EMT) scores (Fig. S3E), which have previously been shown to correlate with therapy resistance[22]. Our cell line data demonstrate that SCLC cells are highly sensitive to TXNRD1 inhibitors and that cisplatin-resistant SCLC cells maintain this sensitivity, suggesting that TXNRD1 inhibition is a promising strategy for maintenance therapy of SCLC.

Intriguingly, the mechanism of cell death within the SCLC cell panel appears heterogeneous, despite the homogeneous toxicity induced by TXNRD1 inhibition. In HCC33 and H82, in which TXNRD1 inhibition induced high levels of hydrogen peroxide ($H_2O_2$) and hydroxyl radical ($\cdot OH$), the iron/copper chelator deferoxamine (DFO) strongly reduces cell death, presumably by preventing the formation of toxic levels of $\cdot OH$. through Fenton reaction (Fig. S4A–C). Despite this dependency on iron, the involved cell death mechanism appears to be independent of lipid peroxidation and distinct from ferroptosis, as ferrostatin-1 cannot rescue cell death, although it prevents lipid peroxidation (Fig. S4D, E). In H1105, in contrast to the afore mentioned cell lines, DKFZ-682 induces only a moderate increase of $H_2O_2$ and $\cdot OH$. Here, toxicity cannot be rescued by DFO (Fig. S4A–C). Consistent with the observed drug induced oxidation of the marker proteins PRDX1 (Fig. 3F) and roGFP2-Orp1 (Fig. S1E), pretreatment of cells with N-acetyl-L-cystein (NAC) reduced the toxicity of DKFZ-682, supporting the notion that ROS induction is a primary driver of cell death (Fig. S4F). Having established that DKFZ-682 induces in H82 and HCC33 iron-dependent cell death, distinct from ferroptosis (Fig. S4A–E), we examined various cell death mechanisms commonly associated with redox-targeting drugs by using pathway-specific inhibitors or measuring key biomarkers. In H82, DKFZ-682-induced cell death was not mitigated by PARP or caspase inhibitors (olaparib or Z-VAD), suggesting these pathways are not involved (Fig. S4G). In contrast, in H1105 cells, where DKFZ-682 only moderately increased $H_2O_2$ and $\cdot OH$ (Figure. S4B, C), toxicity was partially reduced by olaparib, implicating a role for PARP-dependent mechanisms in this cell line (Fig. S4G). Given that stressed cells can switch between apoptosis and necroptosis when one pathway is blocked[23], we tested whether combined inhibition of apoptosis and necroptosis (using Z-VAD and Nec-1) would reduce DKFZ-682 toxicity. Since this combination did not protect cells (Fig. S4G), we speculated that TXNRD1 inhibitors like DKFZ-682 induce paraptosis[24], a non-apoptotic/necroptotic cell death which depends on the de novo translation of Activating Transcription Factor 4 (ATF4) to execute cell death induced by redox-targeting drugs[25,26]. In line with those reports we observed that DKFZ-682 induces the translation of ATF4, and that the pretreatment of cells with translation inhibitor cycloheximide (CHX) prevented drug induced ATF4 expression and reduced DKFZ-682 toxicity (Fig. S4H), suggesting that DKFZ-682 induces paraptosis. Ultimately, cells undergo necrotic plasma membrane rupture which is evident by the release of LDH to the medium (Fig. S4I).

## Targeting TXNRD1 maintains efficacy of first-line therapy in vivo

SCLC patients typically show high initial response rates to cisplatin-based chemotherapy, followed by relapse shortly after completing first-line therapy. Given the significant efficacy of TXNRD1 inhibition

observed in both naïve and cisplatin-resistant cancer cells, we hypothesized that our TXNRD1 inhibitors could be effective as maintenance therapy following first-line therapy with cisplatin/etoposide. To test this hypothesis, we used our preclinical xenograft model with the human SCLC cell line H209, which was derived from an untreated patient and asked whether DKFZ-608 could prevent tumor recurrence following complete remission after standard chemotherapy with cisplatin/etoposide. DKFZ-608 has a favorable pharmacokinetics profile in that its plasma concentration at the maximal tolerated dose (MTD) remains long enough at levels sufficient to eradicate high density H209 cells (Fig. S5A, B). The experimental setup and the therapeutic regimen closely mirror those used in most previously published SCLC xenograft studies, with the key difference that we extended monitoring of tumor-free survival (time to recurrence) for an additional 4 months beyond the end of treatment to detect delayed recurrence (Fig. 2A–C).

As expected, the tumors in the control group (vehicle only, black line, Fig. 2A) showed rapid growth, necessitating the sacrifice of mice within $37 \pm 9$ days. In contrast, the second group that received 3 weekly cycles of cisplatin (3 mg/kg, interperitoneally, i.p.) and etoposide (7.5 mg/kg, i.p.), responded with rapid tumor shrinkage (gray line, Fig. 2A). After 20 days, tumors were undetectable in 9 out of 10 animals, leading to the cessation of treatment. However, 18 days later, measurable tumors reappeared in all animals with the first animal reaching maximum tumor size of 1500 mm$^3$ within 60 days after treatment discontinuation. The mean survival time, calculated from the start of the experiment, was $100 \pm 11$ days (Fig. 2B). Cells extracted from recurrent tumors had developed substantial resistance against cisplatin (>40-fold) while their response to DKFZ-608 remained largely unaffected, confirming our in vitro data which suggest that DKFZ-608 is not subject to mechanisms involved in cisplatin resistance (Fig. S3B).

The third group received the MTD of DKFZ-608 (15 mg/kg daily, i.p., see Fig. S5C, D for details) immediately following chemotherapy as a maintenance treatment for 40 days (blue line, Fig. 2A, B). Notably, no tumor recurrence was observed until 96 days after the end of chemotherapy or 56 days after discontinuation of DKFZ-608. Thereafter, we observed a faint regrowth of the tumor in a single animal. The remaining nine mice were tumor-free until day 142, at which point the experiment was concluded.

The fourth group, receiving monotherapy of DKFZ-608 (15 mg/kg), also showed tumor shrinkage (green line, Fig. 2A). By the end of the treatment (day 43), 6 animals were tumor-free, while the remaining 4 showed approximately 70% tumor shrinkage. In these four animals, tumor growth resumed after discontinuation of therapy, while no recurrence was observed in the six tumor-free animals during the 142-day experimental period.

The preclinical data presented here clearly demonstrate that the TXNRD1 inhibitor DKFZ-608 is a promising drug candidate for the treatment of SCLC, effectively targeting both first-line sensitive and resistant SCLC cells. It is capable of completely suppressing tumor relapse for several months at a sub-toxic dose. To the best of our knowledge, this is the first instance of such a robust remission being achieved in an SCLC mouse model.

## Reduced ROS-scavenging capacity mediates vulnerability to redox-targeting drugs in SCLC

The activity of NRF2 is widely recognized as a key determinant of resistance to many cancer drugs (reviewed in ref. 27). The hypersensitivity of SCLC cells to TXNRD1 inhibition raises the question of whether NRF2-mediated ROS defense is compromised in these cells. SCLC and NSCLC cells have similar *NRF2* transcript expression levels, but significantly different *NQO1* mRNA levels (Fig. S6A, B). Considering *NQO1* mRNA levels as a proxy for NRF2 activity[28] indicates that SCLC cells exhibit on average lower NRF2 activity than NSCLC cells. This finding aligns with the low NRF2 protein levels observed in our SCLC

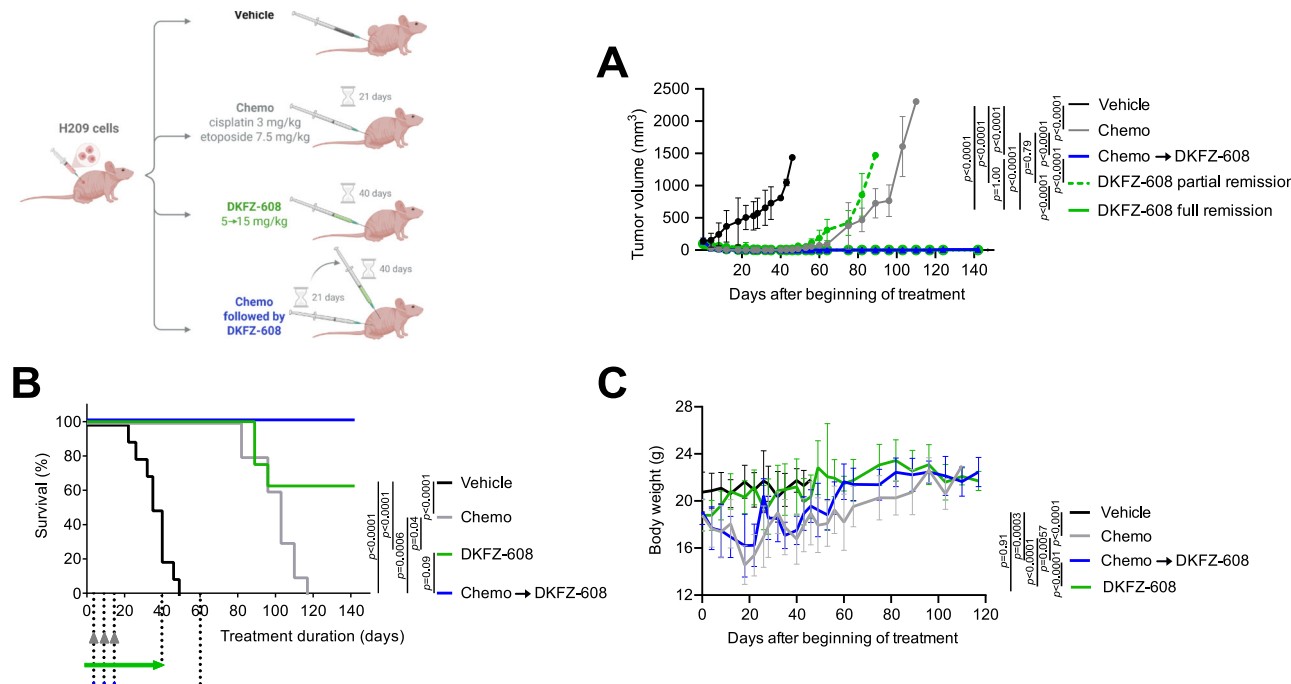

**Fig. 2 | Targeting of TXNRD1 is efficient as a maintenance therapy for chemotherapy-resistant SCLC tumors in vivo.** Human SCLC cell line H209 was transplanted subcutaneously into nude mice (for details see method section Animal experiments). Tumor-bearing mice were distributed into groups ($n = 10$) and treated with vehicle (black), cisplatin/etoposide (gray, chemo, receiving 3 weekly cycles of cisplatin 3 mg/kg, interperitonally, i.p., on Monday and etoposide 7.5 mg/kg, i.p., on Wednesday and Friday) or daily injections of DKFZ-608 (green, starting with dose escalation of 5 -> 15 mg/kg in the first 5 days, continued with 15 mg/kg up to day 40). One group (blue) started on the chemo regimen for 3 weeks followed by DKFZ-608 for 40 days. The tumor size (**A**), survival (**B**) and body weight (**C**) were monitored during the course of the experiment. In (**A**), the mean values ± SD of the tumor size are shown and the group receiving DKFZ-608 mono-therapy (green) is presented as two separate lines in order to allow a better presentation of animals reaching full remission. The comparison of survival groups (**B**) was calculated with log-rank (Mantel-Cox) test. Tumor growth (**A**) and body weight (**C**), presented as mean ± SD, were analyzed using linear mixed-effects models with restricted maximum likelihood estimation (REML) via the lme4 package in R. The model included treatment (categorical), time (continuous; days), and their interaction as fixed effects, with random intercepts for individual animals to account for repeated measurements. Post-hoc pairwise comparisons between treatments were conducted using estimated marginal means in the emmeans package, with Tukey-adjusted $p$ values. Source data are provided as a Source data file.

cell line panel (Fig. S6C), suggesting that low steady-state levels of redox scavenging is determined by low NRF2 activity.

Interestingly, SCLC cells, despite their low ACB expression, show similar or even lower steady-state levels of hydrogen peroxide ($H_2O_2$) and oxygen radicals ($\cdot OH$ and $O_2^-$), as compared to cells with high ROS-scavenging capacity such as high-ACB NSCLC cells (Fig. S7A, B). However, in response to TXNRD1 inhibition, ROS levels increase dramatically in most SCLC cells, while remaining unchanged in high-ACB NSCLC cells, supporting the hypothesis that SCLC cells have a reduced ROS-scavenging capacity (Fig. S7C, D). We also observed that SCLC cells have higher intracellular levels of reactive nitrogen species (RNS), such as nitric oxide (NO) and peroxynitrite ($ONOO^-$), compared to high-ACB cells (Fig. S7E–G), which may be necessary to support their proliferation[15].

We next explored whether the NRF2 system in SCLC is capable of responding to inhibition of KEAP1, the E3 ligase responsible for regulating NRF2 protein levels. As anticipated, treatment with CDDO-Me and other KEAP1 inhibitors led to an increase in NRF2 protein levels in both SCLC cells and non-cancerous Beas-2B and HaCaT cells (Figs. 3A and S8A–C). Among SCLC cell lines treated with the highest non-toxic dose of 50 nM CDDO-Me (Fig. S8D), H82 and H526 exhibited the highest NRF2 protein levels upon KEAP1 inhibition along with marked upregulation of the NRF2 target genes *NQO1* and *AKR1C3*, indicating that the NRF2 regulon is functional in those two cell lines (Fig. 3B, C). However, in contrast to non-cancerous cells, the majority of ACBs and other genes of the redox regulatory system remained at low expression levels, resulting in only marginal activation of pathways involved

in ROS defense (Fig. 3D, E). Consistently, SCLC cells failed to increase their ROS-scavenging capacity to adapt to TXNRD1 inhibition (Figs. 3F and S8E). We specifically examined the role of glutathione (GSH) and nicotinamide adenine dinucleotide phosphate (NADPH), which are essential for maintaining ROS homeostasis. In Beas-2B cells, the depletion of GSH by inhibition of glutathione synthesis (buthionine sulphoximine, BSO) induces cell death, which is rescued by CDDO-Me, although GSH levels remain unchanged (Fig. S9A, B). Notably, the protective effect of CDDO-Me is lost when NADPH regeneration via G6PD is also inhibited with G6PDi-1[29] (Fig. S9B), most likely by reducing the activity of multiple NADPH-dependent antioxidant enzymes like TXNRD1, GSR, AIFM2 (FSP1), AKR1C3, ALDH3A1, and CBR1. Similarly, under conditions of DKFZ-682-induced ROS stress, the protective effect of CDDO-Me is negated only when cells are treated with both BSO and G6PDi-1 (Fig. S9C), suggesting that NADPH-dependent pathways can partially compensate when GSH-mediated redox homeostasis is exhausted. In drug-sensitive SCLC, GSH levels appear elevated compared to more resistant non-cancerous cells (Fig. S9D), suggesting that both steady-state and induced GSH levels may be insufficient to counteract drug-induced ROS stress. The capacity to scavenge ROS seems to be mainly limited by NADPH levels, which are significantly reduced in SCLC cells, in line with their low expression of *G6PD* (Fig. S9E, F). TXNRD1, the DKFZ-682 target, also plays a role in CDDO-Me-mediated cytoprotection. In Beas-2B and HaCaT cell lines, CDDO-Me treatment induces a marked elevation in TXNRD1 enzymatic activity. This upregulation allows CDDO-Me to effectively counteract the suppression of TXNRD1 activity caused by DKFZ-682 when both

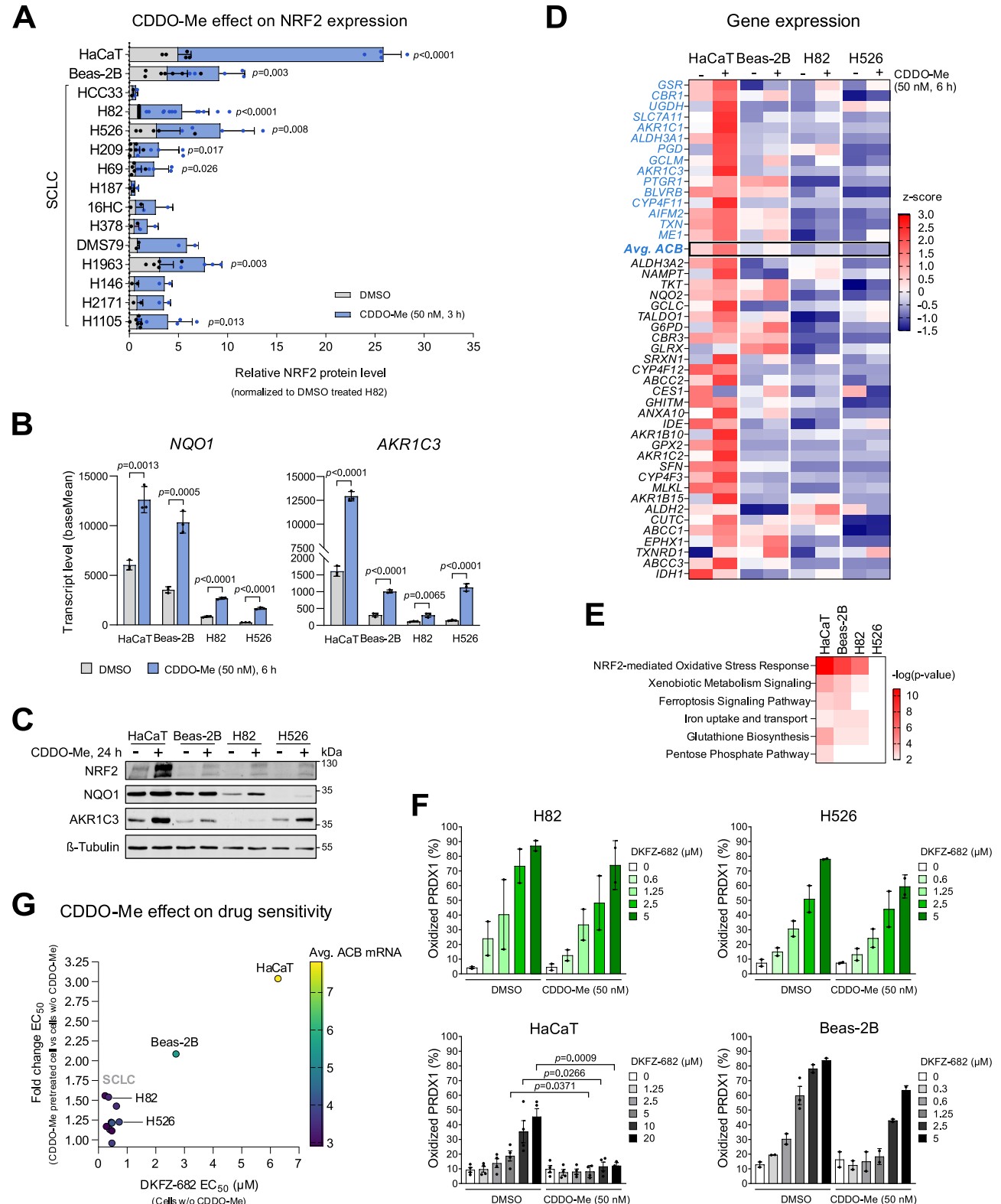

**A** CDDO-Me effect on NRF2 expression

**B** *NQO1* and *AKR1C3* transcript levels

**C** Western blot

**D** Gene expression

**E** Pathway analysis

**F** Oxidized PRDX1 (%) for H82, H526, HaCaT, Beas-2B

**G** CDDO-Me effect on drug sensitivity

compounds are administered together. Notably, this compensatory mechanism is absent in SCLC-derived cells, where CDDO-Me fails to stimulate TXNRD1 activity (Fig. S9G).

Most significantly, the differential response to KEAP1 inhibition observed in SCLC compared to non-cancerous cells effectively broadens the cellular therapeutic window of TXNRD1 inhibition. SCLC cells remain largely sensitive despite NRF2 induction, whereas non-cancerous cells exhibit increased resistance (Fig. 3G). The absence of

resistance induction upon KEAP1 inhibition in SCLC cells was unexpected, given that ROS-induced inactivation or genetic loss of KEAP1 is a common mechanism of resistance to redox-targeting drugs in many cancer types[27]. One possible explanation is the constitutive nuclear activity of BACH1, a transcriptional repressor of a subset of NRF2 target genes[30], or the lack of a positive cofactor like MAFG[31]. However, our detailed analysis revealed that neither BACH1 nor MAFG is predominantly responsible for the lack of adaptation to drug induced ROS

**Fig. 3 | The activation of NRF2 pathway by CDDO-Me protects non-cancerous cells but not SCLCs against the cytotoxic effect of TXNRD1 inhibitors. A** The NRF2 protein level in total cell extracts was analyzed in SCLC and non-cancerous cell lines upon treatment with DMSO (control) or CDDO-Me by immunoblotting. NRF2 expression in DMSO-treated H82 cells was set to 1 for each experiment. Relative data represent mean ± SD of two independent experiments for H2171, H146, DMS79, H378, 16HC, H187, and HCC33, and at least three independent experiments for the other cell lines. Statistical significance was assessed using a two-tailed unpaired $t$-test. Exact $p$ values are indicated in the figure when significant (<0.05). **B** The induction of *NQO1* and *AKR1C3* transcripts upon CDDO-Me treatment in different cell lines was determined by expression profiling ($n = 3$, mean ± SD, two-tailed unpaired $t$-test). Exact $p$ values are indicated in the figure when significant (<0.05). **C** The induction of NQO1 and AKR1C3 proteins upon CDDO-Me (50 nM) treatment was determined by immunoblotting (representative of three independent experiments). **D** The top 45 genes highly correlated with resistance to TXNRD1 inhibition[15] were analyzed by expression profiling of two non-cancerous and two SCLC cell lines treated with CDDO-Me compared to a DMSO control ($n = 3$).

ACBs are labeled in blue. **E** The most upregulated pathways upon CDDO-Me treatment according to a pathway enrichment analysis based on mRNA expression profiling data using Ingenuity Pathway Analysis (IPA, QIAGEN Inc., Redwood City, CA, USA). Canonical pathway enrichment was assessed using right-tailed Fisher's exact test, and $p$ value were adjusted for multiple testing where applicable. **F** Oxidized and reduced levels of PRDX1 protein were analyzed by immunoblotting in cells treated first with CDDO-Me (50 nM) for 24 h and then with the indicated concentrations of DKFZ-682 for 3 h. Bar diagrams summarize the quantitative results from independent experiments ($n = 2$ for H526, H82 and Beas-2B, $n = 4$ for HaCaT, mean ± SD). Statistical significance was assessed using a two-tailed unpaired $t$-test. Exact $p$ values are indicated in the figure when significant (<0.05). **G** The cells were treated with CDDO-Me (50 nM) or DMSO (control). After 24 h, a concentration series of DKFZ-682 was added for another 24 h and the cell viability was measured by the CellTiter-Glo assay. The fold change of $EC_{50}$ for DKFZ-682 upon CDDO-Me compared to cells without CDDO-Me pretreatment was calculated. Source data are provided as a Source data file.

stress (experimental details in Supplementary Results, Figs. S10 and 11).

An alternative explanation for the lack of NRF2 response could be the epigenetic silencing of redox genes. Using public data sets of CpG clusters[32], we observed that several ACB promoters, in particular ME1, PTGR1, PGD and ALDH3A1 show a higher degree of methylation in SCLC and low-ACB NSCLC cells compared to high-ACB NSCLC cells. This suggests that epigenetic silencing could contribute to the failure to adapt to oxidative stress (Fig. 4A). To avoid loss of resolution caused by CpG clustering, we performed methylation analysis on 3 SCLC and 3 NSCLC cell lines, quantifying individual CpG loci. Our results indicate that TXN1, a member of the ACB gene set, may also be subject to epigenetic silencing. We identified a high degree of DNA methylation at positions -995 and -1418 (CpG1 and CpG2, respectively) in the distal promoter region of *TXN1*, both in cell lines and patient tumors. This region has previously been shown to contain an oxidative stress response element[33] (Fig. 4B, C). Consistent with this, TXN1 protein levels are strongly reduced in SCLC compared to NSCLC tumors, which exert lower *TXN1* promoter methylation (Fig. 4D).

To test whether hypermethylation was indeed involved in the lack of NRF2 responsiveness of ACB promoters, we pre-treated H82 and H69 cells with DNA methyl transferase inhibitors decitabine (DAC) or 5-aza-2'-deoxycytidine (AZA), followed by CDDO-Me treatment. Contrary to our expectation, DNA-demethylation did not lead to more efficient drug resistance by NRF2 induction (Fig. S12A), in line with inconsistent induction of hypermethylated ACB promoters (Fig. S12B). This indicates that, despite the high correlation of promoter methylation with ACB expression and drug sensitivity, epigenetic silencing is not a primary cause for the lack of ACB induction by NRF2 and the inability of SCLC to adapt to drug-induced ROS stress.

In summary, SCLC cells suppress ROS defense enzymes through yet unidentified but very likely overlapping layers of epigenetic and transcriptional mechanisms. Consequently, SCLC cells exhibit a reduced ROS-scavenging capacity and, unlike non-cancerous cells, are unable to adapt to drug-induced ROS stress via NRF2-mediated mechanisms.

### Widening the therapeutic window for redox-targeting drugs

Given that SCLC cells are unable to adapt to redox-targeting drugs, such as TXNRD1 inhibitors, we hypothesized that activating the NRF2 pathway would enhance antioxidant defenses specifically in non-cancerous cells, thereby further widening the therapeutic window. To test this hypothesis in vivo, we chose an experimental design which highlighted aspects of a narrow therapeutic window. First, we utilized the hydroxy-substituted derivative DKFZ-682. While this compound demonstrates a fast cytotoxic effect and reaches plasma concentrations sufficient for tumor control (Fig. S5A, B), it is less potent than

DKFZ-608 against SCLC cells and shows reduced selectivity toward non-cancerous cells (Fig. S2D). In addition, its lower maximum tolerated dose (MTD) of 4 mg/kg (Fig. S5C) may limit its therapeutic efficacy. Second, in order to monitor drug efficacy more precisely, we administered DKFZ-682 as monotherapy rather than maintenance therapy, allowing us to eliminate confounding effects from first-line therapy. Lastly, we chose a tumor model based on H526, which exhibits partial resistance to TXNRD1 inhibition (Fig. S2D), to ensure that tumor growth would not be fully suppressed at the 4 mg/kg MTD in unprotected mice.

Upon treatment of mice with the KEAP1 inhibitor CDDO-Me at a concentration previously established for the i.p. route[34] (Fig. 5A), we observed the induction of NRF2 proxies, such as NQO1, GSR and GCLM in the liver, kidney and lung (Fig. 5B). On a physiological level, we observed a reduction of stress markers, including blood urea nitrogen (BUN), alanine aminotransferase (ALT), and aspartate aminotransferase (AST) (indicators of kidney and liver damage), induced by DKFZ-682 (Fig. 5C), supporting the hypothesis of organ protection. Consistent with this, we were able to increase the therapeutic dose by 2.5-fold without causing disproportionate stress or weight loss (Fig. 5A, right panel). Remarkably, stress markers remained suppressed over a period of 3 weeks, indicating that NRF2-mediated organ protection can be maintained long-term without desensitization (Fig. 5D).

We next compared tumor growth reduction in unprotected (MTD 4 mg/kg DKFZ-682) with CDDO-Me-protected mice (MTD 10 mg/kg DKFZ-682) (Fig. 6A). It is important to note that treatment of unprotected mice with 10 mg/kg DKFZ-682 was not tolerable and therefore ineligible to study antitumor effects (Fig. S5C).

Upon detection of palpable tumors, mice were randomized into groups receiving either vehicle or compounds daily for 3 consecutive weeks. The groups treated with CDDO-Me received injections for 2 days prior to the initiation of DKFZ-682 treatment (Fig. 6A). As anticipated, the antitumor response of DKFZ-682 at 4 mg/kg (Fig. 6B, gray line) was only moderate, resulting in a median survival of 20 days compared to 15.5 days in the vehicle-treated group ($p = 0.4$) (Fig. 6B, black line). In contrast, the group treated with high-dose DKFZ-682 (10 mg/kg, Fig. 6B, red line) showed enhanced efficacy, with median survival increasing nearly 2-fold from 15 days in the CDDO-Me-only group (blue line) to 27 days in the CDDO-Me/high-dose DKFZ-682 group ($p = 0.001$). These findings confirm our hypothesis that tolerance to an increased therapeutic dose can indeed translate into improved tumor-suppressive effects. The differential impact of CDDO-Me on tumor versus normal cells is reflected in intracellular TXNRD1 activity, which serves as an indicator of DKFZ-682 target engagement. In tumors, DKFZ-682 treatment reduced TXNRD1 activity by 3-fold, whereas CDDO-Me caused only a slight but nonsignificant increase,

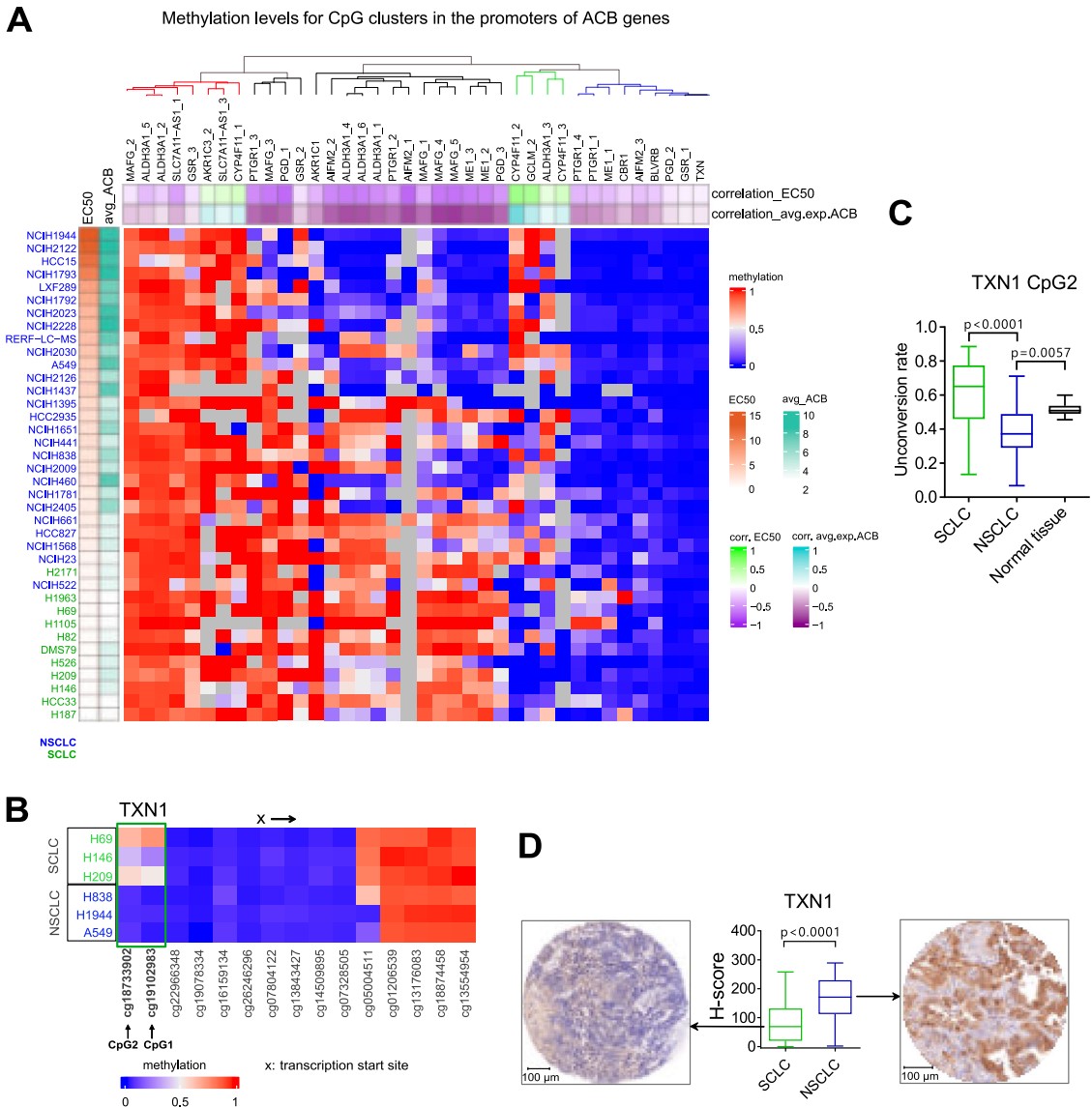

**Fig. 4 | Increased promoter methylation of ACB genes in SCLCs correlates with the sensitivity to TXNRD1 inhibitor. A** Methylation values of CpG clusters previously identified within promoters of *MAFG* and ACB gene set for NSCLC and SCLC cell lines profiled in DepMap project (CCLE CpG clusters were arranged by hierarchical clustering algorithm using euclidean distance metric and complete linkage method (default clustering settings of ComplexHeatmap package) into 4 groups of promoter methylation profiles distinguished by dendrogram colors: red – hypermethylated across all cell lines; black – hypermethylated in low-ACB cells; green – hypermethylated in high-ACB cells; blue – hypomethylated across all cell lines). Left annotations: $EC_{50}$ – DKFZ-682 $EC_{50}$ per cell line (rows are sorted based on these values); avg_ACB – average expression of genes from the ACB gene set per cell line. Top annotations: first row – correlation between vector of cell lines' methylation values and vector of cell lines' DKFZ-682 $EC_{50}$ values per each CpG cluster; second row – correlation between vector of cell lines' methylation values and vector of cell lines' average expression of genes from ACB gene set per each CpG cluster. **B** The methylation status of genomic DNA. The heatmap depicts beta values with 0

meaning no methylation and 1 indicating high methylation status. The probes are indicated by Illumina IDs and sorted according to their localization in the *TXN1* gene. CpG1 and CpG2 are indicated by their distance to the transcription start site. **C** Methylation levels at CpG2 within the *TXN1* promoter were analyzed in samples from patients with SCLC ($n = 40$) and NSCLC ($n = 46$) using bisulfite sequencing PCR. Normal lung tissue samples derived from NSCLC patients ($n = 10$) served as controls. Data are presented as box plots showing the median (center line) and the minimum and maximum values (whiskers). Box plots depict conversion ratios, where 0 indicates no methylation and 1 indicates high methylation (differences analyzed by a two-tailed unpaired *t*-test). **D** *TXN1* levels in SCLC ($n = 72$) and NSCLC ($n = 88$) specimens were determined by digital evaluation of *TXN1* immunohistochemical staining using tissue microarrays. H-scores were calculated based on staining intensity (analyzed by a two-tailed unpaired *t*-test). A tumor specimen representative of the median H-score for each group is shown. Data are presented as box plots showing the median (center line) and the minimum and maximum values (whiskers). Source data are provided as a Source data file.

compared to tumors resected from untreated animals. This aligns with the largely unresponsive nature of the NRF2 regulon in SCLC cells. As a result, CDDO treatment failed to compensate the loss of TXNRD1 activity caused by DKFZ-682 treatment, leaving tumor cells susceptible to oxidative stress. Notably, these changes in TXNRD1 activity occurred independently of protein levels, which remained largely

unaffected by drug treatments (Fig. 6C). In contrast, in normal tissue, exemplified by liver, CDDO treatment led to a nearly 3-fold increase in TXNRD1 protein levels, corresponding to a more than 4-fold increase in TXNRD1 activity. DKFZ-682 treatment only partially reverted this gain of target activity, allowing for a net-increase in ROS buffer capacity by TXNRD1 (Fig. 6D).

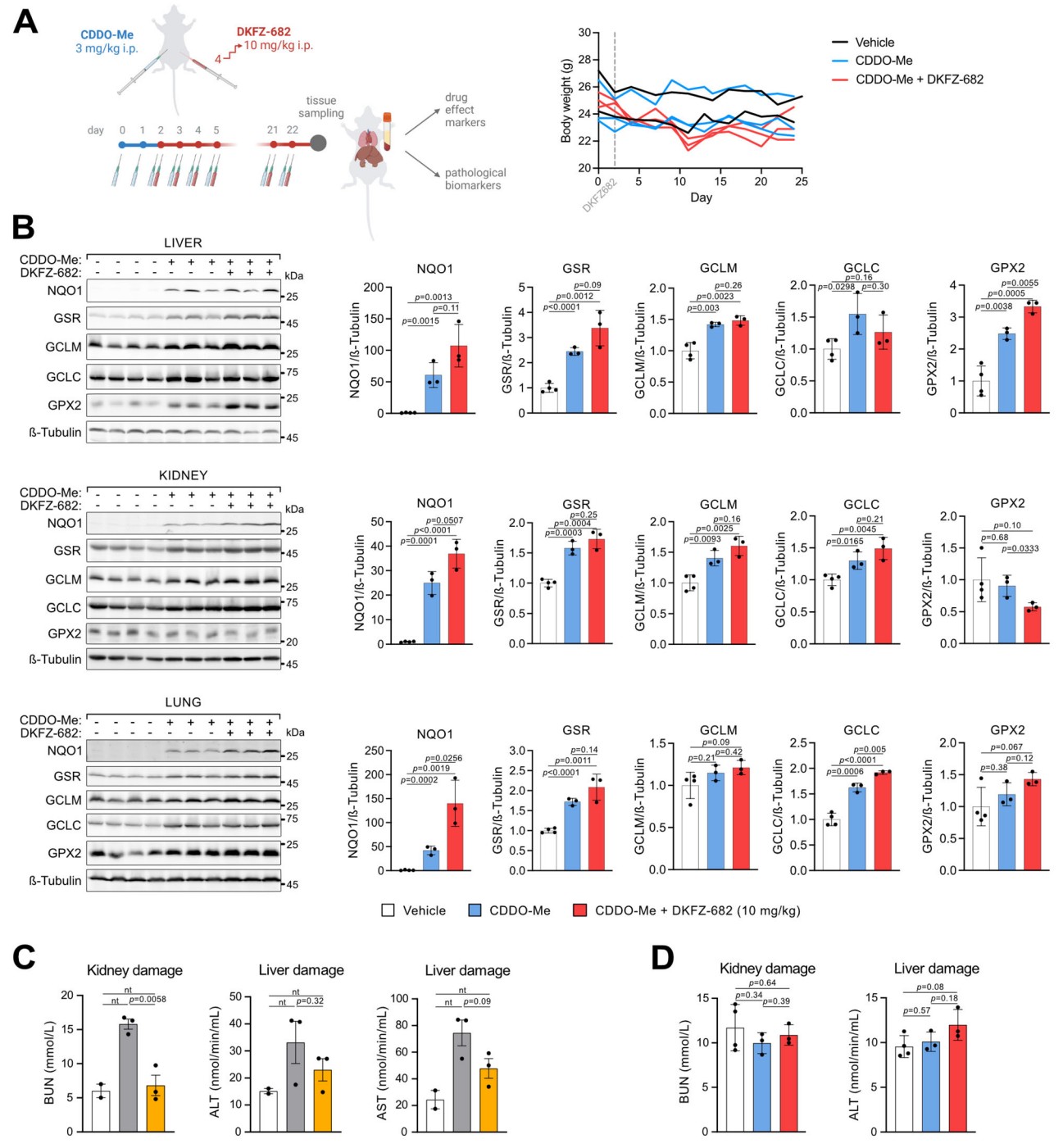

**Fig. 5 | CDDO-Me widens the therapeutic window for a TXNRD1-targeting inhibitor in vivo. A** After CDDO-Me pre-treatment, the NSG mice were daily injected intraperitoneally with DKFZ-682, gradually increasing the dose from 4 mg/kg to 10 mg/kg that was then maintained for up to 3 weeks. The body weight of individual animals was monitored during the course of the experiment. The comparison of weight gain in individual treatment groups was performed by non-parametric, unpaired *t*-test. **B** The induction of drug effect markers of CDDO-Me in mouse tissues was investigated by immunoblotting (vehicle: *n* = 4 animals, treated groups: *n* = 3 animals). The bar graphs show quantification of signal intensities (mean ± SD, pairwise differences compared by two-tailed unpaired *t*-test). **C** The organ damage markers were determined in the blood plasma of mice after a single

injection of 4 mg/kg DKFZ-682 with or without CDDO-Me pre-treatment for 48 h (vehicle: *n* = 2 animals, treated groups: *n* = 3 animals), mean ± SD, differences between the treated groups analyzed by a two-tailed unpaired *t*-test; nt, not tested due to an insufficient number of biological replicates). As markers of kidney and liver damage, blood urea nitrogen (BUN) and alanine aminotransferase (ALT) as well as aspartate aminotransferase (AST) were measured, respectively. **D** The organ damage markers were determined in the blood plasma after the 3-week treatment as presented in (**A**) (mean ± SD; vehicle: *n* = 4 animals, treated groups: *n* = 3 animals; differences between groups analyzed by a two-tailed unpaired *t*-test). As markers of kidney and liver damage, BUN and ALT were measured, respectively. Source data are provided as a Source data file.

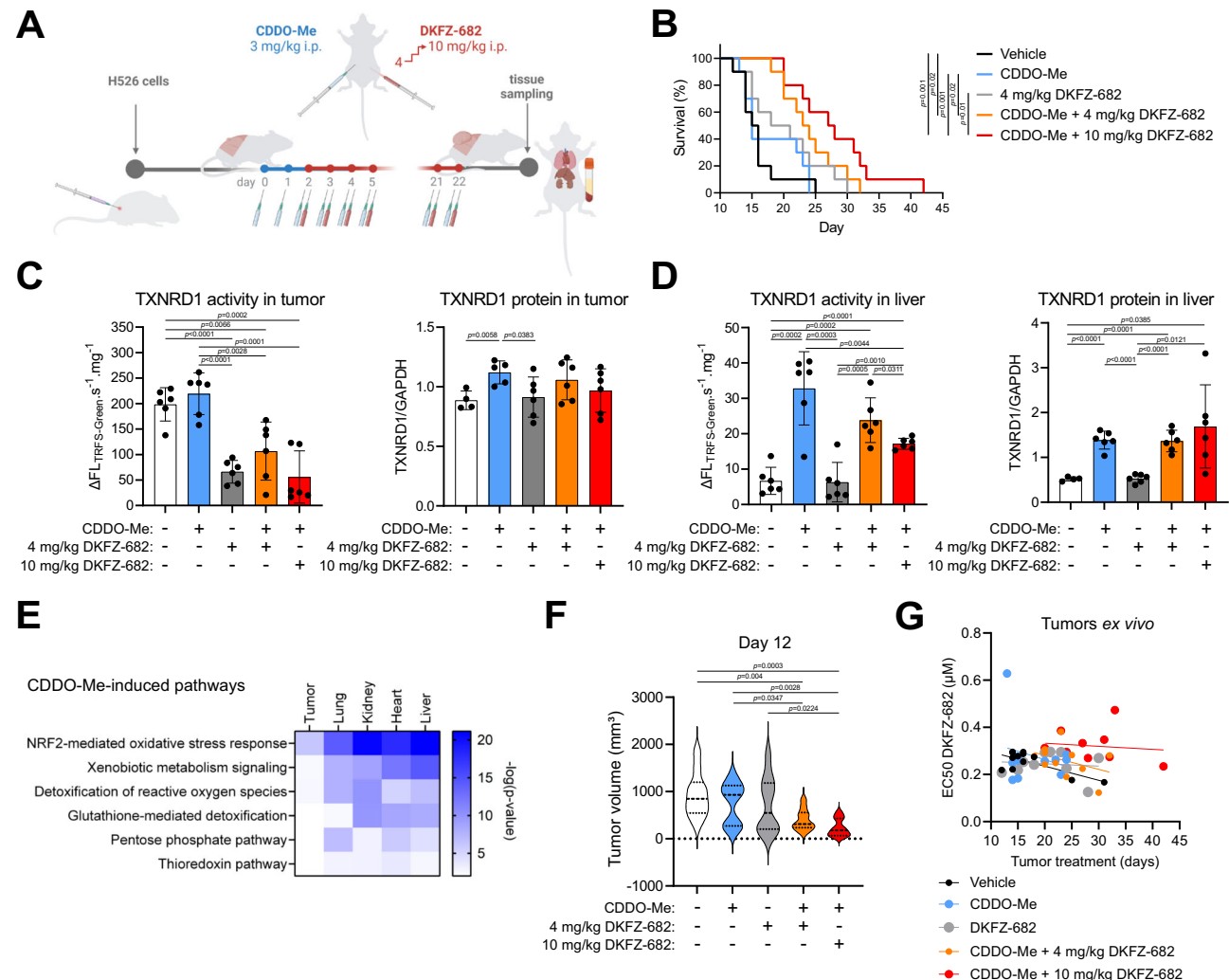

**Fig. 6 | A selective activation of NRF2 pathway by CDDO-Me in normal tissues expands the therapeutic potential of TXNRD1 inhibitors in vivo. A** As a tumor model, NSG mice were engrafted subcutaneously with the human SCLC cell line H526 (for details see "Method" section Animal experiments). After CDDO-Me pre-treatment, the NSG mice were daily injected intraperitoneally with DKFZ-682, gradually increasing the dose from 4 mg/kg to 10 mg/kg that was then maintained for up to 3 weeks. The tumor size was monitored daily for reaching the humane end-points. **B** Mice were randomly distributed into groups ($n = 10$) that were daily injected with vehicle, CDDO-Me (3 mg/kg), a low dose of DKFZ-682 (dose escalation from 1 mg/kg to 4 mg/kg), CDDO-Me (3 mg/kg) and a low dose of DKFZ-682 (4 mg/kg), CDDO-Me (3 mg/kg) and a high dose of DKFZ-682 (dose escalation from 4 mg/kg to 10 mg/kg). The plot shows the survival of individual groups (compated by the log-rank (Mantel-Cox) test). **C, D** Impact of CDDO-Me and DKFZ-682 treatment on TXNRD1 activity and protein levels in tumor and liver tissue derived from treated mice (mean ± SD; $n = 6$ animals per group, measured in technical duplicates). The residual enzymatic activity of TXNRD1 in resected tumor and liver tissue was

assessed by the activity probe TRFS-Green. Significance of differences was calculated with a two-tailed unpaired $t$-test. **E** The pathways upregulated by CDDO-Me were analyzed in mouse tissues and tumors, performing RNAseq. Samples from tumor-bearing mice (H526 tumors) that were injected with CDDO-Me (3 mg/kg) for 4 days were compared to vehicle ($n = 5$). Pathway enrichment analysis was performed using Ingenuity Pathway Analysis (IPA, QIAGEN Inc., Redwood City, CA, USA). Canonical pathway enrichment was assessed using right-tailed Fisher's exact test, and $p$ values were adjusted for multiple testing where applicable. **F** The tumor size measured 12 days after the daily mouse treatment was started ($n = 10$, one-way ANOVA $p = 0.0025$; pairwise differences compared by two-tailed unpaired $t$-test). **G** The sensitivity to DKFZ-682 was tested in explanted tumors after mouse treatment for the indicated time. Tumor cells were treated with a concentration series of DKFZ-682 for 24 h and cell viability was quantified using CellTiter-Glo. Each dot represents a tumor from one mouse measured in a triplicate. Source data are provided as a Source data file.

Pathway analysis of RNAseq data from organs and tumors demonstrated substantial induction of the NRF2 regulon in the lung, kidney, heart, and liver, in particular genes involved in redox homeostasis, but only a weak response in tumors (Fig. 6E). This differential response was not due to reduced efficacy of CDDO-Me in tumors since we detected substantial induction of the NRF2 proxy GPX2, at both protein (Fig. S13A) and RNA levels (Fig. S13B). Interestingly, mice that did not receive CDDO-Me protection during the 3 weeks of low-dose DKFZ-682 treatment failed to express higher levels of ROS response markers such as NQO1, GPX2, GCLM, or GSR in their organs (Fig. S13C), suggesting that sustained NRF2-mediated adaptation to

drug-induced ROS stress does not occur without pharmacological KEAP1 inhibition.

Previous studies using mouse models of lung and prostate cancer have demonstrated that CDDO-Me exhibits antitumor effects and enhances the efficacy of combination therapies (reviewed in ref. 35). Contrary to these findings, our results indicated a positive interaction between CDDO-Me and the cytotoxic drug only during the early phase of the trial (Fig. 6F), however without a corresponding improvement in long term survival (Fig. 6B, orange versus gray line). Importantly, tumor cells isolated from CDDO-Me-treated mice did not show significant signs of acquired resistance to DKFZ-682 (Fig. 6G) and

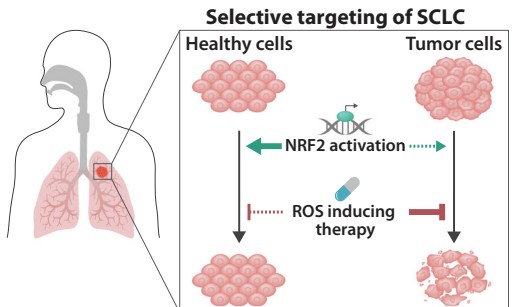
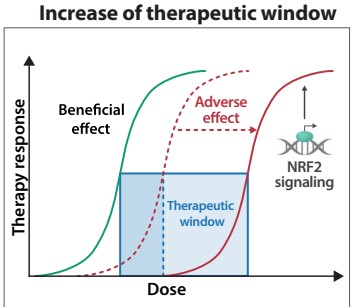

**Fig. 7 | Schematic illustration depicting the differential cytoprotective effect of pharmacological NRF2 pathway activation in normal versus malignant cells.** Selective induction of NRF2 signaling in healthy tissues enhances antioxidant capacity and confers resistance to ROS-inducing agents, such as TXNRD1 inhibitors (presented in this study). In contrast, tumor cells remain susceptible to ROS induction due to impaired or non-responsive NRF2 activation. This tissue-selective protection mitigates off-target toxicity and reduces adverse effects, thereby enabling dose escalation and improved therapeutic efficacy.

remained unresponsive to NRF2 induction when tested ex vivo. This suggests that CDDO-Me treatment did not promote the development of resistance in SCLC tumors (Fig. S13D).

Taken together, our data provide evidence that selective activation of the KEAP1/NRF2 pathway in normal tissues minimizes dose-limiting toxicity and expands the therapeutic potential of TXNRD1 inhibitors (Fig. 7). This approach paves the way for a paradigm shift in the treatment landscape of SCLC and in the use of ROS-inducing drugs in cancers with an impaired NRF2 pathway.

## Discussion

Despite decades of intense research and numerous clinical trials investigating targeted therapeutics, drug combinations, and immune checkpoint inhibitors, treatment options for SCLC patients remain limited. Maintenance therapy, however, holds significant promise for advancing SCLC treatment. Standard therapy with cisplatin/etoposide is highly effective, often resulting in near-complete remission, thus creating an ideal window for suppressing residual cancer cells. Unfortunately, none of the therapies tested so far have prevented tumor relapse. Successful maintenance therapy must target a critical vulnerability shared across all tumor cells present after first-line therapy and remain effective against emerging resistance mechanisms. The limited success of previous attempts suggests that the drug combinations have either failed to effectively target such a vulnerability in first-line-treated SCLC cells or were compromised by the same resistance mechanisms that blunted the efficacy of cisplatin/etoposide.

The TXNRD1 inhibitor DKFZ-608 exhibits a unique activity profile. In tumors derived from therapy-naïve cells, we observed sustained remission in mice. Remarkably, TXNRD1 inhibition appears fully compatible with first-line therapy, as it neither delayed the recovery of animals from chemotherapy-induced side effects nor caused cumulative toxicity, even after 40 consecutive daily doses (Fig. 2C). It is also noteworthy that no signs of neurotoxicity were observed, even though redox homeostasis plays a crucial role in neurotransmission and neuronal cells possess relatively low steady-state levels of ROS-scavenging enzymes[36]. We identified several factors contributing to the remarkable efficiency of TXNRD1 inhibition on SCLC tumors. First, targeting redox homeostasis via TXNRD1 inhibition appears to hit an Achilles heel in SCLC, which is hallmarked by an inherently low steady-state capacity to scavenge drug-induced ROS stress. Additionally, SCLC cells are unable to upregulate their multimodal ROS-scavenging pathways to adjust to this stress (Fig. 3). Another, particularly important factor is that SCLC cells retain their low ROS buffer capacity and sensitivity to TXNRD1 inhibition even after acquiring resistance to chemotherapy. This is evidenced by the uniform potency of TXNRD1 inhibition across a panel of cell lines and circulating cancer cells, which exhibit heterogeneous responses to cisplatin (Fig. 1). We did not expect this, as cisplatin not only affects DNA replication but also induces ROS, thus contributing to its efficacy on tumor cells and systemic toxicity (reviewed in refs. 37, 38). The events leading to cisplatin resistance in SCLC are still not fully understood, with conflicting reports on adaptive changes in redox systems (reviewed in ref. 39). Notably, the fact that patient-derived paired samples of pre- and post-therapy SCLC cells show comparable levels of ACBs encourages us to hypothesize that breaking of cisplatin-resistance can also be achieved in real-world SCLC maintenance therapy.

A third reason for the high potential of TXNRD1-targeting drugs is their efficacy, which is independent of the molecular subtype of SCLC. This contrasts with the heterogeneous toxicity of GPX4 inhibitors, which have previously shown resistance in certain NE subtypes[40]. In agreement with this earlier report, we observed up to a 250-fold difference in drug sensitivity to RSL3 in our SCLC cell line panel, highlighting the more homogeneous activity profile of TXNRD1 inhibitors (Fig. S2A, B, D).

Why do SCLCs exhibit a low ROS-scavenging capacity and how is this achieved? Our data show that SCLC cells maintain low levels of basal ROS such as $O_2^-$, •OH, and $H_2O_2$ (Fig. S7). One possible explanation for this phenomenon is that SCLC cells generate lower levels of $O_2^-$ through NADPH oxidases or mitochondrial respiration, which reduces the need for a high ROS-scavenging capacity to sustain the redox homeostasis required for rapid proliferation. This hypothesis is consistent with a recent report showing that SCLC cells have significantly fewer mitochondria, compared to NSCLC[41]. Reduced enzymatic ROS-scavenging could, at least under homeostatic conditions, be compensated by elevated levels of NO, which we detect in SCLC cells and which may act as a ROS scavenger by reacting with superoxide to form the non-radical, slow-reacting molecule peroxynitrite ($ONOO^-$, reviewed in refs. 42 and 43). Additionally, SCLC cells may utilize hydropersulfide-based mechanisms, which have recently been shown to eliminate intracellular radicals[44]. Notably, among lung cancer cell lines, SCLC-derived cells express higher levels of cystathionine beta-synthase (CBS), the rate-limiting enzyme of the trans-sulfuration pathway, involved in the production of cysteine and hydrogen sulfide, which are essential for the formation and function of hydropersulfides (reviewed in ref. 45).

An alternative hypothesis is inspired by the finding that PNECs, the proposed cells-of-origin for NE SCLC, also exhibit a low-ACB profile (Fig. 1C). Oxygen sensing by PNECs relies on the precise regulation of $H_2O_2$, which is produced by an oxygen-consuming NADPH oxidase. $H_2O_2$ functions as a second messenger that modulates the activity of oxygen-sensitive $K^+$ channels[46]. To accurately sense changes in $H_2O_2$ levels both under normoxic and hypoxic conditions, it is plausible that PNECs may limit their response to fluctuating $H_2O_2$

concentrations by avoiding the activation of adaptive ROS-scavenging pathways.

Currently, it is enigmatic how PNECs or SCLC cells establish a low ROS buffer status. Redox homeostasis is regulated by multiple transcription factors[47], with NRF2 playing a major role both in steady-state conditions and, more prominently, under ROS-induced stress. The fact that *NRF2* mRNA levels are comparable to non-cancerous cells but NRF2-response genes, like *NQO1* and the ACB gene set, are under-expressed suggests that SCLC cells have constitutively low NRF2 activity, most likely mediated by post-translational mechanisms like KEAP1-mediated degradation. An additional factor potentially leading to the inhibition of the NRF2 regulon is BRD4, which is overexpressed in SCLC-derived cells and acts as an activator of KEAP1 expression[48]. Contrary to our expectation, BRD4 inhibition by JQ1 did not decrease the toxicity of DKFZ-682 nor allowed CDDO-Me induced resistance (Fig. S14).

The limited capacity to scavenge drug-induced ROS observed under steady-state conditions is only one aspect contributing to the effectiveness of TXNRD1 inhibitors. An equally important factor in real-world therapy is adaptation to drug-induced ROS stress. Our data demonstrate that SCLCs are unable to adjust their ROS buffer capacity to drug-induced ROS challenge. We still have only a fragmented understanding of the underlying mechanisms. Our findings show that the hypermethylation of ACB promoter genes, while strongly correlated with ACB expression and drug sensitivity, is not a primary cause of impaired adaptation to ROS stress since pre-treatment with demethylating drug did not unlock the NRF2 regulon (Fig. S12A). We hypothesized that in addition to an epigenetic repression, SCLC may lack a cofactor needed by NRF2 for efficient ACB induction. *FOSL1* (FRA1), which is expressed at very low levels in SCLC and has been linked to reduced expression of certain NRF2 response genes[49], was considered a potential candidate. However, induced expression of *FRA1*, in combination with epigenetic modulations, did not enable SCLC cells to translate CDDO-Me treatment into drug resistance (Fig. S15).

Despite the fact that TXNRD1 inhibition shows more pronounced toxicity towards SCLC compared to non-cancerous cells, both in cell culture and mice, we cannot rule out the possibility that dose-limiting toxicity in certain patients may lead to reduced efficacy, even in a susceptible entity like SCLC. However, the observation that SCLCs do not respond to NRF2 induction, while non-cancerous cells become resilient against drug toxicity, enabled us to demonstrate that the therapeutic window for TXNRD1 inhibition can be substantially expanded (Fig. 6). Previous studies have shown that organ protection by KEAP1 inhibitors/NRF2 inducers is effective for cisplatin[50] and doxorubicin[51], but a discriminative effect on cancer cells has not been reported before and was, up to this report, counterintuitive.

NRF2 inducers such as CDDO-Me and DMF were primarily developed to treat inflammation associated with chronic diseases (reviewed in ref. 52) and have shown a toxicity profile that seems compatible with cancer treatments. Furthermore, studies on the immunomodulatory effects of NRF2 pathway activators within the tumor microenvironment (TME) have shown that CDDO-Me attenuates immunosuppression by promoting shifts in macrophage polarization and reducing the numbers of regulatory T cells while simultaneously increasing the number of cytotoxic T cells[53,54]. These effects lead to more enhanced chemotherapy efficiency in a murine model of lung cancer[55], underscoring the potential of NRF2 activation not only in widening the therapeutic window but also in actively reshaping the immune landscape within the TME.

Having demonstrated that TXNRD1 is a critical vulnerability of SCLC, the question arises whether there are suitable inhibitors available for clinical trials. Auranofin, initially developed for the treatment of rheumatoid arthritis, has been tested repeatedly in clinical trials but has shown limited success. Based on our data, we attribute this primarily to the lack of appropriate stratification for susceptible, low-ACB tumors. In addition, the molecule itself has physicochemical shortcomings that pose challenges, including poor bioavailability, limited stability, and dose-limiting side effects[56]. A non-metal-containing compound discovered by the Arner lab has shown in vivo activity but requires optimization to enhance its stability[57]. DKFZ-608 on the other hand, has shown superior activity and selectivity, compared to auranofin, with favorable bioavailability in mouse models. However, further toxicity and pharmacokinetic studies in non-rodent species are required to provide data for advancing DKFZ-608 into human clinical trials.

In summary, our study demonstrates that low expression of genes involved in ROS-scavenging in SCLC results in the accumulation of lethal ROS levels when a key enzyme of redox homeostasis, TXNRD1, is inhibited. SCLC cells not only experience a rapid exhaustion of their ROS-scavenging capacity but also fail to adapt to oxidative stress by upregulating NRF2-mediated ROS defense.

We conclude that TXNRD1 inhibition is a highly effective strategy for maintenance therapy following cisplatin/etoposide treatment. This is because the resistance mechanisms that emerge in residual cancer cells during first-line therapy do not compromise the effectiveness of TXNRD1 inhibition. Lastly, we propose that combining TXNRD1 inhibitors with NRF2 inducers will maximize the therapeutic potential of TXNRD1 inhibition in maintenance therapy by protecting healthy tissues and enhancing the selectivity towards cancer cells (Fig. 7).

## Methods

### Synthesis of DKFZ-608

[Au (diethyldithiocarbamate)]$_2$ (DKFZ-608): To a solution of aurothiomalate (2.0 g, 5.13 mmol, 1.0 eq.) in H$_2$O (100 mL) was added a solution of tetraethyldithiuram disulfide (1.52 g, 5.13 mmol, 1.0 eq.) in EtOH (100 mL). An orange precipitate formed. The suspension was stirred at room temperature for 3 h, the obtained precipitate was filtered through a sintered glass frit (pore size 4), thoroughly washed with water, and dried under high vacuum overnight. The crude product was recrystallized from hot DMF to afford 1.21 g (68% yield) of DKFZ-608 as orange needles. Elemental analysis [M]: calculated C, 17.39; H, 2.92; N, 4.06; found C, 17.45; H, 3.02; N, 4.22. Analysis of the obtained crystals by X-ray crystallography revealed a molecular unit as previously reported[58].

Due to the low solubility of this compound, a solution of DKFZ-608 (5 mM) was prepared in 150 mM ß-Cyclodextrin sulfobutyl ether/PBS[17] by heating up in a water bath at 100 °C and repeated vortexing every 10–15 min until fully dissolved. The stock solution was stable for more than 2 months at 4 °C and continuously checked for activity in biochemical and cellular assays. In order to use equivalent amounts of gold atoms when comparing DKFZ-608 to auranofin, we calculated concentrations according to gold content.

### Pharmacokinetics of DKFZ-608 and -682

Upon oral treatment with drugs, 5 animals were sacrificed at each time point 0, 20, 40, 80, 160, 320 min and 24 h. Blood was collected, mixed with heparin and centrifuged to obtain plasma. 0.25 ml of each sample was mixed with 1 ml of concentrated HNO$_3$ und 0.2 ml H$_2$O$_2$ and incubated in a closed system by a microwave (Ethos 1200 MLS) at 220 °C and 50 bar for 1 h. The digestion solution was filled up to 10 ml with purified water. Samples were analyzed by inductively coupled plasma optical emission spectrometry (Agilent 720 ICP-OES) using a wavelength of 242.794 nm. Due to the absence of a certified reference material (CRM) for Au in cells, quality of measurement was controlled by the following procedures: Samples ($n = 5$) were spiked with 20 µg/l Au. Subsequent measurements showed a recovery between 98% and 101.4%.

## Cell culture and siRNA transfection

Human SCLC suspension cell lines (ATCC: NCI-H69 HTB-119, NCI-H82 HTB-175, NCI-H526 CRL-5811, NCI-H209 HTB-172, NCI-H1105 CRL-5856, NCI-H187 CRL-5804, DMS79 CRL-2049, NCI-H146 HTB-173, NCI-H2171 CRL-5929, NCI-H1963 CRL-5982, NCI-H378 CRL-5808, Beas-2B CRL-3588; DSMZ: HCC33 487), SCLC-16HC[59] and the NSCLC cell line (ATCC: NCI-H1944 CRL-5907) were cultivated in RPMI-1640 (Gibco). Other NSCLC cell lines (ATCC: A549 CRM-CCL-185 and NCI-H838 CRL-5844) were grown in low-glucose and high-glucose DMEM (Gibco), respectively. Resistance to cisplatin in H69CPR and H526CPR was developed by gradually increasing the concentration of cisplatin in the growth medium once a week over a period of six months. H209CPR was isolated from relapsed xenograft tumors in mice after cisplatin therapy. The human non-cancerous Beas-2B (ATCC CRL-3588) and HaCaT (AddexBio T0020001) were grown in DMEM/F-12 (Gibco) and in high-glucose DMEM (Sigma-Aldrich) medium, respectively. Prior to seeding for experiments, SCLC cells were harvested by centrifugation (4 min at $130 \times g$ at room temperature). H838, A549, H1944, Beas-2B and HaCaT cells were split by incubation with trypsin-EDTA (0.25 %) for 5 min at 37 °C. All the media contained 10% fetal bovine serum (FBS-12A, Capricorn Scientific), 100 U/mL penicillin and 100 μg/mL streptomycin (Gibco). MAFG overexpressing SCLC cells were cultured in medium with puromycin (0.5 μg/mL) to make sure that only cells that carry insertion survive. The cells were cultivated at 37 °C in a 5% $CO_2$ atmosphere. Cultures derived from circulating cancer cells were handled as described in ref. 60.

Cell lines were tested negative for *mycoplasma* contamination (Eurofins Genomics). Cell lines were authenticated using Multiplex Cell Authentication by Multiplexion (Heidelberg) as described in ref. 61.

Explanted tumors were stored in a cryopreservation medium (10% DMSO, 36% FBS, 54% RPMI-1640) in liquid nitrogen. After thawing, cells were filtered through EASYstrainer 40 μm (Greiner), washed with PBS and cultivated in RPMI-1640 (Gibco) supplemented with 10% fetal bovine serum (FBS-12A, Capricorn Scientific), 200 U/mL penicillin and 200 μg/mL streptomycin (Gibco).

Small interfering RNA (siRNA) to knock down target genes was ordered from siTOOLS (Planegg, Germany). Transient transfection was performed using Lipofectamine® RNAiMAX Transfection Reagent according to the manufacturer's instructions (Thermo Fisher Scientific, Germany).

## Chemical compounds

Chemical compounds used in this study are listed in Supplementary Table S1.

## Generation of cisplatin-resistant cells

Cisplatin-resistant (CPR) SCLC cell lines were established from each original parental cell line through continuous exposure to cisplatin over a period of approximately 6 months. Each cell line ($2 \times 10^6$ cells in 6 mL culture medium per flask) was treated with freshly prepared cisplatin for 72 h. The initial cisplatin concentration was individually optimized for each cell line (e.g., 0.25 or 0.5 μM for H69 and 0.1 μM for H526 cell lines). Following treatment, the medium was replaced, and the cells were allowed to recover in fresh, drug-free medium for an additional 96 h. The treatment was repeated sequentially until the cells demonstrated adaptation to the currently treated drug concentration. Adaptation was assessed by observing changes in the color of the culture medium (from pink to yellow, indicating culture medium consumption) and an increase in cell aggregate size. Upon confirmation of adaptation, the drug concentration was incrementally increased. For the experiments, cells cultured for 2 weeks in cisplatin-free medium were used. Cell resistance to the final cisplatin treatment concentration (2 μM for H69 and 1 μM for H526) was confirmed using CellTiter-Glo assay.

## Cell viability assays and determination of EC$_{50}$

Cells were seeded in white 96-well plates at a density of 5000 cells/well in 100 μL of medium. After 24 h, test compounds were added (all concentrations in triplicate) without medium change. After an incubation for the indicated time period, viable cells were quantified using either CellTiter-Glo or CellTiter-Blue cell viability assay (Promega) according to the manufacturer's instructions. $EC_{50}$ values were calculated from dose response curves by GraphPad Prism using the following operations: normalization by averaging the replicates of untreated cells and setting their mean as 100%. On normalized data, we performed non-linear regression analysis using the setting "log(inhibitor) vs. response – Variable slope (four parameters)". Cells were treated with a concentration series of cisplatin (72 h), DKFZ-608 (72 h), DKFZ-682 (24 h) and auranofin (24 h) and the cell viability was quantified by the CellTiter-Glo or CellTiter-Blue assay (for H209/H209CPR).

Circulating SCLC cells were seeded at a density of 10,000 cells in 100 μL of medium in transparent 96-well plates. Test compounds were added in triplicate and the plates were incubated for four days under tissue culture conditions and viable cells detected using a modified MTT assay (EZ4U, Biomedica). $EC_{50}$ values were determined using Origin 9.1 software (OriginLab).

## Monitoring of TXNRD1 activity in tissue homogenates with TRFS-Green

Pieces of tissue (mouse liver and tumor) were first homogenized in PBS supplemented with a protease inhibitor cocktail (Serva) using Tissue-Lyser II (Qiagen) with one stainless steel bead (5 mm) per sample ($4 \times 30$ s at 25 Hz). Filtered homogenates were aliquoted and stored at −80 °C. After thawing, homogenates were sonicated with a probe sonicator for 10 s on ice (10% amplitude). Protein concentration was determined by DC Protein Assay (Bio-Rad) using BSA as a standard and liver and tumor homogenates were diluted with a TE buffer (50 mM Tris-Cl, 1 mM Na-EDTA, pH 7.4) to the concentration of 2 mg/mL and 0.5 mg/mL, respectively.

For the TXNRD1 activity assay on a 96-well plate, 100 μL of the diluted homogenate was mixed with 200 μM NADPH and 10 μM TRFS-Green. Fluorescence was recorded with the CLARIOstar plate reader in 2 min intervals for 2 h at 37 °C using the filters 438/15 for excitation and 538/20 for emission. Each sample was measured as a technical duplicate. The slope of Δ fluorescence per s was calculated in the linear phase, which was the time range of 60–120 min and 34–94 min in the case of liver and tumor, respectively. TXNRD1 activity was expressed as Δ fluorescence per s per mg of protein.

## Gel electrophoresis and western blotting

For total protein extraction, cells were harvested using Pierce™ RIPA buffer (Thermo Scientific, 89900) supplemented with a protease inhibitor cocktail (Serva), 100 U/ml benzonase (Merck) and, if necessary, PhosSTOP phosphatase inhibitor cocktail (Roche). Protein concentrations were measured by using Pierce BCA Protein Assay Kit (Ref. 23225).

Tissue homogenates were prepared from tissue pieces stored at −80 °C. Samples were quickly weighed and ice-cold PBS supplemented with the protease inhibitor cocktail was added to each sample to make 10% homogenate while using TissueLyser II (Qiagen) with one stainless steel bead (5 mm) per sample ($4 \times 30$ s at 25 Hz). Protein concentration was determined by DC Protein Assay (Bio-Rad) using BSA as a standard.

Cell lysates and tissue homogenates were denatured by incubation with Laemmli sample buffer (5.1% glycerol, 0.51 % SDS, 0.051% bromophenol blue, 30 mM DTT, 10.6 mM Tris-HCl, pH 6.9) for 5 min at 95 °C. Proteins were separated with Tris-glycine SDS-PAGE, transferred onto a nitrocellulose membrane, and detected with primary antibodies according to manufacturers' instructions, followed by incubation with secondary antibodies conjugated to fluorophores. Fluorescence was recorded with either the Odyssey Sa Imager (Li-Cor) or the Chemidoc Imager (Bio-Rad) and quantified using the Image Studio Lite (Li-Cor) or

the Image Lab software (Bio-Rad), respectively. The antibodies used in this study are listed in Supplementary Table S3.

## Detection of oxidized PRDX1 and PRDX3

For protein isolation, medium was removed from cells seeded in 6 cm dishes and cold thiol-block buffer (100 mM N-ethylmaleimide (NEM) in PBS) was added and incubated for 5 min on ice. Cells were lysed with 250 μL cold lysis buffer (1 % Triton X-100, 20 mM NEM in TBS (50 mM Tris, 150 mM NaCl, pH 7.4), complete protease inhibitor cocktail tablets (Serva, Cat No. 39101)) for 5 min on ice, sonicated and then centrifuged for 15 min at $12,000 \times g$ at 4 °C. From each sample 100 μg protein was mixed with 4x Laemmli buffer (277.8 mM Tris-HCl pH 6.8, 26.3 % (w/v) glycerol, 2.1% SDS, 0.01 % bromphenol blue (Na-salts), 40 mM NEM) for non-reducing condition and with 4x Laemmli buffer plus 20% (v/v) 1 M DTT for reducing condition. Samples were denatured for 5 min at 95 °C and separated on a denaturing gel (6% stacking, 15% separating). Proteins were transferred to a 0.2 μm nitrocellulose membrane using wet blot. Antibodies and respective dilutions are listed in Supplementary Table S3.

## DNA methylation analysis of cell lines

Genomic DNA extracted from 3 SCLC lines (H209, H146 and H69) and 3 NSCLC lines (H838, H1944 and A549) was subjected to methylation analysis using the Infinium MethylationEPIC BeadChips (Illumina) allowing the simultaneous quantitative measurement of the methylation status at 865,918 CpG sites. Approximately two million cells were harvested either by centrifugation at $1000 \times g$ for 5 min (suspension cultures) or by scraping from the dishes, followed by centrifugation at $1000 \times g$ for 5 min (adherent cells).

## DNA methylation analysis of patient samples

FlexiGene DNA Kit (Qiagen) was used for isolation of DNA according to the manufacturer's instructions. No technical replicates were performed. DNA concentrations were determined using PicoGreen staining (Molecular Probes). The quality of genomic DNA samples was checked by agarose-gel analysis, and samples with an average fragment size >3 kb were selected for methylation analysis. The laboratory work was done in the Genomics and Proteomics Core Facility at the German Cancer Research Center, Heidelberg, Germany (DKFZ). 500 ng of genomic DNA from each sample was bisulfite converted using the EZ-96 DNA Methylation Kit (Zymo Research) according to the manufacturer's recommendations. The DNA was applied to Infinium MethylationEPIC BeadChip and hybridization was performed for 16–24 h at 48 °C. Allele-specific primer annealing was followed by single-base extension using DNP- and Biotin-labeled ddNTPs. After extension, the array was fluorescently stained, scanned, and the intensities at each CpG were measured. Microarray scanning was done using an iScan array scanner (Illumina). DNA methylation values, described as beta values, were recorded for each locus in each sample. Data analysis was performed with the Illumina's GenomeStudio 2011.1 (Modul M Version 1.9.0). The complete raw Illumina data was quantile normalized.

## Methylation heatmaps based on CpG clusters

We plotted methylation values for CpG clusters (file CCLE_RRBS_TSS1kb_20181022.txt.gz from DepMap) within promoters of ACB gene set (and MAFG) using ComplexHeatmap R package for cell lines from DepMap project (CCLE) that have both promoter methylation and expression data (Fig. 4A). We calculated correlation between vector of cell lines's methylation values and vector of cell lines's average expression of genes from ACB set per each CpG cluster, and plotted these correlation values as heatmap's column annotation. As heatmap's row annotation we plotted average expression of genes from ACB gene set per cell line and sorted heatmap's rows according to this value (from highest avg. ACB expression to the lowest). We also provided an additional layer of row annotation – cell line's EC$_{50}$ values,

and additional layer of column annotation – correlation between vector of cell lines' methylation values and vector of cell lines' EC$_{50}$ per each CpG cluster.

## Immunohistochemistry (IHC)

The IHC staining in human samples was supported by the tissue bank of the National Center for Tumor Diseases (NCT, Heidelberg) in accordance with the ethical regulations of the NCT tissue bank established by the local ethics committee. IHC was performed by applying a rabbit monoclonal antibody against the TXN1 (1:2000; Abcam, ab133524) to tissue microarrays (TMA, two cores per case) of formalin fixed and paraffin embedded (FFPE) NSCLC and SCLC specimens. The staining procedure was carried out using an autostainer (Bench-Mark ULTRA, Ventana Medical Systems) according to the manufacturer's instructions.

Staining intensities were evaluated digitally using the QuPath software (v. 0.1.2)[62] and applying random tree algorithms for classification. After verification by a pathologist, the classifiers were used to differentiate between non-tumor cells (e.g., fibroblast, lymphocyte) and tumor cells and to count the latter subdivided by three staining intensities (weak, moderate, strong) in each TMA core. For a weighted analysis, we calculated H-scores, yielding from 0 to 300 as the sum of the percentage of weakly, moderately and strongly stained tumor cells multiplied by 1, 2 and 3, respectively. We averaged the H-scores over the two TMA cores of each case.

## RNA isolation and RT-qPCR

Total RNA from cells was isolated using NucleoSpin RNA Kit according to the manufacturer's protocol (Macherey-Nagel). To isolate RNA from mouse tissues stored in RNAlater (Sigma-Aldrich), samples were first homogenized in the TRIzol reagent (Invitrogen) using TissueLyser II (Qiagen) with one stainless steel bead (5 mm) per sample ($2 \times 2$ min at 25 Hz) and then the manufacturer's instructions were followed. Purified RNA was measured by Nanodrop (Thermo Fisher) and reverse transcription was performed using Revert Aid First Strand cDNA Synthesis Kit (Thermo Scientific). qPCR was performed on the Roche Lightcycler 480 system using Blue S'Green qPCR kit (Biozym). Primer sequences used for quantitative real-time PCR analyses are listed in Supplementary Table S4. *GAPDH* or *ACTB* was used for normalization of gene expression.

## ACB expression profile

Figure 1A: ACB expression was calculated as average per sample and are based on the RNAseq datasets of CCLE cell lines (left panel). For expression data normalization, values were transformed to log2(TPM + 1) units. After that average expression value for the ACB gene set were calculated for each sample (using log2(TPM + 1) expression values of individual genes).

Figure 1C: Expression data for normal tissues, lung adenocarcinoma (LUAD) and lung squamous cell carcinoma (LUSC) were downloaded from The Cancer Genome Atlas (TCGA) and converted to Log2(TPM + 1) values. SCLC expression values were obtained from Rudin et al.[11] and converted to Log2(TPM + 1) values. All expression values were then z-scored and normalized to normal lung tissue. To determine the ACB scores, the expression of ACB genes was averaged.

Expression profiles for pulmonary neuroendocrine cells (PNECs) were obtained from the Human Lung Cell Atlas through Synapse (https://www.synapse.org/#!Synapse:syn21041850)[63]. Smart-seq2 single-cell profiles from PNECs were aggregated by patient and processed as described above.

## Expression profiling (cell lines)

Total RNA isolated from Beas-2B, HaCaT, H82 and H526 cells using NucleoSpin RNA Kit according to the manufacturer's protocol (Macherey-Nagel) was submitted to transcriptome profiling on Human Clariom S Assay chips (Applied Biosystems) that allow for

quantification of >20,000 well-annotated genes. All the necessary quality controls and the analysis itself were performed by the Microarray Core Facility of the German Cancer Research Center.

## RNAseq (mouse tissues and tumors)

Total RNA extracted from mouse tissues and tumors was submitted for RNAseq analysis to the NGS Core Facility of the German Cancer Research Center. After RNA passed the quality control, sequencing libraries were prepared using TruSeq Stranded mRNA Library Prep Kit (Illumina) and IDT Unique Dual Indexes for Illumina (Integrated DNA Technologies) and sequenced using NovaSeq 6000 sequencing system (paired-end mode, read length 100 bp, S4 flow cell; Illumina), yielding on average 52 million reads per sample (32–76 million). For the analysis of the known reference transcriptome assembly, RNA sequencing data were processed automatically via the One Touch Pipeline (OTP)[64].

## Animal experiments

Animal Handling: All studies involving mice were conducted in compliance with German Cancer Research Center guidelines and approved by the governmental review board of the state of Baden-Württemberg, Karlsruhe District Council, under authorization no. G-191/16, G-259/18, and G-176/19, according to German legal regulations. Mice were housed in individually ventilated cages (IVCs) under controlled temperature ($22 \pm 2\,°C$) and humidity (50–60%) conditions on a 12-h light/dark cycle. They received a standard diet (Catalog No. 3307, Kliba Nafag) ad libitum, with cages enriched by bedding material. Animal health was monitored daily, and mice were euthanized immediately upon reaching predefined termination criteria. Sample size was calculated with the help of a biostatistician. Assumptions for the power analysis were as follows: Alpha error, 5%; Beta error, 20%. Mice were randomized into treatment groups before treatment. In case animals had to be euthanized before the predefined end point (due to weight loss or other termination criteria), they were excluded from any downstream analyses.

Determination of the Maximum Tolerated Dose (MTD): Female mice (6–7 weeks old) of the nude strain BALB/c (BALB/cAnNCrl, Charles River) were used for the dose escalation study and the repeated-dose efficacy study with xenografts. To determine the MTD of DKFZ-608 for intraperitoneal administration, three animals were injected daily, starting from 1 mg/kg, with doses of DKFZ-608 increasing every three days to 7.5, 10, 12.5, 15 and finally 20 mg/kg. Body weight was monitored daily and weight loss was observed only at a dose of 20 mg/kg. Therefore, we defined the MTD of DKFZ-608 as 15 mg/kg, which is the dose that was used for tumor treatment. The MTD of DKFZ-682 for intraperitoneal administration was determined in the same fashion, starting with 0.5 mg/kg. At 6 mg/kg, we observed a weight loss which could only be reversed when the dose was reduced to 4 mg/kg. Accordingly, we defined the MTD of DKFZ-682 as 4 mg/kg (Fig. S5C). Female mice (8–10 weeks old) of the immunodeficient strain NSG (NOD.Cg- Prkdc[scid] Il2rg[tm1Wjl]/SzJ; DKFZ Heidelberg) were used for the dose escalation study of DKFZ-682 in combination with CDDO-Me and the dose efficacy study with xenografts. To determine the MTD of DKFZ-682 for i.p. administration, three animals were injected daily, starting from 4 mg/kg, with doses of DKFZ-682 increasing every 1–3 days to 6, 8, 10, and finally 12 mg/kg. At the highest dose, the animals reached the human endpoint. Body weight was monitored daily and weight loss did not exceed 20% at a dose of 10 mg/kg. Therefore, we defined the MTD of DKFZ-682 upon CDDO-Me co-treatment as 10 mg/kg.

Generation of Subcutaneous SCLC-Tumor Models: To generate a subcutaneous SCLC-tumor model, $3 \times 10^7$ H209 cells in 100 μL of ice-cold PBS were injected subcutaneously into the flank. 6 weeks later tumor bearing animals (tumor size 4–8 mm) were selected and randomly distributed to experimental groups of 10 animals each. Tumor

size was determined twice a week in two dimension using calipers and tumor volumes were calculated by the formula (width$^2 \times$ length)/2. To generate a subcutaneous SCLC-tumor model, H526 cells were resuspended in a 1:1 (vol/vol) mix of ice-cold growth factor-reduced Matrigel (Corning) and PBS. Overall, 100 μl of this cell suspension containing $2 \times 10^6$ H526 cells was injected subcutaneously into the flank of anesthetized NSG mice. After detection of palpable tumors 2 weeks later, tumor bearing animals were randomly distributed to experimental groups.

Drug Treatments: The solution of DKFZ-608 for injections (2 mg/mL) was prepared in 200 mM ß-Cyclodextrin sulfobutyl ether/PBS. To completely dissolve the compound, the vial was first placed into a sonication bath (30 min, 37 °C) and then heated up in a 98 °C water bath for 30 min. Similarly, the solution of DKFZ-682 (4 mg/mL) was prepared[15]. Cisplatin and etoposide were dissolved in 60 mM ß-Cyclodextrin sulfobutyl ether/PBS at a concentration of 0.3 mg/mL and 0.75 mg/mL, respectively. CDDO-Me was first dissolved in DMSO to make a stock at a concentration of 20 mg/mL. This stock was then mixed with PEG300 (40%), Tween-80 (5%) and PBS (50%) to dilute it to 1 mg/mL. Tested substances were administered for 3 weeks (chemotherapy) or 40 days (DKFZ-608). Specifically, 0.2 ml of each drug solution was injected per 20 g mouse to obtain a dose of 15 mg/kg DKFZ-608 (starting with dose escalation of 5- > 15 mg/kg in the first 5 days, continued with 15 mg/kg up to day 40, daily i.p. injections), 3 mg/kg cisplatin (i.p. injection on Monday) and 7.5 mg/kg etoposide (i.p. injection on Wednesday and Friday).

DKFZ-682, in combination with CDDO-Me or solvent control were administered via i.p. injection daily for 3 weeks. CDDO-Me was used at a concentration of 3 mg/kg, DKFZ-682 at 4 mg/kg and 10 mg/kg alone and in combination with CDDO-Me, respectively. Tumor volumes were measured every 2–3 days using calipers and calculated as (width$^2 \times$ length)/2. Mice were sacrificed upon reaching the humane endpoint of 1500 mm$^3$.

Evaluation of organ damage markers: To evaluate organ damage markers, the animals were sacrificed by isoflurane overdose and blood was collected from the portal vein using 24G needles (BD Microlance 3) into microtubes with clot activator (Microvette 500 CAT-Gel, Saarstedt). After incubation for 30 min at room temperature, blood samples were centrifuged for 5 min at $10,000 \times g$ at room temperature and the upper cell-free layer consisting of serum was transferred into a new tube and stored at −80 °C until further analysis.

Pieces of tissues (liver, kidney, lung, heart, tumor) were collected for further protein (dry) and transcript (into RNAlater, Sigma-Aldrich) analysis and stored at −80 °C until further processing.

Kidney damage was quantified as the level of urea in serum using the Urea Nitrogen (BUN) Colorimetric Detection Kit (Invitrogen) according to the manufacturer's instructions, adding 4 μL of serum per well. As proxies for liver damage, the activities of alanine aminotransferase (ALT) and aspartate aminotransferase (AST) were measured in serum samples using Alanine Aminotransferase Activity Assay Kit (MAK052, Sigma-Aldrich) and Aspartate Aminotransferase Activity Assay Kit (MAK055, Sigma-Aldrich), respectively, according to the manufacturer's instructions and adding 3 μL and 5 μL of serum per well for ALT and AST activity, respectively.

## Statistics and reproducibility

Statistical analyses were performed in RStudio (Posit) or GraphPad Prism 9 software. No statistical method was used to predetermine sample size in cell culture experiments. The sample sizes for animal experiments were determined with the one-sided log-rank test ($\alpha = 0.05$, power = 80%) based on hazard ratios from similar experiments. No data were excluded from the analyses. In vitro treatments of cells and animal experiments were performed randomized. The Investigators were not blinded to allocation during experiments and outcome assessment. Data are presented as mean ± SD.

## Software

All figures were generated using Affinity Designer (Canvas). Schematic overview figures were created in BioRender (Nuskova, H. (2026) https://BioRender.com/xjxr0qn).

## Reporting summary

Further information on research design is available in the Nature Portfolio Reporting Summary linked to this article.

## Data availability

The RNA sequencing and microarray data generated in this study have been deposited in the NCBI Gene Expression Omnibus (GEO) database under the primary accession number GSE280643. The publicly available data used in this study are available in the European Genome-Phenome Archive under the primary accession number EGAS00001000334 and in refs. 65,66. The remaining data are available within the Article, Supplementary Information or Source Data file. Source data are provided with this paper.

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

## Acknowledgements

We thank the Helmholtz Validation Fund (grant number HVF-0069) and Else Kröner-Fresenius-Stiftung Translationsförderung (EKFS-Theropo) for supporting this work. M.M. acknowledges support from the Deutsche Forschungsgemeinschaft (FerrOs- FOR5146; GRK2727; SPP2306). H.A. acknowledges the support of DFG priority program SPP230. T.D. acknowledges support by funding from the DFG (SPP2306, TRR186). M.M. acknowledges support by funding from CellNetworks (EXC81), ERC StG No 804710, and the Hector Stiftung II GmbH. Fellowships were provided by the Helmholtz International Graduate School to B.L. We thank the Microarray Core Facility of the German Cancer Research Center (DKFZ) for providing excellent Expression Profiling services and the Illumina methylation arrays and related services. We thank the NGS Core Facility of the German Cancer Research Center (DKFZ) for providing excellent RNAseq services. We also thank Jonas Gross, Tobias Herrmann, Nawid Albinger, and Jonas Kolibius for the valuable work during their lab internship, which helped to build a robust understanding of the mode of action of DKFZ-608.

## Author contributions

J.S., H.N., P.F., M. Malz, E.A., M.J.T., D.P.-F., and D.K. have made substantial contributions to the acquisition, analysis, or interpretation of data. R.K. and J.F. have substantially contributed by bioinformatics analysis of data and interpretations. J.H.-E., N.d.V., L.R., H.P., F.D., K.K., B.L., T.K., and G.K. made substantial contributions to the acquisition of data. G.H. was instrumental for data obtained with circulating tumor cells. M. Mall and T.P.D. made substantial contributions to the interpretation of data. M.Mu. and M.S. have substantially contributed to drafting the work and substantively revised it. A.K.M., H.A., and N.G. have made substantial contributions to the conception and design of the work. A.K.M. was instrumental for the design and synthesis of TXNRD1 inhibitors. J.S., H.N., A.K.M., H.A., and N.G. wrote the paper, with input from all other authors. H.A. and N.G. jointly supervised this work.

## Funding

## Competing interests

The authors declare no competing interests.

## Additional information

[1]German Cancer Research Center (DKFZ) Heidelberg, Research Group Cancer Drug Development, Heidelberg, Germany. [2]German Cancer Research Center (DKFZ) Heidelberg, Division of Redox Regulation, DKFZ-ZMBH Alliance, Heidelberg, Germany. [3]Faculty of Biosciences, Heidelberg University, Heidelberg, Germany. [4]Institute of Pathology, Heidelberg University, Heidelberg, Germany. [5]Center for Lung Research (DZL), Heidelberg, Germany. [6]Division of Applied Bioinformatics, German Cancer Research Center (DKFZ) Heidelberg, Heidelberg, Germany. [7]Heidelberg Institute for Stem Cell Technology and Experimental Medicine, Heidelberg, Germany. [8]Institute of Pharmacology, Medical University of Vienna, Vienna, Austria. [9]Department of Pediatric Hematology, Oncology and Immunology, Heidelberg University, Heidelberg, Germany. [10]German Cancer Research Center (DKFZ) Heidelberg, Research Group Cell Fate Engineering and Disease Modeling, Heidelberg, Germany. [11]HITBR Hector Institute for Translational Brain Research GmbH, Heidelberg, Germany. [12]Central Institute of Mental Health, Medical Faculty Mannheim, Heidelberg University, Mannheim, Germany. [13]Metabolomics Core Technology Platform, Centre for Organismal Studies (COS), Heidelberg University, Heidelberg, Germany. [14]Department of Translational Oncology, German Cancer Research Center (DKFZ) Heidelberg, Heidelberg, Germany. [15]German Cancer Consortium (DKTK), partner site Munich, a partnership between DKFZ and Ludwig-Maximilians-University, Munich, Germany. [16]Department of Medicine III, LMU University Hospital, LMU Munich, Germany. [17]Department of Translational Genomics, Faculty of Medicine and University Hospital Cologne, University of Cologne, Cologne, Germany. [18]Technical University of Munich, School of Medicine and Health, Institute of Clinical Chemistry and Pathobiochemistry, TUM University Hospital, Munich, Germany. [19]German Cancer Consortium (DKTK), Heidelberg, Germany. [20]These authors contributed equally: Jana Samarin, Hana Nůsková. [21]These authors jointly supervised this work: Hamed Alborzinia and Nikolas Gunkel. ✉e-mail: n.gunkel@dkfz.de

