## [Transparent Peer Review file · Nature Communications]

Differential KEAP1/NRF2 mediated signaling widens the therapeutic window of redox-targeting drugs in SCLC therapy

Corresponding Author: Dr Nikolas Gunkel

Version 0:

Reviewer comments:

Reviewer #1

(Remarks to the Author)

The manuscript "Differential KEAP1/NRF2 mediated signaling widens the therapeutic window of redox-targeting drugs in SCLC therapy" by Jana Samarin et al. highlights the therapeutic strategy by targeting TXNRD1 inhibitors in SCLC treatment while protecting non-cancerous tissues via KEAP1/NRF2 pathway modulation. The use of TXNRD1-specific inhibitors in treating SCLC cancer is innovative, and the study is comprehensive. However, the following major points need to be addressed.

1. The authors utilized two novel TXNRD1 inhibitors, DKFZ-608 and DKFZ-681, across multiple SCLC and NSCLC cell lines and in vivo models. However, critical details regarding their efficacy are missing. Specifically: (1) Did the administered dosages effectively inhibit TXNRD1 activity in both cell lines and animal models? No data has been provided demonstrating TXNRD1 protein expression or activity following treatment with the two inhibitors. (2) The authors should also include information on the in vivo half-life of DKFZ-608 and DKFZ-681 to evaluate their pharmacokinetic properties.
2. The authors claim that enhanced NRF2 expression can widen the therapeutic range of a TXNRD1 inhibitor. However, in Figure 6B, G, and F, the data for 10 mg/kg DKFZ-682 treatment alone are missing. This data is critically needed to be compared with the 10 mg/kg DKFZ-682 + CDDO-Me treatment data to support the authors' conclusion.
3. The authors used the same concentration of CDDO-Me (50 nM) to treat both SCLC and non-cancerous cell lines in Figure 3F and Figure 3G. However, CDDO-Me is an electrophilic agent that induces NRF2 by irreversibly and covalently binding to cysteine residues of KEAP1, making its effectiveness critically dependent on KEAP1 protein expression levels. The following points should be addressed: (1) The authors did not provide data showing KEAP1 protein expression levels in these cell lines. Without this, it is unclear whether the chosen concentration of 50 nM is appropriate for all cell types tested. (2) Generalizing the use of 50 nM CDDO-Me across SCLC and non-cancerous cell lines undermines the conclusion that "SCLC cell lines failed to increase ROS-scavenging capacity to adapt to TXNRD1 inhibition" or that "SCLC cells remain largely sensitive despite NRF2 induction." (3) The authors failed to provide a mechanistic explanation for the lack of ACB induction by NRF2 in SCLC cells overall.
4. The manuscript lacks critical details regarding the experimental procedures, particularly in the animal studies:

Figure 2 Legend: The authors state that "tumor-bearing mice were distributed into groups (n=10) and treated with cisplatin/etoposide (chemo) for 3 weeks or DKFZ-608 for 40 days."

In Lines 203-204, where it is mentioned that the second group received 3 cycles of cisplatin (3 mg/kg, i.p.) and etoposide (7.5 mg/kg, i.p.).

For these in vivo studies,

- (1) How was the tumor model established?
- (2) How many cells were injected into the nude mice?
- (3) What was the dosing schedule and interval for the "3 cycles" of cisplatin and etoposide?

Similarly, Figure 6: The number of H526 cells injected to establish the tumor model is not specified. This information is essential for reproducibility and clarity. The authors need to provide detailed descriptions of the experimental procedures, including the tumor establishment procedures, cell numbers injected, and treatment protocols, to ensure transparency and

enable proper evaluation of the study.

5. In Figure 1A, the authors state that “the expression of 15 ACB genes for each SCLC and NSCLC cell line was calculated as average and based on the CCLE dataset and PDX, obtained from Champions Oncology (left panel). Units were converted to $\log_2(\text{TPM}+1)$.” However, the following points require clarification: Data Type: The authors need to specify the nature of the data obtained from Champions Oncology. Is this transcriptional data (RNA-seq) or protein expression data (proteomics data)? Averaging Method: How was the expression data “averaged” across the 15 ACB genes? Was this done per sample or across multiple samples? Normalization: What normalization method was used to compare the SCLC and NSCLC data? Was the $\log_2(\text{TPM}+1)$ transformation applied after averaging or at the individual sample level? As this figure presents a key dataset for the manuscript, providing detailed and transparent information regarding data type, processing, and analysis methods is crucial for understanding and reproducing the results.

6. In Figure 1B, the authors compare ACB protein expression between SCLC and NSCLC but do not include data from non-cancerous cell lines. This comparison is crucial to support the claim that TXNRD1 inhibition can widen the therapeutic window for SCLC while causing limited toxicity to normal lung tissue. The authors should include the ACB protein expression levels in non-cancerous cell lines alongside SCLC and NSCLC data.

7. In Lines 176-179, the authors state: “Our results demonstrate that SCLC cells are highly sensitive to TXNRD1 inhibitors and that cisplatin-resistant SCLC cells maintain this sensitivity, suggesting that TXNRD1 inhibition is a promising strategy for SCLC therapy.” Additionally, in Figure 2A, the in vivo experiments show a significant portion of animals experiencing remission with DKFZ-608 treatment.

Has the sensitivity of DKFZ-608-resistant SCLC cells to TXNRD1 inhibitors been tested? This is a critical question to evaluate whether resistance to DKFZ-608 develops over time and whether cross-resistance to other TXNRD1 inhibitors occurs.

8. The authors need to address inconsistencies and clarify data integrity:

Supplementary Figure 7A vs. Figure 3A: In Supplementary Figure 7A, SCLC cell line H82 shows higher NRF2 induction than the non-cancerous Beas-2B cell line. However, in Figure 3A, the summarized data show the opposite, with higher NRF2 induction in Beas-2B than in H82. Similarly, H526 demonstrates much higher NRF2 induction than H82 in Supplementary Figure 7A. However, in Figure 3A, the two cell lines appear to have similar NRF2 induction levels. Further, H526 (SCLC) shows higher NRF2 induction than the non-cancerous cell lines HaCaT and Beas-2B in Supplementary Figure 7A. Yet, Figure 3A summarizes NRF2 induction as significantly higher in non-cancerous cells than in H526. The authors should clarify these discrepancies and ensure data consistency between the figures.

Figure 3E: The authors did not specify the source of the data presented. Is this transcriptional (e.g., RNA-seq) or translational (e.g., protein expression) data? Without this clarification, it is difficult to interpret or validate the conclusions drawn from Figure 3E.

9. The data presented in Supplementary Figure 8C are incompletely interpreted. Both TXNRD1 and GSR are NADPH-dependent antioxidant enzymes that rely on NADPH for their activity. Consequently, when G6PD inhibition (G6PD-i) reduces NADPH generation, the protective effect of CDDO-Me is impaired.

10. The manuscript lacks an in-depth investigation into the mechanisms by which TXNRD1 inhibitors induce cell death and what types of cell death.

Reviewer #2

(Remarks to the Author)

In this study, Jana Samarin et al. demonstrate the potential of TXNRD1 inhibitors as a promising maintenance therapy for small cell lung cancer (SCLC). SCLC cells, which exhibit low reactive oxygen species (ROS) defense, are hypersensitive to TXNRD1 inhibition and fail to adapt to drug-induced ROS stress due to epigenetic and transcriptional suppression of the defense system. Through pharmacologically activating the NRF2 stress response pathway, the authors are able to safely increase the dose of TXNRD1 inhibitors, improving tumor control without additional toxicity to healthy tissues. These findings suggest that TXNRD1 inhibitors could effectively overcome resistance in relapsed SCLC cases, offering a novel therapeutic approach for controlling this challenging disease. The idea behind this work is very interesting and indeed brings certain conceptual advance. I have some minor concerns for the authors to further improve the manuscript.

1. The authors state that the antioxidant capacity biomarker (ACB) scores of pulmonary neuroendocrine cells (PNECs) are similarly low to those of SCLC. Given this, the authors might want to consider carefully about the potential side-effects of TXNRD1 inhibitors upon the nervous system in mouse models, or at least make some discussion.

2. DKFZ-608 seems have higher potency against SCLC and reduced selectivity towards non-cancerous cells. Given this, it remains unclear why the authors chose to use DKFZ-682 in subsequent experiments.

3. Other issues: statistical analyses for the mouse experiments should be included in Figures 2A, 2B, 2C and 5A. In Figure 4D, the scale bar should be consistent across all panels. In Figure 5B, if not significant, please indicate "ns".

Reviewer #3

(Remarks to the Author)

Tumor relapse poses a significant challenge in the treatment of SCLC patients. In this study, the authors discovered that SCLC cells exhibit heightened sensitivity to TXNRD1 inhibition due to their impaired ROS defense mechanisms. Notably, even SCLC cells resistant to first-line therapies remain susceptible to TXNRD1 inhibition. This suggests that targeting TXNRD1, in combination with standard first-line treatments, could represent a promising therapeutic strategy. Furthermore, the authors observed that high doses of TXNRD1 inhibitors may induce host toxicity. Activation of the NRF2 pathway was found to protect host cells from the toxic effects of TXNRD1 inhibitors without conferring protection to SCLC cells. Consequently, co-administration of an NRF2 activator could potentially widen the therapeutic window for TXNRD1 inhibitors in the treatment of SCLC. Overall, this study is well-conducted, and the findings are both intriguing and of substantial scientific value. This reviewer only has one major point and a few minor suggestions to offer.

Major point:

1. Figure 2: The anti-tumor effect of DKFZ-608 appears highly promising. However, to substantiate the claim that DKFZ-608 effectively eradicates first-line therapy-resistant SCLC cells, it is crucial to demonstrate that these cells are indeed resistant to first-line treatment. In the current study, the authors did not administer any treatment to the "Chemo" group mice after tumor relapse, leaving it unclear whether the relapsed tumors were resistant to first-line therapy. To support this claim, the authors should include two additional experimental groups. Both groups would initially receive three cycles of first-line treatment and be monitored until tumor relapse. Upon relapse, one group would be treated with the first-line therapy again, while the other would receive DKFZ-608. If DKFZ-608 successfully eliminates the tumors while the first-line treatment fails, the conclusion that DKFZ-608 targets resistant SCLC cells would be well-supported.

Minor points:

1. Figure S1A: It would be valuable to include the expression levels of TXNRD1 and GPX4 in SCLC cell lines.
2. Figure S2A: RSL3 may exhibit off-target effects on other selenoproteins, including TXNRD1, particularly at higher concentrations. To ensure specificity, the authors should consider using ML210, a more selective GPX4 inhibitor. Additionally, including a ferroptosis inhibitor (e.g., ferrostatin-1 or liproxstatin-1) as a control would strengthen the study. If the observed effects are indeed mediated by GPX4 inhibition, cell viability should be fully restored by the ferroptosis inhibitor.
3. Figure S4D: The authors could use N-acetyl-L-cysteine, an antioxidant, to assess whether it rescues the observed cell death, which would further clarify the role of oxidative stress in the mechanism.
4. Figure S8D: The unit of the y-axis should be indicated.

Version 1:

Reviewer comments:

Reviewer #1

(Remarks to the Author)

Some of the concerns have been addressed, but several important points remain unresolved.

1. While the authors tested plausible mechanisms (e.g., promoter methylation, MAFG/FRA1 levels, STAT3 repression), no definitive explanation is provided for the failure of NRF2 to activate antioxidant capacity biomarkers (ACBs) in SCLC. This mechanistic gap limits the biological insight of the study.
2. Using a single dose of 50 nM CDDO-Me across cell lines may not fully capture NRF2 inducibility in SCLC. A dose-response analysis of NRF2 target gene expression or activity across cell types would strengthen conclusions about limited responsiveness in SCLC.
3. The NRF2 protein induction data in Supplementary Figure 7A appear inconsistent with Figure 3A still. These discrepancies should be reconciled clearly, either by standardizing quantification methods or providing an explanation for inter-assay variability.
4. While liver TXNRD1 activity is shown, it is unclear whether DKFZ treatment leads to functional toxicity in normal tissues. Including histology, liver enzyme markers, or animal weight trajectories would help clarify on-target effects in non-cancerous cells.
5. A direct comparison to known TXNRD1 inhibitors like auranofin would contextualize the relative potency, selectivity, and toxicity profile of DKFZ-608/682.
6. A critical point that requires clarification is the role of KEAP1/NRF2 mutations in NSCLC. Approximately one-third of NSCLC cases harbor mutations in the KEAP1/NRF2 pathway. Therefore, it remains unclear whether the differential responses observed between SCLC and NSCLC are due to intrinsic lineage differences or simply reflect the absence of KEAP1/NRF2 mutations in SCLC cells. Moreover, the authors use H1944—an NSCLC cell line with a KEAP1 mutation—for comparison in several assays. This may not accurately represent the majority (~70%) of NSCLC cases with wild-type KEAP1, and could confound the interpretation of NRF2 activity and drug response. This concern is clearly illustrated in Figure 1B.
7. The authors suggest that promoter hypermethylation contributes to the downregulation of a subset of ACB genes and the NRF2 cofactor MAFG, as shown in Figure 4. This correlation is observed in both cell lines and patient tumor DNA.

However, in Figure S11A, the authors perform DAC and AZA pre-treatment but do not observe effective drug resistance through NRF2 induction. This raises concerns about the robustness of the proposed mechanism.

Overall, the manuscript contains a large volume of data, but the mechanistic insights are limited. In particular, the comparison between SCLC cell lines and a single KEAP1-mutant NSCLC line H1944 (which has constitutive NRF2 activation) is not appropriate and undermines the conclusions.

Additionally, while the authors describe a novel TXNRD1 inhibitor developed by their group, there is insufficient in vivo validation of its efficacy or comparison with the current TXNRD inhibitor, auranofin. The manuscript lacks adequate in vivo data to support the therapeutic potential of this compound.

Reviewer #2

(Remarks to the Author)

The authors have significantly improved the manuscript.

Reviewer #3

(Remarks to the Author)

My previous concerns have been well addressed. I only have two minor suggestions before the manuscript can be accepted for publication:

1. The data presented in Figure S3B support the conclusion that the relapsed tumors are resistant to cisplatin. However, the manuscript currently lacks a clear textual connection between these data and Figure 2. The authors are advised to revise the text to better integrate these findings.
2. Line 205: The manuscript jumps from Figure S4 directly to Figure S13. Supplemental figures should be referenced in sequential order.

Reviewer #4

(Remarks to the Author)

I share the view of the three initial reviewers: this is an interesting and valuable manuscript. It shows that thioredoxin reductase inhibitors can act against small-cell lung cancer and their effect can be augmented by drugs that boost NRF2 signalling. The idea is that SCLC has a low antioxidant response and inhibitors of thioredoxin reductase exploit the dependency on this antioxidant system by cancer cells. The animal data demonstrate the viability and efficacy of the combined NRF2/TR approach that allows the usage of higher dosages of anti-thioredoxin reductase drugs delaying cancer relapse and minimizing resistance. Let's hope that this approach can move further into the clinic. The manuscript also discusses some experiments conducted to explore hypotheses about the molecular and cellular mechanisms underlying the sensitivity of SCLC to this inhibition. Though not conclusive, the data provide some hints that remain open for further investigations.

The previous reviewers gave a number of comments. The revised manuscript carefully addresses all of them. I offer a few suggestions that would require only minimal changes in the manuscript with the only aim of improving clarity.

- 1) I would move the supplement section 1 to the main text as it discusses interesting mechanistic data.
- 2) Honestly, I have been struggling with Figure 4A. Can the authors revise the legend and improve clarity with a more extensive explanation of the tree at the top of the figure, the meaning of the two top lines and how the general reader can grasp the correlation between methylation levels and drug sensitivity.
- 3) The data and associated text shown in Figures R3 and R4 in the rebuttal might be incorporated into the SI. They will be useful also to guide future studies as they rule out certain mechanistic hypotheses.

Version 2:

Reviewer comments:

Reviewer #1

(Remarks to the Author)

1. Figure 1B:

The data presented in Figure 1B suggest that the observed difference in ACB protein expression is not attributable to SCLC per se. Rather, it appears that a subpopulation of NSCLC cell lines harboring KEAP1 mutations displays higher ACB expression. SCLC and KEAP1/NRF2 WT NSCLC exhibit comparable levels of ACB protein, indicating that there is nothing inherently distinct about SCLC in this regard. This conclusion is well supported by the rest of the data of the paper.

Therefore, it would be more accurate to state that the difference arises from the elevated expression in the KEAP1-mutant NSCLC subset, rather than a generalized downregulation in SCLC.

Additionally, the lack of a significant difference between non-cancer cell lines, SCLC, and KEAP1/NRF2 WT NSCLC further supports this interpretation. The claim regarding lower ACB in SCLC should therefore be moderated to avoid overstatement.

2. TXNRD1 inhibitor experiments:

While the authors have clearly invested substantial effort into the TXNRD1 inhibitor studies, it is important to note that similar

findings have previously been published by the same group. Given this context, descriptive observations alone may not provide sufficient novelty or mechanistic insight. Strengthening this section with more direct mechanistic data or functional validation would enhance its impact.

3. Use of H1944 cell line:

Since H1944 carries a KEAP1 mutation, it may not be an appropriate reference for comparison in this context, especially for the KEAP1 Western blot in Figure R1. Including a KEAP1 wild-type reference cell line would help clarify the interpretation of these results.

4. Figure R2.1:

- (A–B) The rationale for restricting CDDO-Me treatment to 6 hours is not clearly explained. Extending or justifying this time course would improve the rigor of the experiment.
- (B) The choice to remove CDDO-Me prior to DKFZ-682 treatment rather than performing co-treatment warrants clarification. A co-treatment strategy could better reflect potential interactions between these compounds.
- Including parallel experiments with non-cancerous cell lines would further strengthen the claim that “NRF2 induction does not enhance the ROS-scavenging capacity of SCLC.”

Reviewer #3

(Remarks to the Author)

My concerns have been fully addressed. The manuscript is now ready for acceptance.

Reviewer #4

(Remarks to the Author)

The manuscript has been further improved by addressing all comments made by the reviewer. It is going to be an interesting study with clinical implications.

Dear editor

We herewith address all points raised by the reviewers.

Reviewer #1 (Remarks to the Author): Expert in NRF2, KEAP1, and redox biology

The manuscript "Differential KEAP1/NRF2 mediated signalling widens the therapeutic window of redox-targeting drugs in SCLC therapy" by Jana Samarin et al. highlights the therapeutic strategy by targeting TXNRD1 inhibitors in SCLC treatment while protecting non-cancerous tissues via KEAP1/NRF2 pathway modulation. The use of TXNRD1-specific inhibitors in treating SCLC cancer is innovative, and the study is comprehensive. However, the following major points need to be addressed.

1. The authors utilized two novel TXNRD1 inhibitors, DKFZ-608 and DKFZ-681, across multiple SCLC and NSCLC cell lines and in vivo models. However, critical details regarding their efficacy are missing.

Specifically: (1) Did the administered dosages effectively inhibit TXNRD1 activity in both cell lines and animal models? No data has been provided demonstrating TXNRD1 protein expression or activity following treatment with the two inhibitors.

We want to thank the reviewer for the positive assessment of our study and appreciate the constructive points raised.

@Point1: We fully agree with the reviewer in that the demonstration of target engagement of an inhibitor is an important aspect for the assessment of the efficacy of a treatment concept. We had previously demonstrated in a NSCLC-cell line [1] that the cytotoxicity induced by DKFZ-682 directly correlates with cellular enzyme inhibition, as reflected by residual TXNRD1 activity, and the oxidative induction of disulfides. In the current manuscript we compared resistant A549 (NSCLC) with highly sensitive H209 (SCLC) cells and demonstrate a strong correlation between cell viability and residual TXNRD1 activity, upon treatment with DKFZ-608 (**Figure S1D**). We noticed that the presentation of these data was suboptimal and therefore added now a second Y-axis to the graph, which indicates the residual activity of TXNRD1, as quantified by the activity probe TRFS-green.

Revised Figure S1D:

(D) To quantify intracellular TXNRD1 inhibition, cell lines were treated with TRFS, immediately followed by a dilution series of DKFZ-608. Fluorescence induction was measured for 8 h and dose response effects were analyzed at time point 100 min. All data were normalized (untreated control was set to 100%). Data are presented as mean \pm SD of three technical replicates.

Inspired by the reviewer's comment, we also quantified TXNRD1-activity and protein levels in both tumors and normal tissue (represented by liver) from mice treated with DKFZ-682 with and without co-administration of the NRF2-inducer CDDO-Me. This set of data (now added as new **Fig. 6C, D**) is particularly informative, as it provides mechanistic insight at the level of target engagement into why SCLC cells do not benefit from NRF2 induction, whereas normal tissue exhibits increased resilience to drug-induced oxidative stress.

As requested by the reviewer, we show that the maximum tolerated dose of DKFZ-682 in unprotected mice (4 mg/kg) reduces the residual activity of TXNRD1 in tumors by 3-fold (**Fig. 6C**, left graph). Considering that TXNRD1 is a central node in redox homeostasis and that SCLC cells have a low intrinsic activity of ROS-scavenging enzymes and, most importantly, that those cells are unable to react to drug-induced ROS stress by upregulating their ROS-buffer capacity, it is likely that the observed reduction of TXNRD1 activity inhibits tumor growth. Treatment with DKFZ-682 did not affect TXNRD1 protein levels, arguing that the observed reduction of TXNRD1 activity is caused by enzyme inhibition (**Fig. 6C**, right graph). Liver tissue, resected from DKFZ-682 treated mice, also demonstrated a reduction of TXNRD1 activity relative to vehicle, but the effects were much smaller and non-significant (**Fig. 6D**, left graph). The important difference to tumor tissue is that treatment with the NRF2-inducer CDDO-Me caused a 4-fold increase in TXNRD1 activity, mirrored by a about 3-fold increase in TXNRD1 protein levels. Treatment with 4 mg/kg DKFZ-682 reduced this boost of TXNRD1 activity slightly, while 10 mg/kg caused an about 2-fold reduction, compared to the CDDO-Me treated mice. Importantly however, the residual activity of TXNRD1 was still more than 2-fold higher than observed in unprotected mice, consistent with the observation that CDDO-Me protects mice from drug-induced ROS stress.

Added **Figure 6C, D**: Impact of CDDO-Me and DKFZ-682 treatment on TXNRD1 activity and protein levels in tumor and liver tissue derived from treated mice.

The residual enzymatic activity of TXNRD1 in resected tumor and liver tissue was assessed by the activity probe TRFS green. Significance of differences was calculated with a two-tailed unpaired t-test (* $p < 0.05$, ** $p < 0.01$, *** $p < 0.001$, **** $p < 0.0001$). Non-significant differences between treatment groups are not specifically marked.

Accordingly, we incorporated the new data as Fig. 6C, D, updated the corresponding figure legend revised the results-section (from line 357 onward): The differential impact of CDDO-Me on tumor versus normal cells is reflected in intracellular TXNRD1 activity, which serves as an indicator of DKFZ-682 target engagement. In tumor cells, DKFZ-682 treatment reduced TXNRD1 activity by 3-fold, whereas CDDO-Me caused only a slight but nonsignificant increase, compared to tumors resected from untreated animals. This aligns with the largely unresponsive nature of the NRF2 regulon in SCLC cells. As a result, CDDO-Me-treatment failed to compensate the loss of TXNRD1 activity caused by DKFZ-682 treatment, leaving tumor cells susceptible to oxidative stress. Notably, these changes in TXNRD1 activity occurred independently of protein levels, which remained largely unaffected by drug treatments (Fig. 6C). In contrast, in normal tissue, exemplified by liver, CDDO-Me treatment led to a nearly 3-fold increase in TXNRD1 protein levels, corresponding to a more than four-fold increase in TXNRD1 activity. DKFZ-682 treatment only partially reverted this gain of target activity, allowing for a net-increase in ROS buffer capacity by TXNRD1 (Fig. 6D).

In line with the above-mentioned observations, we recapitulated these *in vivo* findings in a panel of 2 non-cancerous and 3 SCLC cell lines, which we had used throughout the project for hypothesis building. The results are placed in a new Fig. S8G.

Fig. S8 (G) TXNRD1 activity was measured using the fluorescent dye TRFS-Green as a substrate in cells pre-incubated with CDDO-Me or DMSO as a control for 24 h. After re-seeding immediately before the measurement, DKFZ-682 was added to the final concentration of 0.2 μM or 20 μM , the latter serving to determine TXNRD1 unrelated background in each cell line. TXNRD1 activity was calculated as Δ fluorescence of TRFS-Green per s per μg (the protein concentration was determined using BCA assay in the cell suspension used for re-seeding). The activities in each experiment were normalized to the average activity of HaCaT and Beas-2B ("batch-normalization"). All cell lines were measured in 3–5 independent experiments. Statistical analysis was performed using two-way ANOVA to evaluate main effects and interactions between cell lines and experimental treatments. Post-hoc pairwise comparisons were conducted using Tukey's test, with adjusted p-values reported for statistically significant intra-cell line comparisons (**** adjusted $p < 0.0001$). Non-significant differences are specifically marked only for relevant treatment groups.

The interpretation of these results is now referred to in line 277ff: TXNRD1, the DKFZ-682 target, also plays a role in CDDO-Me-mediated cytoprotection. In Beas-2B and HaCaT cell lines, CDDO-Me treatment induces a marked elevation in TXNRD1 enzymatic activity. This upregulation allows CDDO-Me to effectively counteract the suppression of TXNRD1 activity caused by DKFZ-682 when both compounds are administered together. Notably, this compensatory mechanism is absent SCLC derived cells, where CDDO-Me fails to stimulate TXNRD1 activity (Fig. S8G).

(2) The authors should also include information on the *in vivo* half-life of DKFZ-608 and DKFZ-681 to evaluate their pharmacokinetic properties.

@ Point 2: We fully agree with reviewer's request to show data on the pharmacological properties of DKFZ-608 and DKFZ-682. We had produced these data to prepare for the animal studies but had decided to omit them to reduce the length and complexity of the manuscript. We now provide the data in a new **Fig. S13**. We added data used to define the maximum tolerated dose (MTD) for *i.p.* application of DKFZ-608 and DKFZ-682 (**Fig. S13A**). The MTD levels of DKFZ-608 are 15 mg/kg and for DKFZ-682 4 mg/kg. At the beginning of dose confirmation (**Fig. S13B**) we noticed that animals had to be adapted to the MTD by a short dose escalation.

In order to assess the effective dosing of both drugs, we determined the time required to completely reduce the viability of H209 cells, grown in large spheroids. To this end we incubated cells with 1 μ M DKFZ-608 or DKFZ-682 for increasing time intervals, starting at 30 min, after which we discarded the drug containing medium, washed cells and incubated with drug free medium up to timepoint 1080 min (18 h), for subsequent determination of residual metabolic activity (assessed by Cell Titer Blue). We found that 1 μ M DKFZ-682 required 60 min and DKFZ-608 300 min of exposure to fully reduce viability of H209 spheroids (**Fig. S13C**). In addition, we quantified the plasma concentrations of both compounds after *i.p.* administration of MTD levels by ICP-MS and found that the plasma levels of DKFZ-608 and DKFZ-682 were sufficient to sustain effective drug exposure for targeting residual cancer cells after first-line therapy (**Fig. S13D**).

Figure S13: (A) Effect of dose escalation of DKFZ-608 and DKFZ-682 on the weight gain of *i.p.* injected mice. The MTD of DKFZ-608 was defined as 15 mg/kg and of DKFZ-682 as 4 mg/kg. (B) Treatment of mice with DKFZ-608 and 682 for 34 consecutive days (For details see method section). (C) Quantification of cell killing in a time course using 1 μ M DKFZ-608 or DKFZ-682. Solid red and green line: H209 cells were cultured at high density (200,000 cells per well in V-bottom 96 well plates) and incubated with 1 μ M DKFZ-608 or DKFZ-682. At the indicated time, drug containing medium was discarded, cells were washed and incubated with drug free medium up to timepoint 1080 min (18 h).

Viability (metabolic activity, assessed by CTB) is calculated relative to control cells which have undergone the same procedure with drug-free medium. **(D)** Quantification of plasma levels of gold after i.p. application (left y-axis) and calculated drug concentration (right y-axis). Drugs were applied at the MTD of each compound (DKFZ-608 15 mg/kg, DKFZ-682 4 mg/kg). Dotted line indicates 1 μ M drug levels at which DKFZ-608 requires 300 min and DKFZ-682 requires 60 min to eradicate H209 spheroids.

2. *The authors claim that enhanced NRF2 expression can widen the therapeutic range of a TXNRD1 inhibitor. However, in Figure 6B, G, and F, the data for 10 mg/kg DKFZ-682 treatment alone are missing. This data is critically needed to be compared with the 10 mg/kg DKFZ-682 + CDDO-Me treatment data to support the authors' conclusion.*

The reviewer has raised an important point, which we appreciate and would like to clarify further. The aim of the mouse study presented in figure 6 was to demonstrate that the increased in the maximum tolerated dose (MTD) of DKFZ-682, enabled by pre- and co-treatment with CDDO-Me, translates into improved anti-tumor efficacy. In other words, we compared the efficacy of two MTDs: 4 mg/kg without CDDO-Me protection versus 10 mg/kg with CDDO-Me protection. A direct comparison of 10 mg/kg DKFZ-682, with versus without CDDO-Me protection, as suggested by the reviewer is unfortunately not feasible, as CDDO-ME unprotected mice do not tolerate 10 mg/kg dose. Therefore, our conclusion that CDDO-Me widens the therapeutic window of TXNRD1 inhibitors is based on the observation that co-treatment with CDDO-Me allows for higher drug dosing, which in turn results in improved tumor control.

In order to make these circumstances more transparent to the reader we have modified the text in line 345 and added data describing the determination of MTD levels: *“It is important to note that treatment of unprotected mice with 10 mg/kg DKFZ-682 was not tolerable and therefore ineligible to study anti-tumor effects (Fig. S13A)”*.

We would like to take the opportunity to emphasize that the design of this mouse experiment (**Figure 6**) did not aim to reiterate the potential of TXNRD1 inhibitors in maintenance therapy but rather towards demonstrating the opportunity to protect normal tissue without compromising on anti-tumor efficacy. Therefore, we chose components which highlighted the problem of a narrow therapeutic window. First, we chose the least selective TXNRD1-inhibitor in our compound collection, DKFZ-682, in order to “provoke” dose-limiting side effects, which we could manage with NRF2 inducing drugs. Second, we selected an SCLC cell line (H526) which was, at least in tissue culture experiments, less responsive to inhibitors like DKFZ-608, DKFZ-682 or auranofin. This allowed us to use 4 mg/kg of DKFZ-682 (MTD in unprotected animals) without the “risk” of eradicating the tumors. To the same end we chose a monotherapy setup, rather than the maintenance-therapy design (as used in **Fig. 2**), as maintenance therapy with 4 mg/kg is likely to fully eradicate residual tumor cells after first line therapy and therefore would not allow us to demonstrate improved efficacy at 10 mg/kg + CDDO-Me. We have rewritten the intro to the results chapter accordingly (line 322-333): *... To test this hypothesis in vivo, we chose an experimental design which highlighted aspects of a narrow therapeutic window. First, we utilized the hydroxy-substituted derivative DKFZ-682. While this compound demonstrates a fast cytotoxic effect and reaches plasma concentrations sufficient for tumor control (Fig. S13C, D), it is less potent than DKFZ-608 against SCLC cells and shows reduced selectivity toward non-cancerous cells (Fig. S2D). In addition, its lower maximum tolerated dose (MTD) of 4 mg/kg (Fig. S13A, B) may limit its therapeutic efficacy. Second, in order to monitor drug efficacy more precisely, we administered DKFZ-682 as monotherapy rather than maintenance therapy, allowing us to eliminate confounding effects from first-line therapy. Lastly, we chose a tumor model based on H526, which exhibits partial resistance to TXNRD1 inhibition (Fig. S2D), to ensure that tumor growth would not be fully suppressed at the 4 mg/kg MTD in unprotected mice.*

3. The authors used the same concentration of CDDO-Me (50 nM) to treat both SCLC and non-cancerous cell lines in Figure 3F and Figure 3G. However, CDDO-Me is an electrophilic agent that induces NRF2 by irreversibly and covalently binding to cysteine residues of KEAP1, making its effectiveness critically dependent on KEAP1 protein expression levels. The following points should be addressed:

(1) The authors did not provide data showing KEAP1 protein expression levels in these cell lines. Without this, it is unclear whether the chosen concentration of 50 nM is appropriate for all cell types tested.

We thank the reviewer for raising this important issue. We fully align with the comment that the efficacy of a covalent binder or inhibitor can be sensitive to the expression level of its target. In fact, this is not restricted to covalent binders, but can apply to reversible binders when the K_D and dose of the inhibitor is in a concentration range similar to that of the target itself. In the case of CDDO-Me, the mode of target engagement to Cys151 in the BTB domain of KEAP1 appears to be covalent but reversible in biochemical assays [2]. However, to our knowledge it remains unclear whether this mechanism is maintained *in vivo* and consequently, to what extent KEAP1 protein expression levels impact the capacity of CDDO-Me to fully inhibit KEAP1 and stabilize newly synthesized NRF2. Using public proteomics data (DEPMAP) we estimated that the expression levels of KEAP1 protein were quite homogeneous (avg. 2.2, SD 0.5), a finding which seems to be confirmed by our own analysis in a panel of lung cancer and non-cancerous cells (**Figure R1**).

Figure R1: Protein levels of KEAP1 in total cell extracts from 13 SCLC and non-cancerous cell lines (Beas-2B, HaCaT) were analysed by immunoblotting (representative image, qualitatively similar to 2 biological replicates). The NSCLC cell line H1944 was used as reference.

This dataset is not currently included in the manuscript but can be provided upon request.

The protein levels of KEAP1 in the non-cancerous cells Beas-2B and HaCaT appear to be slightly higher as compared to SCLC cells. If one assumes a covalent binding mode of the KEAP1 inhibitor, it would require a slightly higher concentration of CDDO-Me in non-cancerous cells, to avoid under-dosing in comparison to SCLC's. As shown in **Fig. 3**, 50 nM CDDO-Me induces the NRF2 regulon and ROS-scavenging capacity more effectively in the non-cancerous cells which express higher KEAP1 levels, as compared to the lower expressing SCLC cells, suggesting that 50 nM CDDO-Me is unlikely to be underdosed in non-cancerous cells.

An additional factor we had to consider when selecting the concentration of CDDO-Me was its inherent toxicity, which constrains its protective capacity against drug-induced ROS stress. To establish the upper limit of a safe CDDO-Me dose, we evaluated its cytotoxicity in four representative SCLC cell lines and two non-cancerous cell lines. Consistently across all tested lines, 50 nM was identified as the maximum non-toxic concentration. We have added this dataset as Figure S7D and modified the text in line 256:Among SCLC cell lines treated with the highest non-toxic dose of 50 nM CDDO-Me (Fig. S7D).

Figure S7D: Cytotoxicity of CDDO-Me. Cells were treated with the indicated concentrations of CDDO-Me for 24 or 72 h. Cell viability was assessed using CellTiter-Glo assay. Graphs on the left panel are representative of one experiment, each performed in triplicate. Quantitative results from independent experiments ($n=2$, mean \pm SD) are summarized in the bar diagrams in the right panel.

(2) Generalizing the use of 50 nM CDDO-Me across SCLC and non-cancerous cell lines undermines the conclusion that “SCLC cell lines failed to increase ROS-scavenging capacity to adapt to TXNRD1 inhibition” or that “SCLC cells remain largely sensitive despite NRF2 induction.”

We acknowledge the reviewers concern that applying a single concentration of CDDO-Me across multiple cell lines bears the risk that we had missed a concentration (e.g., above 50 nM) which causes a strong induction of the NRF2 regulon in SCLC cells and thereby induces resistance to our TXNRD1 inhibitors. As noted in our response to the previous comment, we carefully selected a concentration that would maximize NRF2 stabilization most efficiently without being toxic to the cell, as overdosing would mask a potential cyto-protective effect. In the case of SCLC, we chose cell lines H82 and H526 which demonstrate a particularly high induction of NRF2 protein levels (**Fig. 3A**) and a robust induction of NRF2-target genes upon treatment with 50 nM CDDO-Me (**Fig. 3B**). Nevertheless, they fail to enhance their ROS-scavenging capacity upon CDDO-Me treatment, which is reflected by pathway analysis (**Fig. 3E**) and quantification of the ROS induced dimerization of PRDX1 (**Fig. 3F**).

(3) The authors failed to provide a mechanistic explanation for the lack of ACB induction by NRF2 in SCLC cells overall.

We thank the reviewer for this comment and appreciate the opportunity to clarify the conclusions drawn from the data presented in the manuscript. Additionally, we would like to introduce new data that contribute to our mechanistic understanding.

After we discovered the unexpected lack of resistance induction by CDDO-Me (and other KEAP1 inhibitors) in SCLC, we initially established that resistance induction in non-cancerous cells depends on NRF2 and MAFG (**Fig. S10A**). Despite our efforts to determine why this mechanism does not function similarly in SCLC, we have not yet identified a definitive explanation. This reflects the broader challenge of mechanistically explaining the absence of a response, particularly when multiple layers of regulation may be involved.

One contributing mechanism could be promoter hyper-methylation which strongly correlates with the expression of a subset of ACB genes and the NRF2-cofactor MAFG, both in cell lines and patient tumor DNA (**Fig. 4**). This finding is highly plausible to hypothesize that hypermethylation of NRF2 target gene promoters serves as a barrier to NRF2-mediated gene activation. To validate the causal relation of

promoter methylation and CDDO-responsiveness, we reduced DNA-methylation by decitabine (DAC) or 5-aza-2'-deoxycytidine (AZA) pre-treatment, aiming to allow resistance induction by CDDO-Me. In the manuscript we stated in lines 309 ff *“Contrary to our expectation, DNA-demethylation did not lead to more efficient drug resistance by NRF2 induction (Fig. S11A), in line with inconsistent induction of hypermethylated ACB-promoters (Fig. S11B). This indicates that, despite the high correlation of promoter methylation with ACB expression and drug sensitivity, epigenetic silencing is not a primary course for the lack of ACB induction by NRF2 and the inability of SCLC to adapt to drug-induced ROS stress.”* This statement quite possibly undersells the importance of DNA methylation in SCLCs inability to activate ROS defence. For once, our attempt to demonstrate causality was compromised by the fact that SCLC cells are particularly sensitive to DAC or AZA treatment, which makes it likely that we had to under-dose the drugs to avoid cytotoxicity, which would mask resistance induction by the combined AZA/CDDO or DAC/CDDO treatment. A second reason, why DAC or AZA pre-treatment did not increase CDDO responsiveness might be due to the fact that not all ACB's promoters are subject to increased DNA methylation, suggesting that their expression would not be enhanced by demethylation, resulting in an insufficient boosting of ROS buffer capacity.

We also tested whether the lack of a positive co-factor like MAFG, which plays an important role in non-cancerous cells, could account for a lack of CDDO-responsiveness. Although this option appeared to be well supported by literature, we found that MAFG was not the limiting factor for CDDO-response in SCLC (Fig. S10). The same applies to BACH1, which we had suspected to block NRF2 activation of ACB genes (Fig. S9).

Since the initial submission of the manuscript, we further investigated 2 additional mechanisms which are suggested by the literature: The first, FRA1 (*FOSL1*), a transcription factor uniformly under-expressed in SCLC and linked to reduced expression of certain NRF2 response genes ([3], see discussion line 467) could be a limiting factor for CDDO-Me-mediated resistance induction. DEPMAP data confirmed statements of this earlier study (Figure R2).

Figure R2: Expression levels of AP1 transcription factor subunits in SCLC and NSCLC (LUAD) cell lines. The panel of 58 CCLE cell lines derived from SCLC isolates express significantly lower levels of FRA1 (*FOSL1*) as compared to a NSCLC panel (140 cells). mRNA expression levels are derived from DEPMAP ($\log_2(\text{TPM}+1)$).

On this basis, we armed HCC33 and H69 cells with a DOX-inducible FRA1 expression plasmid and tested whether FRA1 expression, in combination with CDDO-Me treatment, induced drug resistance to DKFZ-682 (see Figures below). As this was not the case, we pre-treated cells with DAC or AZA, as we reasoned that epigenetic silencing of ACB promoters might prevent transcriptional activation of redox pathways by FRA1/NRF2. The combined pre-treatment with CDDO-Me (NRF2-induction), DOX (FRA1-induction), and DAC and AZA (demethylation of ACB-promoters) did, again, not induce resistance to DKFZ-682, arguing that the lack of FRA1 in SCLC cells was not responsible for the low ACB status and non-responsiveness to NRF2 induction (Figure R3).

These data are currently not shown in the manuscript but can be added upon the reviewer's request.

Figure R3: Combined NRF2 induction by CDDO-Me and overexpression of FOSL1 in SCLC cells does not lead to a desensitization of cells to TXNRD1 inhibition. (A, B) HCC33_{FOSL1} and H69_{FOSL1} cells with a tet-inducible FOSL1 expression construct were treated with doxycycline (DOX, 0.125 or 1 µg/ml) for 24 h to increase FOSL1 expression. FRA1 expression was verified using immunoblotting (A) and IF (B). (C) HCC33_{FOSL1} and H69_{FOSL1} cells were treated with decitabine (DAC, 0.5 µM) or azacitidine (AZA, 2.5 µM) three times over one week (on days one, three, and five). Afterward, the medium was replaced with DAC/AZA-free medium. Cells were seeded in a 96-well plate and pre-treated with DOX and/or CDDO-Me (50 nM) for 24 h to increase FRA1 and NRF2 expression respectively. Then the cells were treated with a concentration series of DKFZ-682 for 24 h and cell viability was measured by the CellTiter-Glo assay. The graph is representative of two independent experiments each performed in triplicate. The expression of FRA1 and NRF2 was verified using immunoblotting.

An additional option we tested is STAT3/5. As shown by us previously, STAT3/5 is involved in the repression of ACBs in KEAP1 wild type NSCLC cell lines [1]. Inspired by the finding that pre-treatment of low-ACB NSCLC cells with the STAT3 inhibitor C188-9 increased the efficacy of NRF2 to induce ACB genes

and resistance to TXNRD1 inhibition, we pre-treated a panel of 12 SCLC cell lines with C188-9 to test whether STAT3/5 inhibition would allow NRF2 induction to induce resistance in SCLC cells. The results of this pilot experiment shows that STAT3/5 does not act as a repressor of ROS-scavenging pathways and is not involved in the failure of SCLCs to respond to NRF2 induction (Figure R4).

In summary, although we have systematically tested several plausible mechanisms, none fully explain the impaired NRF2-mediated ACB induction in SCLC. We believe this remains an important and open question for future studies.

Figure R4: Inactivation of STAT3 does not enhance cell resistance to ROS-inducing drug. (A) SCLC cell lines were treated with a concentration series of C188-9 for 48 h and cell viability was measured by the CellTiter-Glo assay. **(B)** SCLC cell lines were pre-treated with CDDO-Me (50 nM) and 3 μM C188-9 or a combination of both for 24 h. Next, the cells were treated with series dilutions of DKFZ-682, and after 24 h, cell viability was quantified by the CellTiter-Glo assay. The graphs are representative of two independent experiments each performed in triplicate.

4. *The manuscript lacks critical details regarding the experimental procedures, particularly in the animal studies:*

Figure 2 Legend: The authors state that “tumor-bearing mice were distributed into groups (n=10) and treated with cisplatin/etoposide (chemo) for 3 weeks or DKFZ-608 for 40 days.”

In Lines 203-204, where it is mentioned that the second group received 3 cycles of cisplatin (3 mg/kg, i.p.) and etoposide (7.5 mg/kg, i.p.).

For these in vivo studies,

(1) How was the tumor model established?

(2) How many cells were injected into the nude mice?

(3) What was the dosing schedule and interval for the “3 cycles” of cisplatin and etoposide?

We appreciate the comment of the reviewer as it shows us that we haven't transferred enough of the details described in the method section to the results text and figure legends. When composing the manuscript, we decided to mention those details in the method section in order to improve the readability of the results section. In the following we will elaborate on the individual points raised:

4.1. and 4.2. As detailed in the Method section, Animal experiments, we injected 3×10^7 H209 cells in 100 μ L of ice-cold PBS subcutaneously into the flank of female mice. We also stated that we waited until animals had developed tumors of palpable size (4-8 mm) which took 6 weeks for H209, before randomly distributing them to experimental groups.

4.3. We wrote in the method section, Animal experiments, “Specifically, 0.2 ml of each drug solution was injected per 20 g mouse to obtain a dose of 15 mg/kg DKFZ-608 (starting with dose escalation of 5->15 mg/kg in the first 5 days, continued with 15mg/kg up to day 40, daily i.p. injections), 3 mg/kg cisplatin (i.p. injection on Monday) and 7.5 mg/kg etoposide (i.p. injection on Wednesday and Friday).” Bolded words were added to ensure the treatment regimen is described clearly and without ambiguity. We also added this information to the figure legend of **Fig. 2**.

Similarly, Figure 6: The number of H526 cells injected to establish the tumor model is not specified. This information is essential for reproducibility and clarity. The authors need to provide detailed descriptions of the experimental procedures, including the tumor establishment procedures, cell numbers injected, and treatment protocols, to ensure transparency and enable proper evaluation of the study.

In the method section Animal Experiments, we write: “To generate a subcutaneous SCLC-tumor model, H526 cells were resuspended in a 1:1 (vol/vol) mix of ice-cold growth factor-reduced Matrigel (Corning) and PBS. Overall, 100 μ L of this cell suspension containing 2×10^6 H526 cells was injected subcutaneously into the flank of anesthetized mice.” We also stated that we waited until animals had developed tumors of palpable size (4-8 mm) which took 2-3 weeks for H526, before randomly distributing them to experimental groups.

In order to improve the structure of the “Animal experiment section” we regrouped the statements into Animal Handling, Determination of the Maximum Tolerated Dose (MTD), Generation of Subcutaneous SCLC-Tumor Models, Drug Treatments. Also, we added the sentence “(For details see method section Animal experiments)” to figure legends 2 and 6.

5. *In Figure 1A, the authors state that “the expression of 15 ACB genes for each SCLC and NSCLC cell line was calculated as average and based on the CCLE dataset and PDX, obtained from Champions Oncology (left panel). Units were converted to $\log_2(\text{TPM}+1)$.” However, the following points require clarification: Data Type: The authors need to specify the nature of the data obtained from Champions Oncology. Is this transcriptional data (RNA-seq) or protein expression data (proteomics data)?*

Champions Oncology has provided us with RNA-seq data, converted to $\log_2(\text{TPM}+1)$. We have added this information to the legend of **Fig. 1**.

Averaging Method: How was the expression data “averaged” across the 15 ACB genes? Was this done per sample or across multiple samples?

We have averaged the expression values of the 15 ACB genes per sample. We have added this information to figure legend 1.

Normalization: What normalization method was used to compare the SCLC and NSCLC data? Was the $\log_2(\text{TPM}+1)$ transformation applied after averaging or at the individual sample level? As this figure presents a key dataset for the manuscript, providing detailed and transparent information regarding data type, processing, and analysis methods is crucial for understanding and reproducing the results.

$\log_2(\text{TPM}+1)$ transformation was done on the individual sample level and averaging was done with such normalized values. Accordingly, we added the following sentence to the figure legend 1A: For expression data normalization, values were transformed to $\log_2(\text{TPM}+1)$ units. After that average expression value for the ACB gene set were calculated for each sample (using $\log_2(\text{TPM}+1)$ expression values of individual genes).

6. *In Figure 1B, the authors compare ACB protein expression between SCLC and NSCLC but do not include data from non-cancerous cell lines. This comparison is crucial to support the claim that TXNRD1 inhibition can widen the therapeutic window for SCLC while causing limited toxicity to normal lung tissue. The authors should include the ACB protein expression levels in non-cancerous cell lines alongside SCLC and NSCLC data.*

We appreciate the comment of the reviewer as it concerns an aspect of the project, which is indeed essential for the observation and the outlook. We have added protein quantifications of HaCaT and Beas-2B to **Fig. 1B** (see below).

Fig. 1 (B) The heatmap with the protein levels of SLC7A11, GSR and TXN, analyzed in total cell extracts by immunoblotting. Protein expression in H1944 was set to 1 for each protein and for each experiment. Relative data represent mean of independent experiments ($n=2-6$).

Moreover, we have adapted the order of SCLC cells to Fig. 1D

In previous study [1] we established that ACB transcripts correlate very well with the corresponding proteins. Therefore, the protein-level comparison of those cell lines serves as an extension of **Figure 1A**, which shows that, based on averaged ACB transcript levels, SCLC cells with low ACB expression are more sensitive to TXNRD1 inhibition.

However, it is important to note that the 3 members of the ACB-set, selected here for Western blot quantification (SLC7A11, GSR and TXN) may not be fully representative for the overall ROS-scavenging capacity in every cell type. In developing the ACB concept, we found that averaging the expression of 15 combined ACB genes provides a much more robust and reliable prediction of drug sensitivity than relying on individual ACB genes or proteins. To make this finding more accessible to the reader of the manuscript, we have modified the introduction (lines 123-126) in the first result section as follows: *“We previously identified a set of 15 antioxidant capacity biomarkers (ACBs) involved in interconnected redox pathways. By calculating the ACB score as the average expression of these genes, we were able to accurately predict sensitivity to redox-targeting agents, such as inhibitors of TXNRD1 and GPX4.”*

7. *In Lines 176-179, the authors state: “Our results demonstrate that SCLC cells are highly sensitive to TXNRD1 inhibitors and that cisplatin-resistant SCLC cells maintain this sensitivity, suggesting that TXNRD1 inhibition is a promising strategy for SCLC therapy.” Additionally, in Figure 2A, the in vivo experiments show a significant portion of animals experiencing remission with DKFZ-608 treatment.*

Has the sensitivity of DKFZ-608-resistant SCLC cells to TXNRD1 inhibitors been tested? This is a critical question to evaluate whether resistance to DKFZ-608 develops over time and whether cross-resistance to other TXNRD1 inhibitors occurs.

We thank the reviewer for this important question as we had spent many months to establish DKFZ-608 and DKFZ-682 resistant cells in tissue culture and failed to select for drug resistant survivor cells. These experiments were performed with multiple SCLC cell lines (H209, H69 and H526) and various doses of inhibitors (EC₅₀-EC₉₀). Those “negative” data have not been included in the manuscript as we wanted to avoid overloading the manuscript with data. Since we feel that conclusions should not be drawn based on “data not shown” we did not mention our failed attempt to select for cells with acquired resistance to DKFZ-682 or DKFZ-608 in cell culture.

Selecting resistant survivor cells after cisplatin treatment, using the same method applied for DKFZ-608 and DKFZ-682, was, successful (**Fig. S3A, B**), arguing that the experimental procedure was suitable to select for resistant cells.

Since attempts to generate resistant cells in culture were unsuccessful, we turned to the in vivo models and analysed H526 tumors resected from mice treated with either single CDDO-Me, DKFZ-682, or their combinations over a period of up to 3 weeks (**Fig. 6G** and **Fig. S12D**). In lines 384-387 we stated *“Importantly, tumor cells isolated from CDDO-Me-treated mice did not show significant signs of acquired resistance to DKFZ-682 (Fig. 6G) and remained unresponsive to NRF2 induction when tested ex vivo. This suggests that CDDO-Me treatment did not promote the development of resistance in SCLC tumors (Fig. S12D)”*.

We have not tested the development of possible cross resistance to other TXNRD1 inhibitors.

8. *The authors need to address inconsistencies and clarify data integrity: Supplementary Figure 7A vs. Figure 3A: In Supplementary Figure 7A, SCLC cell line H82 shows higher NRF2 induction than the non-cancerous Beas-2B cell line. However, in Figure 3A, the summarized data show the opposite, with higher NRF2 induction in Beas-2B than in H82. Similarly, H526 demonstrates much higher NRF2 induction than H82 in Supplementary Figure 7A. However, in Figure 3A, the two cell lines appear to have similar NRF2 induction levels. Further, H526 (SCLC) shows higher NRF2 induction than the non-cancerous cell lines HaCaT and Beas-2B in Supplementary Figure 7A. Yet, Figure 3A*

summarizes NRF2 induction as significantly higher in non-cancerous cells than in H526. The authors should clarify these discrepancies and ensure data consistency between the figures. Figure 3E: The authors did not specify the source of the data presented. Is this transcriptional (e.g., RNA-seq) or translational (e.g., protein expression) data? Without this clarification, it is difficult to interpret or validate the conclusions drawn from Figure 3E.

We thank the reviewer for pointing out this inconsistency in the data. We re-analysed the Western blots and quantifications and reformatted Fig. 3A to improve resolution (bar graphs > colour code) and transparency of the data (individual experiments indicated).

Fig. 3... (A) The NRF2 protein level in total cell extracts was analyzed in SCLC and non-cancerous cell lines upon treatment with DMSO (control) or CDDO-Me by immunoblotting. NRF2 expression in DMSO treated H82 cells was set to 1 for each experiment. Relative data represent mean of independent experiments (n=2-6; *p < 0.05, **p < 0.01, ***p < 0.001, ****p < 0.0001, two-tailed unpaired t-test).

Over the course of this study, we performed up to 11 Western blots, based on independent experiments and carried out by different team members. This extensive dataset allows to ensure statistical robustness despite inherent variability of the method. Given the volume of data, selecting representative images was challenging, however, we have now conducted additional replicates and selected 2 representative blots which replace the image in Fig. S7A.

Fig.S7 (A-C) The cells were treated with DMSO (control), CDDO-Me, dimethyl fumarate (DMF) or CPUY192018 (CPUY) for 3 h and the protein level of NRF2 was analyzed by immunoblotting. The blots are representative of up to 6 independent experiments.

Moreover, we have changed the colour code of DMSO versus CDDO-Me treatment in **Fig. 3B** to improve consistency with **Fig. 3A**.

Figure 3E: The authors did not specify the source of the data presented. Is this transcriptional (e.g., RNA-seq) or translational (e.g., protein expression) data? Without this clarification, it is difficult to interpret or validate the conclusions drawn from Figure 3E.

The data used by the Ingenuity Pathway Analysis software was RNA-seq data. Accordingly, we have modified the text in the figure legend 3:

9. *The data presented in Supplementary Figure 8C are incompletely interpreted. Both TXNRD1 and GSR are NADPH-dependent antioxidant enzymes that rely on NADPH for their activity. Consequently, when G6PD inhibition (G6PD-i) reduces NADPH generation, the protective effect of CDDO-Me is impaired.*

We thank the reviewer for this comment. GSR in addition to TXNRD1 are indeed NADPH-dependent components of ROS-scavenging pathways, similar to other members of the ACB-set, like AIFM2 (FSP1), AKR1C3, ALDH3A1 and CBR1. Accordingly, G6PDi-1 can be expected to “hit” multiple nodes in ROS-scavenging/redox homeostasis pathways, which are interconnected through their need of reductive equivalents, provided by NADPH. In a pilot study, we attempted to pinpoint key nodes driving resistance to ROS-inducing drugs by overexpressing or knocking down individual components. The absence of a single dominant factor component suggests that, at least in the lung cancer cells investigated, ROS-scavenging operate through a complex, interconnected, and redundant network.

Based on these findings, we chose to draw a rather general conclusion in stating that “... *NADPH-dependent pathways can partially compensate when GSH-mediated redox homeostasis is exhausted*”. In order to clarify how broad the biochemical impact of impaired NADPH regeneration can be, we suggest to insert the following sentence in line 268: “..... *inhibited with G6PDi-1 [4] (Fig. S 8B), most likely by reducing the activity of multiple NADPH-dependent antioxidant enzymes like TXNRD1, GSR, AIFM2 (FSP1), AKR1C3, ALDH3A1 and CBR1.*”

10. *The manuscript lacks an in-depth investigation into the mechanisms by which TXNRD1 inhibitors induce cell death and what types of cell death.*

We thank the author for this comment. When working on this project, we made an effort to investigate every aspect with multiple cell lines, in order to assess the general validity of a given statement. Cell death

mechanisms induced by DKFZ-682 in SCLC appear to be rather heterogeneous in that two cell lines, which demonstrated high induced levels of hydrogen peroxide (H_2O_2) and hydroxyl radical ($HO\cdot$), experienced an iron dependent cell death which was independent of ferroptosis, as it could be rescued by the iron chelator DFO but not by ferrostatin. Interestingly, in two of the investigated cell lines (HCC33 and H1105) we detected clear evidence of lipid peroxidation, which was suppressed by ferrostatin-1, but to our surprise without an impact on cell viability (Fig. S4A-E). A similar observation has been made with the ROS-inducing compound Ironomycin [5] in AML cells, another cancer entity demonstrating particularly low ROS-scavenging capacity [1].

Upon request of Reviewer #1 and #3 we have expanded our analysis of cell death mechanisms. First, we confirmed that ROS induction was indeed a major driver of cell death by pretreating cells with N-acetyl-L-cysteine (NAC), which, according to previous reports acts either directly as a weak ROS scavenger, boosts GSH production or serves as a donor of sulfane sulfur species, which scavenge ROS. We have added this set of data as Fig. S4F.

F

Fig.S4 (F) HCC33, H1105 and H82 cell lines were treated with 3 mM NAC for 30 min, followed by 24h treatment with a concentration series of DKFZ-682. Cell viability was measured by the CellTiter-Glo assay ($n=2$).

Using inhibitors of caspase- and PARP-dependent apoptosis (Z-VAD and olaparib) showed that drug-induced cell death in the cell line H82 could not be reduced efficiently at any concentration of DKFZ-682. In H1105, however, we observed a moderate reduction of toxicity upon pretreatment with olaparib. We also investigated whether treated cells switch between apoptosis and necroptosis and found that cotreatment of inhibitors, which block both apoptosis and necroptosis (Z-VAD and Nec-1) does not reduce cell death. We have added the modified set of data as Fig. S4G.

Fig.S4 (G) H1105 and H82 cell lines were pre-treated with DMSO, necrostatin-1 (Nec-1, 10 μ M), Z-VAD-FMK (20 μ M) alone or in combination (Z-VAD + Nec-1), or olaparib (100 nM). Then the cells were treated with a concentration series of DKFZ-682 for 24 h and the cell viability was measured by the CellTiter-Glo assay. Results are shown as the fold change in EC_{50} for DKFZ-682 in the presence of cell death inhibitors compared to the DMSO-treated control ($n=2-4$, each performed in triplicate; $*p < 0.05$, two-tailed unpaired t-test).

Since none of the inhibitors of apoptosis, necroptosis or ferroptosis were able to significantly reduce drug-induced cell death, we speculated (at the end of supplementary section 1) *that TXNRD1 inhibitors like DKFZ-682 induce **paraptosis** [6], a non-apoptotic/necroptotic cell death which depends on the de-novo translation of Activating Transcription Factor 4 (ATF4) to execute cell death induced by redox-targeting drugs [7] [8]. In line with those reports we observed that DKFZ-682 induces ATF4, and that the pretreatment of cells with translation inhibitor cycloheximide (CHX) prevented drug induced ATF4 expression and reduced DKFZ-682 toxicity (Fig. S4H), suggesting that DKFZ-682 induces paraptosis.*

Moreover, we also show that DKFZ-682 ultimately causes necrosis [9], *which is evident by the release of LDH to the medium (Fig. S4I).*

(H) Cells were pre-treated with DMSO (control) or cycloheximide (CHX, 25 μg/ml) for 1 h and then treated with DKFZ-682 for 24 h. Protein levels of ATF4 in total lysate were analyzed by Western blotting (representative of 2 independent experiments). β-Tubulin served as a loading control to take into account potential effects of CHX on global protein expression. Cell viability was analyzed by CellTiter-Glo assay. The graph (EC₅₀ mean ± SD; * $p < 0.05$, *** $p < 0.001$, two-tailed unpaired t-test) summarizes the data of two to three independent experiments each performed in triplicate. **(I)** Cells were treated with the lowest full effect dose (3 μM) or 10 μM of DKFZ-682, and lactate dehydrogenase (LDH) release was quantified at 3, 6, and 24 hours post-treatment. The LDH level in lysed cells served as a reference for maximum release. Three biological replicates were performed for each cell line. Significance of differences was calculated with a two-tailed unpaired t-test (* $p < 0.05$, ** $p < 0.01$, *** $p < 0.001$, **** $p < 0.0001$). Non-significant differences between treatment groups are not specifically marked.

To maintain a clear and focused narrative, we structured the cell death analysis into two parts: the main results section, highlights the discovery of an iron-dependent, non-ferroptotic cell death, while Supplementary Section 1, systematically addresses the exclusion of classical programmed cell death pathways. This approach was taken to avoid overloading the main text with negative data and to guide the reader through the mechanistic conclusions in a stepwise manner. We recognize that this division may make it challenging to gain a comprehensive overview of our investigation, and we are willing to merge these sections if preferred.

Reviewer #2 (Remarks to the Author): Expert in targeted therapy of lung cancer

In this study, Jana Samarin et al. demonstrate the potential of TXNRD1 inhibitors as a promising maintenance therapy for small cell lung cancer (SCLC). SCLC cells, which exhibit low reactive oxygen species (ROS) defence, are hypersensitive to TXNRD1 inhibition and fail to adapt to drug-induced ROS stress due to epigenetic and transcriptional suppression of the defence system. Through pharmacologically activating the NRF2 stress response pathway, the authors are able to safely increase the dose of TXNRD1 inhibitors, improving tumor control without additional toxicity to healthy tissues. These findings suggest that TXNRD1 inhibitors could effectively overcome resistance in relapsed SCLC cases, offering a novel therapeutic approach for controlling this challenging disease. The idea behind this work is very interesting and indeed brings certain conceptual advance. I have some minor concerns for the authors to further improve the manuscript.

We would like to thank the reviewer for their positive evaluation of our study and highly appreciate the constructive and thoughtful feedback provided.

- 1. The authors state that the antioxidant capacity biomarker (ACB) scores of pulmonary neuroendocrine cells (PNECs) are similarly low to those of SCLC. Given this, the authors might want to consider carefully about the potential side-effects of TXNRD1 inhibitors upon the nervous system in mouse models, or at least make some discussion.*

We appreciate the reviewer's comment highlighting the importance of differential cytotoxicity in cancer therapies. Since PNECs originate from endodermal epithelial progenitors rather than the neural crest—as evidenced by their lack of neural crest markers—it did not initially seem obvious to specifically address neural toxicity in the context of this manuscript.

However, considering reports, which claim that neuronal cells have lower steady state levels of ROS scavenging enzymes and potentially a lower NRF2 mediated response to induced stress, the question of intrinsic neurotoxicity of redox targeting drugs is relevant. Our decision not to specifically discuss potential neurotoxicity of TXNRD1 inhibitors also stems from our in vivo findings: prolonged treatment at tumoricidal concentrations was well tolerated by mice, and even during MTD determination—which involves brief overdosing—no signs of neuronal toxicity were observed.

In order to make this remarkable observation transparent to the reader, we have added the following sentence to the discussion (lines 408-410): *It is also noteworthy that no signs of neurotoxicity were observed, even though redox homeostasis plays a crucial role in neurotransmission and neuronal cells possess relatively low steady-state levels of ROS-scavenging enzymes[10].*

- 2. DKFZ-608 seems have higher potency against SCLC and reduced selectivity towards non-cancerous cells. Given this, it remains unclear why the authors chose to use DKFZ-682 in subsequent experiments.*

We thank the author for this question as it has been raised by multiple consultants to the project. Before answering, we would like to hint at a potential mistake in the phrasing of the question. DKFZ-608 is the more selective compound, as compared to DKFZ-682. One reason for this is that it has reduced toxicity towards non-cancerous cells. We assume that this is the statement intended by the reviewer.

To better explain why we chose DKFZ-682, we have modified the introductory paragraph of the last result chapter (“Widening the therapeutic window for redox-targeting drugs”) as follows (lines 320-333): *“Given that SCLC cells are unable to adapt to redox-targeting drugs, such as TXNRD1 inhibitors, we hypothesized that activating the NRF2 pathway would enhance antioxidant defenses specifically in non-cancerous cells, thereby further widening the therapeutic window. To test this hypothesis in vivo, we chose an experimental design which highlighted aspects of a narrow therapeutic window. First, we utilized the hydroxy-substituted derivative DKFZ-682. While this compound demonstrates a fast cytotoxic effect and reaches plasma concentrations sufficient for tumor control (Fig. S13C, D), it is less potent than DKFZ-608 against SCLC cells and shows reduced selectivity toward non-cancerous cells (Fig. S2D). In addition, its lower maximum tolerated dose (MTD) of 4 mg/kg (Fig. S13A, B) may limit its therapeutic efficacy. Second, in order to monitor drug efficacy more precisely, we administered DKFZ-682 as monotherapy rather than maintenance therapy, allowing us to eliminate confounding effects from first-line therapy. Lastly, we chose a tumor model based on H526, which exhibits partial resistance to TXNRD1 inhibition (Fig. S2D), to ensure that tumor growth would not be fully suppressed at the 4 mg/kg MTD in unprotected mice.”*

In other words, the weakness of DKFZ-682 (reduced selectivity) made this compound particularly suitable to demonstrate that dose limiting toxicity of a redox targeting drug can be managed without losing antitumor efficacy (at least in low-ACB cancers). Having decided for DKFZ-682 in the mouse experiment (**Fig. 5, 6**), we also used it for mechanistic studies in cells in order to maximize the connect between *in vitro* and *in vivo* experiments.

According to the data presented in **Fig. 1D** and **Fig. S2D**, auranofin is an even stronger example of a TXNRD1 inhibitor with reduced selectivity. Its limited success in clinical trials could be attributed to the lack of appropriate stratification for susceptible, low-ACB tumors. An additional factor is its dose-limiting toxicity, which, we speculate, could be reduced by cotreatment with an NRF2-inducer.

On request of reviewer #1 we have now added data on the determination of MTD and plasma concentration as a new **Fig. S13**.

3. *Other issues: statistical analyses for the mouse experiments should be included in Figures 2A, 2B, 2C and 5A. In Figure 4D, the scale bar should be consistent across all panels. In Figure 5B, if not significant, please indicate "ns".*

We thank the reviewer for this valuable comment and added statistical analysis and significance calls to the indicated figures and corresponding figure legends.

For **Figure 4D**, we have adjusted the scale:

Reviewer #3 (Remarks to the Author): Expert on cell death and TXNRD1 inhibitors

Tumor relapse poses a significant challenge in the treatment of SCLC patients. In this study, the authors discovered that SCLC cells exhibit heightened sensitivity to TXNRD1 inhibition due to their impaired ROS defence mechanisms. Notably, even SCLC cells resistant to first-line therapies remain susceptible to TXNRD1 inhibition. This suggests that targeting TXNRD1, in combination with standard first-line treatments, could represent a promising therapeutic strategy. Furthermore, the authors observed that high doses of TXNRD1 inhibitors may induce host toxicity. Activation of the NRF2 pathway was found to protect host cells from the toxic effects of TXNRD1 inhibitors without conferring protection to SCLC cells. Consequently, co-administration of an NRF2 activator could potentially widen the therapeutic window for TXNRD1 inhibitors in the treatment of SCLC. Overall, this study is well-conducted, and the findings are both intriguing and of substantial scientific value. This reviewer only has one major point and a few minor suggestions to offer.

We want to thank the reviewer for the positive assessment of our study and appreciate the constructive points raised.

Major point:

1. Figure 2: The anti-tumor effect of DKFZ-608 appears highly promising. However, to substantiate the claim that DKFZ-608 effectively eradicates first-line therapy-resistant SCLC cells, it is crucial to demonstrate that these cells are indeed resistant to first-line treatment. In the current study, the authors did not administer any treatment to the "Chemo" group mice after tumor relapse, leaving it unclear whether the relapsed tumors were resistant to first-line therapy. To support this claim, the authors should include two additional experimental groups. Both groups would initially receive three cycles of first-line treatment and be monitored until tumor relapse. Upon relapse, one group would be treated with the first-line therapy again, while the other would receive DKFZ-608. If DKFZ-608 successfully eliminates the tumors while the first-line treatment fails, the conclusion that DKFZ-608 targets resistant SCLC cells would be well-supported.

You raised the question whether DKFZ-608 is able to eradicate first line therapy-resistant SCLC cells. In other words, are TXNRD1 inhibitors, like DKFZ-608 or DKFZ-682, unaffected by resistance mechanisms which blunt the efficacy of cisplatin? This point is indeed highly relevant for the assessment of the potential of a TXNRD1 inhibitor as maintenance therapy after/upon first line therapy.

When planning the proof of concept for this hypothesis, we felt obliged by the 3-R principle (replacement, reduction, refinement) to design the study such that the central question concerning the efficacy of maintenance therapy is clearly answered by mouse experiments, and that plausibility aspects, like resistance breaking, could be assessed without the use of additional animal groups. By comparing tumor growth in animals which had received only chemotherapy versus chemotherapy, followed by DKFZ-608 treatment, we could show that maintenance therapy with a TXNRD1-inhibitor had eliminated all residual tumor cells which otherwise give rise to the relapsing of tumors we observed in the chemo group. By explanting tumors from relapsed animals (chemo group) we were able to show, without the use of additional animals, that those tumor cells had developed substantial resistance against cisplatin (> 40-fold) while their response to DKFZ-608 remained largely unaffected (**Figure 1E, Figure S3B**).

In addition to our policy to adhere as much as possible to the 3-R principle, there is a practical hurdle to perform the experiment as suggested by the reviewer: The experimental design used for this study involves 3 cycles of chemotherapy. However, repeated cisplatin dosing to mimic consolidation therapy is limited by cumulative toxicity, which we also observed and which has been extensively reported in the literature [11]. While low-dose cisplatin consolidation regimens do exist in the literature [12], we believe that incorporating such protocols is beyond the scope of our current study, which is focused on evaluating the potential of TXNRD1 inhibitors as maintenance therapy.

We hope that the reviewer agrees that the assessment of cisplatin resistance in explanted tumors provides a suitable replacement for the requested treatment groups and respectfully propose not conduct additional mouse experiments.

Minor points:

1. *Figure S1A: It would be valuable to include the expression levels of TXNRD1 and GPX4 in SCLC cell lines.*

As suggested, we have added protein quantifications for GPX4 and TXNRD1 to **Figure S1A**.

(A) Protein levels of selected ACBs and two additional redox proteins, TXNRD1 and GPX4, in total cell extracts from 13 SCLC cell lines were analyzed by immunoblotting ($n=2$). High ACB protein expression in drug resistant H1944 NSCLC cell line was used as a reference.

2. *Figure S2A: RSL3 may exhibit off-target effects on other selenoproteins, including TXNRD1, particularly at higher concentrations. To ensure specificity, the authors should consider using ML210, a more selective GPX4 inhibitor. Additionally, including a ferroptosis inhibitor (e.g., ferrostatin-1 or liproxstatin-1) as a control would strengthen the study. If the observed effects are indeed mediated by GPX4 inhibition, cell viability should be fully restored by the ferroptosis inhibitor.*

This is a point well taken. RSL3 has indeed been reported to inhibit TXNRD1 activities at high concentrations and we observed effects in experiments unrelated to this study which appear to support this notion. Following the reviewer's suggestion, we added ML210, a more selective nitroisoxazole-GPX4 inhibitor. ML210, like RSL3, shows a more heterogeneous response profile in SCLC cells, as compared to TXNRD1 inhibitors, which substantiates our finding. Accordingly, we have expanded **Figure S2A, B** by cell line panel data showing the respective EC₅₀ values. In addition, we modified the following sentence (line 148) *"The homogenous activity profiles of our TXNRD1 inhibitors is especially notable compared to 1S3R-RSL3 (RSL3) and the selective nitroisoxazole-GPX4 inhibitor ML210 [19], all of which exhibits a more variable response pattern, independent of the cell's*

ASCL1/REST or NEUROD1/REST status (Fig. S 2A, B)". We have adjusted the text in figure legend S2 accordingly.

Figure S2: The cytotoxic effect of *TXNRD1* inhibition is equal high for all tested SCLC cell lines, independent of their resistance to the GPX4 inhibitors 1S3R-RSL3 (RSL3), ML210 or to cisplatin. (A-D)

While ML210 exhibited a heterogeneous activity profile similar to RSL3—unaffected by the cell's *ASCL1/REST* or *NEUROD1/REST* status—the correlation between their activity patterns was weaker than anticipated ($r = 0.63$). This discrepancy may stem from RSL3's distinct off-target effects, which could influence its activity profile compared to ML210.

In order to demonstrate that the observed effects are involving ferroptosis mechanisms, we included ferrostatin rescue data for the cell lines H82 and HCC33, which are highly susceptible to ferroptosis induction. The third cell line, H1105, represents ferroptosis-resistant SCLC's where cell death cannot be rescued by ferrostatin. We present the data in an expanded **Fig. S4D**, which collects data describing cell death mechanisms and modified the text in the figure legend:

(D) Cell viability of indicated cell lines, pre-treated with Fer-1 (5 μ M) for 1 h and treated with DKFZ-682 for 24 h or RSL3 or ML210 for 48 h was analyzed by CellTiter-Glo assay. The graphs are representative of two independent experiments each performed in triplicate.

3. Figure S4D: The authors could use N-acetyl-L-cysteine, an antioxidant, to assess whether it rescues the observed cell death, which would further clarify the role of oxidative stress in the mechanism.

As suggested, we have performed experiments testing the hypothesis that pre-treatment of SCLC cells with NAC decreases the toxicity of DKFZ-682 and added the date as a new Fig. S4F.

Fig S4.....(F) HCC33, H1105 and H82 cell lines were treated with 3 mM NAC for 30 min, followed by 24h treatment with a concentration series of DKFZ-682. Cell viability was measured by the CellTiter-Glo assay.

4. Figure S8D: The unit of the y-axis should be indicated.

We thank the reviewer for indicating this mistake. As requested, we have corrected Fig. S8D and added the units (RLU) to the y-axis.

References

1. Samarin, J., et al., *Low level of antioxidant capacity biomarkers but not target overexpression predicts vulnerability to ROS-inducing drugs*. Redox Biol, 2023. **62**: p. 102639.
2. Krenske, E.H., R.C. Petter, and K.N. Houk, *Kinetics and Thermodynamics of Reversible Thiol Additions to Mono- and Diactivated Michael Acceptors: Implications for the Design of Drugs That Bind Covalently to Cysteines*. J Org Chem, 2016. **81**(23): p. 11726-11733.
3. Kepa, J.K. and D. Ross, *DT-diaphorase activity in NSCLC and SCLC cell lines: a role for fos/jun regulation*. Br J Cancer, 1999. **79**(11-12): p. 1679-84.
4. Ghergurovich, J.M., et al., *A small molecule G6PD inhibitor reveals immune dependence on pentose phosphate pathway*. Nat Chem Biol, 2020. **16**(7): p. 731-739.
5. Garciaz, S., et al., *Pharmacologic Reduction of Mitochondrial Iron Triggers a Noncanonical BAX/BAK-Dependent Cell Death*. Cancer Discov, 2022. **12**(3): p. 774-791.
6. Sperandio, S., I. de Belle, and D.E. Bredesen, *An alternative, nonapoptotic form of programmed cell death*. Proc Natl Acad Sci U S A, 2000. **97**(26): p. 14376-81.
7. Seo, M.J., et al., *Dual inhibition of thioredoxin reductase and proteasome is required for auranofin-induced paraptosis in breast cancer cells*. Cell Death Dis, 2023. **14**(1): p. 42.
8. Lin, Y.S., et al., *Arnicolide D induces endoplasmic reticulum stress-mediated oncosis via ATF4 and CHOP in hepatocellular carcinoma cells*. Cell Death Discov, 2024. **10**(1): p. 134.
9. Vanden Berghe, T., et al., *Necroptosis, necrosis and secondary necrosis converge on similar cellular disintegration features*. Cell Death Differ, 2010. **17**(6): p. 922-30.
10. Franco, R. and M.R. Vargas, *Redox Biology in Neurological Function, Dysfunction, and Aging*. Antioxid Redox Signal, 2018. **28**(18): p. 1583-1586.
11. Su, H.W. and C.W. Qiu, *A comparative review of murine models of repeated low-dose cisplatin-induced chronic kidney disease*. Lab Anim (NY), 2025. **54**(2): p. 42-49.
12. Ye, H., et al., *Sustained, low-dose intraperitoneal cisplatin improves treatment outcome in ovarian cancer mouse models*. J Control Release, 2015. **220**(Pt A): p. 358-367.

Dear editor

We herewith address all points raised by the reviewers.

REVIEWER COMMENTS

Reviewer #1 (Remarks to the Author): NRF2, KEAP1, redox biology and DNA methylation

Some of the concerns have been addressed, but several important points remain unresolved.

1. While the authors tested plausible mechanisms (e.g., promoter methylation, MAFG/FRA1 levels, STAT3 repression), no definitive explanation is provided for the failure of NRF2 to activate antioxidant capacity biomarkers (ACBs) in SCLC. This mechanistic gap limits the biological insight of the study.

We agree with the reviewer's view that a definitive explanation of why SCLCs fail to respond to NRF2 induction is not provided in this manuscript. We have expanded our efforts by asking whether promoter methylation and histone deacetylation, in combination with missing cofactors like MAFG or FRA1 could be responsible for the observation. Unfortunately, this was not the case. Earlier reports on the role of BRD4 as a modulator of the NRF2-pathway in SCLC¹ and other cell types^{2 3 4 5} were also tested, but we could not unlock the NRF2 regulon for CDDO-Me resistance induction by cotreatment with the BRD4 inhibitor JQ-1. We have now modified the discussion (lines 484-503) to inform the reader about which mechanisms are unlikely to contribute to the observed phenomenon. In case it is requested by the reviewer, we will provide the data in additional supplementary figures.

We certainly consider a mechanistic understanding of our observations as important. However, we strongly believe that the central discoveries of our work—namely the differential activity of NRF2 inducers like CDDO-Me and DMF in cancer cells compared to normal tissue, and the resulting improvement in therapeutic efficacy of stratified tumors—convey a novel and impactful message that should be shared with the scientific community by publishing it in this journal.

While the manuscript lacks a definitive explanation for why SCLCs do not respond to NRF2 induction, it provides a large body of data on what mechanisms are not involved. We are confident that our findings presented so far will spark others to join us in uncovering the yet unknown mechanism leading to differential NRF2 response.

2. Using a single dose of 50 nM CDDO-Me across cell lines may not fully capture NRF2 inducibility in SCLC. A dose-response analysis of NRF2 target gene expression or activity across cell types would strengthen conclusions about limited responsiveness in SCLC.

We thank the reviewer for this comment and fully agree that a dose response analysis is more suitable to evaluate the robustness and limits of a data driven hypothesis. We had shown in Fig. S8D that 50 nM CDDO-Me is the highest non-toxic effect dose in the SCLC cells of our panel. This “forced” us to avoid higher dosing as the resulting toxicity could obscure the observations. For the reader of the manuscript, this reason to restrict the observation to 50 nM might be insufficient, as a higher concentration of CDDO-Me might indeed induce ACB expression and drug resistance. We have now profiled the dose-dependent response to CDDO-Me treatment by monitoring shifts in drug sensitivity, the induction of NRF2 proxies as well as 6 selected ACB-genes in 4 SCLC lines. The highest dose of CDDO-Me was 100 nM, which causes mild toxicity in H82 and H526 (**see next page, Figure R2.1A-C**). The data show that 100 nM CDDO-Me leads to a more pronounced induction of some NRF2 proxies in some cell lines, as compared to 50 nM (e.g. HMOX1 in H82, HCC33 and H69; NQO1 only in H69). In contrast to this, most ACBs remain at their baseline expression at any concentration of CDDO-Me. The exception to this is GCLM, which shows a moderately higher expression at 100 nM in HCC33. Importantly, none of the cell lines exhibited changes in drug sensitivity at any concentration of CDDO-Me, reinforcing the manuscript’s central claim that NRF2 induction does not enhance the ROS-scavenging capacity of SCLC.

Figure R2.1: Dose-dependent effects of CDDO-Me on NRF2 target gene expression and drug sensitivity in SCLC. (A, B) The indicated cell lines were treated with DMSO (control), CDDO-Me (25 nM, 50 nM or 100 nM) for 6 h. The level of NRF2 target genes were analyzed by qPCR (Control without CDDO-Me was set to 1; relative data represent mean \pm SD of two independent experiments each performed in triplicates; * $p < 0.05$, ** $p < 0.01$, *** $p < 0.001$, **** $p < 0.0001$, one-way ANOVA, compared to the control). (C) The cells were pre-treated with DMSO (control) or CDDO-Me for 24 h, and then treated with dilution series of DKFZ-682 for 24 h. The cell viability was quantified by the CellTiter-Glo assay. The results are representative of two independent experiments, each performed in triplicates.

3. The NRF2 protein induction data in Supplementary Figure 7A appear inconsistent with Figure 3A still. These discrepancies should be reconciled clearly, either by standardizing quantification methods or providing an explanation for inter-assay variability.

We acknowledge that the modifications we introduced in Fig. 3A and Fig. S7A (now it is Fig. S8A) may still not fully address the reviewer’s concerns. The reviewer comment on inter-assay variability is valid and we can assure that we are fully aware of it, not only in this project but in many other instances where labeled antibody and densitometric methods are used for quantifications. The reasons for the inter-assay variability which we and many other labs experience is caused by

- Variations in background subtraction, affecting densitometric measurements.
- Differences in the linear dynamic range of antibodies, leading to signal saturation and non-linear quantification.
- Biological variability in cellular or tissue samples, such as changes due to cell line drift or batch-to-batch differences in culture or extraction conditions.

To compensate for such errors and the resulting shortcoming of western blot-based protein quantification, we do extensive repeating of the analysis and include all data in a summary figure, without suppressing outlier results. The presentation of representative blots follows common practice, but is not well-suited for quantitatively assessing the observed effects. I hope the reviewer can agree that the human eye (and the brain behind it) is not well equipped to accurately quantify grayscale images and will therefore sometimes disagree with quantifications done by a dedicated software. To illustrate this, we have added the reads of the densitometric analysis (without units) below the grayscale image.

We modified Fig. S7A (now it is Fig. S8A) by adjusting the exposure of the photographs such that the apparent pixel-signal of all H82 DMSO-treatments appear as equal. H82, treated with DMSO is set as 1 and serves as a reference to quantify other samples. In addition, we have replaced the blot representing NRF2 induction in HaCaT, as the previous one did not show H82 as a reference. In order to facilitate for the reviewer, the alignment of the representative blot to the summary bar graph, we have highlighted the corresponding datapoints in green: DMSO-treated cells or red: CDDO-Me treated cells.

Figure 3A

New Figure S8A

4. While liver TXNRD1 activity is shown, it is unclear whether DKFZ treatment leads to functional toxicity in normal tissues. Including histology, liver enzyme markers, or animal weight trajectories would help clarify on-target effects in non-cancerous cells.

We thank the reviewer for highlighting the importance of assessing potential functional toxicity in normal mouse tissues, specifically through the use of liver enzyme markers, blood urea, and body weight monitoring. These aspects were already included in the original manuscript, although they may not have been clearly highlighted.

For clarification:

- In Fig. 2C, the data show that treatment with DKFZ-608 alone does not alter body weight compared to vehicle control.
- Fig. S13B (now it is Fig. S5D) illustrates that mice treated with DKFZ-608 (15 mg/kg) and DKFZ-682 (4 mg/kg) for 34 consecutive days exhibited a steady increase in body weight without any observable weight loss.
- Fig. 5A presents results from high-dose DKFZ-682 (10 mg/kg) in combination with CDDO-Me, where a very mild reduction in body weight was seen relative to vehicle or CDDO-Me alone.
- Fig. 5C reports acute organ damage markers: a single injection of DKFZ-682 (4 mg/kg) resulted in increased blood urea nitrogen (BUN) and liver enzymes (AST, ALT) after 8 hours, indicating acute kidney and liver injury. Notably, co-treatment with CDDO-Me prevented this toxicity.
- Importantly, as shown in Fig. 5D, there was no observable kidney or liver damage even after 3 weeks of treatment with high-dose DKFZ-682 (10 mg/kg) administered simultaneously with CDDO-Me.

We think that the animal body weight data and stress markers for organ damage as presented in the manuscript are suitable to support the conclusion that CDDO-Me efficiently protects normal tissues against drug-induced toxicity.

5. A direct comparison to known TXNRD1 inhibitors like auranofin would contextualize the relative potency, selectivity, and toxicity profile of DKFZ-608/682.

We thank the reviewer for this valuable comment as we always try to evaluate novel findings in the context of current knowledge. Accordingly, we had compared various aspects of the activity profiles of DKFZ-608/682 versus auranofin in this manuscript and our previous work on DKFZ-682 in NSCLC⁶ but did not describe this comparison in a dedicated section. In the following we show the data describing similarities and differences of the 3 drugs.

Potency and selectivity: In Fig. 1D we show a heat map depicting the activity profile of the three TXNRD1 inhibitors (plus Cisplatin). In Figure S2D we show the same data as individual bar graphs and indicate differences in selectivity (DKFZ-608>>DKFZ-682>Auranofin, as a function of average EC₅₀ in the SCLC panel versus 2 non-transformed cells). However, we had not mentioned these differences in the main text as we had focused on the comparison of Cisplatin with DKFZ-608. We now added this information at line 167. The data shown in Fig. S2D could be presented in 2 alternative graphs, if requested by the reviewer.

We also compared the efficacy of DKFZ-608 and auranofin in circulating tumor cells (Fig. S3C) and stated in line 177-178 “a homogeneously high response to TXNRD1-inhibitor treatment, with DKFZ-608 being more efficient than auranofin”.

We also characterized changes in the subcellular redox status induced by each of the three TXNRD1 inhibitors by quantifying cysteine oxidation in cytoplasmic and mitochondrial roGFP2-Orp1 reporters (see below, Fig. S1E in this manuscript and Fig. 1C in our previous work on NSCLC⁶). The observations presented in this manuscript and the previous publication is mentioned in line 141-144, stating that

“the two molecules in contrast to auranofin, (are) predominantly acting in the cytoplasm with limited mitochondrial ROS induction”. This differential activity in mitochondria could explain why auranofin exhibits lower selectivity against non-cancerous cells. It is interesting to note that DKFZ-682 also shows the tendency to induce mitochondrial ROS-stress (at 30 μ M), which might explain its lower selectivity, compared to DKFZ-608.

Fig. 1E (this manuscript)

Data shown in Fig. 1C in Samarin et al. 2023

In our previous work on NSCLC⁶, we had attempted to compare the general reactivity of DKFZ-682 versus auranofin by preincubating the drug solutions with bovine serum albumin, which is known to bind cysteine reactive molecules. The data (see below) show that the potency of auranofin is more reduced by BSA than DKFZ-682, arguing that DKFZ-682 is less reactive under these conditions. In the same paper we compare the IC₅₀ of both drugs on TXNRD1 and GSR, showing comparable on-target activity but large differences on glutathione reductase inhibition.

Suppl. Fig. 1A in Samarin et al. 2023

	DKFZ-682	Auranofin
TXNRD1 IC ₅₀	6 nM	2.6 nM
GSR IC ₅₀	>300 μ M	196 nM
TXNRD1 (U498C) IC ₅₀	24 μ M	n.d.

Fig. 1A in Samarin et al. 2023

Toxicity: We have determined the maximum tolerated dose for daily i.p. dosing of DKFZ-608 (15 mg/kg) and -682 (4 mg/kg) and presented the data in Fig. S13A, B (now it is Fig. S5C, D). However, since we had not performed a side-by-side assessment of auranofin, we can only use literature data to compare systemic toxicity in mouse models. A study by Freire Boullosa et al. 2022⁷ stated that after five days of treatment with 10 mg/kg, mice showed clinical signs of cytotoxicity, discomfort and weight loss of approximately 20 % during the treatment period. This might explain why we mostly find the use of 2-4 mg/kg in published xenograft studies.

In small cell lung cancer (SCLC) xenograft model⁸, auranofin was administered at 4 mg/kg once daily for 14 days showing good tolerability without significant toxicity. This observation is comparable to

what we report for DKFZ-682 in this manuscript. At this dose, the authors observed a about 3-fold reduction of TXNRD1 activity in tumors (similar to an about 4-fold reduction by DKFZ-682 in our manuscript). Comparable to our experiments with H526, auranofin at 4 mg only moderately prolonged survival as a monotherapy in their DMS273 model.

We had shown in this manuscript that pharmacological NRF2-induction leads to selective protection of normal tissues, allowing an increase of the therapeutic dose. We expect that the therapeutic efficacy of TXNRD1 inhibitors like auranofin can be improved by the same concept.

6. A critical point that requires clarification is the role of KEAP1/NRF2 mutations in NSCLC. Approximately one-third of NSCLC cases harbor mutations in the KEAP1/NRF2 pathway. Therefore, it remains unclear whether the differential responses observed between SCLC and NSCLC are due to intrinsic lineage differences or simply reflect the absence of KEAP1/NRF2 mutations in SCLC cells.

We thank the reviewer for this comment, as it indicates that we haven't dealt with this aspect clear enough. In the discussion (line 483), we had mentioned that low steady state levels of NRF2 is "most likely mediated by post-translational mechanisms like KEAP1-mediated degradation". Also, we exclude under-expression of NRF2 transcripts (Fig. S6A, line 261) or overexpression of KEAP1 protein (in SCLC cells versus non transformed cells, Fig. R1 in our first response to the reviewers' comments, see below).

Figure R1: Protein levels of KEAP1 in total cell extracts from 13 SCLC and non-cancerous cell lines (Beas-2B, HaCaT) were analysed by immunoblotting (representative image, qualitatively similar to 2 biological replicates). The NSCLC cell line H1944 was used as reference.

Concerning the situation in NSCLC, we had performed an intensive study on resistance to TXNRD1 inhibitors in this cancer entity⁶. Here we showed that the majority of KEAP1-wild type NSCLC cell lines express very low levels of Antioxidant Capacity Biomarkers (which we baptized ACBs) and are hypersensitive to TXNRD1 (and GPX4) inhibitors. Comparable to SCLC cell lines in the current manuscript, those low-ACB NSCLC lines were reported to be responsive to KEAP1 inhibition by increasing NRF2 levels, but, like in SCLC, they were unable to translate this in the upregulation of ROS-defense mechanisms and drug resistance. As expected, and demonstrated in earlier studies, KEAP1 or Cul3-mutant cells express high levels of NRF2 and show high capacity to defend against drug induced ROS stress. Considering the fact that none of the SCLC lines **tested in this study** harbor KEAP1, Cul3 or NRF2 mutations, it is to be expected that the KEAP1 wild type status is a prominent determinant of low steady state NRF2 and ACB levels in those cell lines and we stated this in the discussion. In order to improve the context of our observations in SCLC, we now modified the section where we compared the NSCLC panel of our earlier study with the SCLC panel of this work by mentioning the KEAP1/Cul3 status of the cells. **Line 126:** "Non-small cell lung cancer (NSCLC) cell lines exhibit variable ACB expression levels. Lines harboring wild-type KEAP1 or non-loss-of-function KEAP1 mutations tend to show low basic ACB expression, whereas those with loss-of-function mutations in KEAP1 or CUL3 display elevated ACB levels. In small cell lung cancer (SCLC), ACB expression is low across most cell lines, consistent with their KEAP1 status—suggesting a high susceptibility to redox-targeting therapies."

Since we do not have access to the mutation status of the PDX models of Champions Oncology, we have now omitted the ACB data from Fig. 1A.

Moreover, the authors use H1944—an NSCLC cell line with a KEAP1 mutation—for comparison in several assays. This may not accurately represent the majority (~70 %) of NSCLC cases with wild-type KEAP1, and could confound the interpretation of NRF2 activity and drug response. This concern is clearly illustrated in Figure 1B.

We thank the reviewer for this comment. In Fig. 1B we not only show KEAP1 mutant H1944, H2122, HCC15, H1793 and H1573 but also a set of KEAP1 wild type NSCLC lines (H23, H522, H1693, HCC827 and H661) which represent the majority of NSCLC tumors. The heatmap shows that the protein levels of 3 representative ACBs are similar in KEAP1 wt NSCLCs and SCLCs. As both groups of (KEAP1 wt) cells demonstrate high sensitivity to TXNRD1 inhibition, we conclude that low expression of ROS-scavenging capacity (ACB) is responsible for the hypersensitivity of SCLCs to redox-targeting drugs.

We would like to clarify why, in some figures, we used H1944 rather than another NSCLC line as a reference. H1944 was used as a sole reference (but not as a representative of NSCLC) in Fig. S1A, where we show the expression of 5 ACB proteins plus TXNRD1 and GPX4, in Fig. S6C where we quantified NRF2 and in Fig. S10B (now it is Fig. S11B) where we quantified MAFG. H1944 was used to illustrate what levels of constitutive NRF2 or ACB proteins can be expected in a highly resistant cell line. We and others^{9 10 11} have established the correlation of constitutive NRF2 levels (as a function of KEAP1/Cul3 mutation status), ACB expression and sensitivity to redox targeting drugs in NSCLC, concluding that low steady state levels of NRF2 protein leads to low ROS-scavenging capacity resulting in high sensitivity to redox targeting drugs.

7. The authors suggest that promoter hypermethylation contributes to the downregulation of a subset of ACB genes and the NRF2 cofactor MAFG, as shown in Figure 4. This correlation is observed in both cell lines and patient tumor DNA.

However, in Figure S11A, the authors perform DAC and AZA pre-treatment but do not observe effective drug resistance through NRF2 induction. This raises concerns about the robustness of the proposed mechanism.

Contrary to the reviewer's interpretation, we are not proposing that hyper-methylation of ACB promoters is the main mechanism which prevents SCLC cells to respond to NRF2 induction. In the manuscript (line 333) we state that *“Contrary to our expectation, DNA-demethylation did not lead to more efficient drug resistance by NRF2 induction (now Fig. S12A), in line with inconsistent induction of hypermethylated ACB-promoters (now Fig. S12B). This indicates that, despite the high correlation of promoter methylation with ACB expression and drug sensitivity, epigenetic silencing is not a primary cause for the lack of ACB induction by NRF2 and the inability of SCLC to adapt to drug-induced ROS stress.”* We would like to emphasize that this statement is quite possibly underselling the importance of DNA methylation in SCLCs inability to activate ROS defence. **For once**, our attempt to demonstrate causality was compromised by the fact that SCLC cells are particularly sensitive to DAC or AZA treatment, which makes it likely that we had to under-dose the drugs to avoid cytotoxicity, which would mask resistance induction by the combined AZA/CDDO or DAC/CDDO treatment. **A second reason**, why DAC or AZA pre-treatment did not increase CDDO responsiveness might be due to the fact that not all ACB's promoters are subject to increased DNA methylation but rather are repressed by yet unidentified mechanisms, resulting in an insufficient boosting of ROS buffer capacity by demethylating drugs.

Reviewer #2 (Remarks to the Author): epigenetics and targeted therapy of lung cancer, including SCLC

The authors have significantly improved the manuscript.

Reviewer #3 (Remarks to the Author): targeted therapies including TXNRD1 inhibitors in cancer

My previous concerns have been well addressed. I only have two minor suggestions before the manuscript can be accepted for publication:

1. The data presented in Figure S3B support the conclusion that the relapsed tumors are resistant to cisplatin. However, the manuscript currently lacks a clear textual connection between these data and Figure 2. The authors are advised to revise the text to better integrate these findings.

We fully agree with the authors comment and adapted the relevant section in line 238 ff: *“Cells extracted from recurrent tumors had developed substantial resistance against cisplatin (> 40-fold) while their response to DKFZ-608 remained largely unaffected, confirming our in vitro data which suggest that DKFZ-608 is not subject to mechanisms involved in cisplatin resistance (Fig. S3B).”*

2. Line 205: The manuscript jumps from Figure S4 directly to Figure S13. Supplemental figures should be referenced in sequential order.

Thank you for pointing this out. We have carefully revised the numbering of the supplemental figures to ensure they are referenced in sequential order. The figure previously labeled as Fig. S13 has now been renumbered as Fig. S5, and the subsequent figures have been adjusted accordingly.

Reviewer #4 (Remarks to the Author): medicinal chemistry

I share the view of the three initial reviewers: this is an interesting and valuable manuscript. It shows that thioredoxin reductase inhibitors can act against small-cell lung cancer and their effect can be augmented by drugs that boost NRF2 signaling. The idea is that SCLC has a low antioxidant response and inhibitors of thioredoxin reductase exploit the dependency on this antioxidant system by cancer cells. The animal data demonstrate the viability and efficacy of the combined NRF2/TR approach that allows the usage of higher dosages of anti-thioredoxin reductase drugs delaying cancer relapse and minimizing resistance. Let's hope that this approach can move further into the clinic. The manuscript also discusses some experiments conducted to explore hypotheses about the molecular and cellular mechanisms underlying the sensitivity of SCLC to this inhibition. Though not conclusive, the data provide some hints that remain open for further investigations.

The previous reviewers gave a number of comments. The revised manuscript carefully addresses all of them. I offer a few suggestions that would require only minimal changes in the manuscript with the only aim of improving clarity.

1) I would move the supplement section 1 to the main text as it discusses interesting mechanistic data.

We fully agree with your suggestion and moved the text to what is now line 195 in the first results chapter.

2) Honestly, I have been struggling with **Figure 4A**. Can the authors revise the legend and improve clarity with a more extensive explanation of the tree at the top of the figure, the meaning of the two top lines and how the general reader can grasp the correlation between methylation levels and drug sensitivity.

Thank you for this valuable comment as it helps us to improve the readability of our manuscript. We have modified both **Fig. 4A** and its legend. In particular, we have color coded the clustering tree and moved row annotations to the left to facilitate alignment with the legend text.

Figure 4

Figure 4: Increased promoter methylation of ACB genes in SCLCs correlates with the sensitivity to TXNRD1 inhibitor. (A) Methylation values of CpG clusters previously identified within promoters of MAFG and ACB gene set for NSCLC and SCLC cell lines profiled in DepMap project (CCLE) that have expression and promoter methylation data. CpG clusters were arranged by hierarchical clustering

algorithm using euclidean distance metric and complete linkage method (default clustering settings of ComplexHeatmap package) into 4 groups of promoter methylation profiles distinguished by dendrogram colors: red – hypermethylated across all cell lines; black – hypermethylated in low-ACB cells; green – hypermethylated in high-ACB cells; blue – hypomethylated across all cell lines). Left annotations: EC_{50} – DKFZ-682 EC_{50} per cell line (rows are sorted based on these values); avg_ACB – average expression of genes from ACB gene set per cell line. Top annotations: 1st row – correlation between vector of cell lines' methylation values and vector of cell lines' DKFZ-682 EC_{50} values per each CpG cluster; 2nd row – correlation between vector of cell lines' methylation values and vector of cell lines' average expression of genes from ACB gene set per each CpG cluster. Negative correlation of methylation with drug sensitivity (EC_{50}) or ACB expression levels (purple) indicates that cells with hypermethylated CpGs in the indicated area demonstrate high drug sensitivity and low ACB expression values.

3) The data and associated text shown in Figures R3 and R4 in the rebuttal might be incorporated into the SI. They will be useful also to guide future studies as they rule out certain mechanistic hypotheses.

Thank you for this suggestion. We fully agree that negative results should be made public. It is our understanding that our communications during the review process, including rebuttal figures will be made available after the acceptance of the manuscript. We are more than willing to include all our negative results directly in the primary manuscript, however we would like to suggest to postpone the decision on which of those data should be included, until the final decision by the journal.

References

1. Lv Y, *et al.* BRD4 Targets the KEAP1-Nrf2-G6PD Axis and Suppresses Redox Metabolism in Small Cell Lung Cancer. *Antioxidants (Basel)* **11**, (2022).
2. Hussong M, *et al.* The bromodomain protein BRD4 regulates the KEAP1/NRF2-dependent oxidative stress response. *Cell Death Dis* **5**, e1195 (2014).
3. Michaeloudes C, *et al.* Bromodomain and extraterminal proteins suppress NF-E2-related factor 2-mediated antioxidant gene expression. *J Immunol* **192**, 4913-4920 (2014).
4. Laddha AP, Seyednejad A, Donepudi AC, Goedken MJ, Manautou JE, Sartor GC. Inhibition of bromodomain and extra-terminal (BET) proteins with JQ1 exacerbates acetaminophen-induced hepatotoxicity by altering detoxification pathways and oxidative stress responses. *Chem Biol Interact* **413**, 111491 (2025).
5. Wu Y, Mi Y, Zhang F, Cheng Y, Wu X. Suppression of bromodomain-containing protein 4 protects trophoblast cells from oxidative stress injury by enhancing Nrf2 activation. *Hum Exp Toxicol* **40**, 742-753 (2021).

6. Samarin J, *et al.* Low level of antioxidant capacity biomarkers but not target overexpression predicts vulnerability to ROS-inducing drugs. *Redox Biol* **62**, 102639 (2023).
7. Freire Boulosa L, *et al.* Optimization of the Solvent and In Vivo Administration Route of Auranofin in a Syngeneic Non-Small Cell Lung Cancer and Glioblastoma Mouse Model. *Pharmaceutics* **14**, (2022).
8. Johnson SS, *et al.* Auranofin inhibition of thioredoxin reductase sensitizes lung neuroendocrine tumor cells (NETs) and small cell lung cancer (SCLC) cells to sorafenib as well as inhibiting SCLC xenograft growth. *Cancer Biol Ther* **25**, 2382524 (2024).
9. Falchetti M, *et al.* Omics-based identification of an NRF2-related auranofin resistance signature in cancer: Insights into drug repurposing. *Comput Biol Med* **152**, 106347 (2023).
10. Jiang C, *et al.* A CRISPR screen identifies redox vulnerabilities for KEAP1/NRF2 mutant non-small cell lung cancer. *Redox Biol* **54**, 102358 (2022).
11. Luo G, *et al.* A Core NRF2 Gene Set Defined Through Comprehensive Transcriptomic Analysis Predicts Selective Drug Resistance and Poor Multicancer Prognosis. *Antioxid Redox Signal* **41**, 1031-1050 (2024).

Dear editor

We herewith address all points raised by reviewer#1.

REVIEWER COMMENTS

Reviewer #1 (Remarks to the Author): NRF2, KEAP1, redox biology and DNA methylation

1. Figure 1B:

The data presented in Figure 1B suggest that the observed difference in ACB protein expression is not attributable to SCLC per se. Rather, it appears that a subpopulation of NSCLC cell lines harboring KEAP1 mutations displays higher ACB expression. SCLC and KEAP1/NRF2 WT NSCLC exhibit comparable levels of ACB protein, indicating that there is nothing inherently distinct about SCLC in this regard. This conclusion is well supported by the rest of the data of the paper. Therefore, it would be more accurate to state that the difference arises from the elevated expression in the KEAP1-mutant NSCLC subset, rather than a generalized downregulation in SCLC.

Additionally, the lack of a significant difference between non-cancer cell lines, SCLC, and KEAP1/NRF2 WT NSCLC further supports this interpretation. The claim regarding lower ACB in SCLC should therefore be moderated to avoid overstatement.

We agree with the reviewer's view that low ACB expression is not a distinctive feature of SCLC versus NSCLC cell lines, as there is an **overlap** of ACB expression levels between wild-type KEAP1 NSCLC cell lines and wild-type KEAP1 SCLC lines. In order to illustrate this more clearly, we show below a comparison of cell lines, which are exclusively wild-type KEAP1 (Figure R3.1). We also show data derived from a collection of PDX models established by Champions Oncology. Here, the difference in ACB expression between the two entities appears to be bigger, however there is still an overlap.

Figure R3.1: The expression of 15 ACB genes for each SCLC and NSCLC cell line was calculated as average per sample and are based on the RNAseq datasets of CCLE cell lines (**only KEAP1 wild-type**) and PDX models (**unknown KEAP1-status**), obtained from Champions Oncology. For expression data normalization, values were transformed to $\log_2(\text{TPM}+1)$ units. After that average expression value for the ACB gene set were calculated for each sample (using $\log_2(\text{TPM}+1)$ expression values of individual genes).

Because some individual SCLC lines show ACB levels overlapping with certain NSCLC lines, we did not intend to attribute ACB expression specifically to SCLC, even though the average ACB expression in SCLC is significantly lower than in NSCLC. In order to better describe our assessment of the ACB status in cell lines we modified the statement in line 126ff: “Non-small cell lung cancer (NSCLC) cell

lines show varying levels of ACB expression. Lines carrying loss-of-function mutations in KEAP1 or CUL3 exhibit increased ACB expression, whereas those with wild-type KEAP1 or non-loss-of-function KEAP1 mutations generally display low basal ACB levels. Likewise, in small cell lung cancer (SCLC), ACB expression remains low across most cell lines, consistent with their wild-type KEAP1 status...".

It is worth noting that differences in ACB expression appear to be more pronounced in cells isolated from resected tumors of SCLC versus LUAD or LUSC patients, as opposed to cell lines (Figure 1C).

Figure 1C: ...
(C) Relative expression of ACB genes in normal tissues, LUSC, LUAD and PNEC were compared to SCLC using the Wilcoxon test (ns, not significant, * $p < 0.05$, ** $p < 0.01$, * $p < 0.001$, **** $p < 0.0001$).** Dot size indicates sample size, ranging from $n=1$ (skin) to $n=542$ (LUAD). Expression values are normalized to normal lung tissue.

2. TXNRD1 inhibitor experiments:

While the authors have clearly invested substantial effort into the TXNRD1 inhibitor studies, it is important to note that **similar findings** have previously been published by the same group. Given this context, descriptive observations alone may not provide sufficient novelty or mechanistic insight. Strengthening this section with more direct **mechanistic data** or **functional validation** would enhance its impact.

We thank the reviewer for this comment and would like to respond on the issues of novelty, mechanistic insight and functional validation.

Novelty: We have transparently cited our prior work (on the identification of the ACB biomarker set in NSCLC) as it provides the rationale for drug sensitivity in certain cancer types (low ACB status) and offers a means to identify susceptible patient populations. In the present study we have used the ACB concept only as a rationale to focus on SCLC. Distinct from our previous work we introduce several **novel findings**, including:

- SCLC are homogeneously sensitive to TXNRD1 inhibition, while NSCLC responses are heterogeneous (e.g., Fig 1A, right panel).
- Sensitivity is independent of neuroendocrine status (Fig. S1B, S2B) and cisplatin resistance (Fig. 1D, 1E, S2D, S3D).
- Maintenance therapy with TXNRD1 inhibitors is effective and compatible with standard SCLC treatment. (Fig. 2)
- The therapeutic window can be widened by selectively reducing organ toxicity without compromising anti-tumor efficacy. (Fig. 6)

As these findings, rather than the previously described ACB concept, constitute the core of our study, we are confident that the manuscript presents several novel and impactful discoveries. Importantly, our results highlight TXNRD1 as an attractive drug target by demonstrating that inhibitors can be successfully tested when the correct cancer entity is identified and systemic toxicity is effectively managed.

Mechanistic insight: Although a “master mechanism” explaining non-responsiveness of SCLCs to NRF2 induction remains to be established, we think that the novelty and translational potential of our findings outweigh this limitation.

Why did we fail to identify such a “master mechanism”? After many different attempts we start to believe that (1) such a general mechanism might not exist across the SCLC lines investigated (comparable to the heterogeneity of cell death mechanisms identified) and (2) SCLCs may employ multiple, potentially overlapping strategies to prevent ACB induction by NRF2.

We find the latter particularly plausible, as we were unable to isolate any clones responsive to NRF2 induction, despite months of selection—an unbiased, standard approach in our laboratory for identifying drug response mechanisms. In addition, we explored several hypothesis-driven strategies, spanning from epigenetic repression, the lack of a cofactor required by NRF2 for efficient ACB induction or the expression of a transcription repressor that prevent ACB expression. The results are presented in the revised manuscript and rebuttal materials. We appreciate the recognition of our efforts during the review process and agree with the suggestion that including negative results is important, as they offer valuable insights for future investigations.

Functional validation: A key finding of this study is that SCLC cell lines fail to translate NRF2 induction into resistance against drug-induced ROS stress. This observation motivated the mechanistic analyses described above, as well as comprehensive functional validation at multiple levels. Importantly, we show that systemic NRF2 induction increases the cellular ROS-scavenging capacity and thereby protects normal tissues but not SCLC tumors, allowing the expansion of the therapeutic window. We believe that this functional validation compensates for the absence of a strictly SCLC-specific mechanism, as it is precisely these results that will drive future translational studies in other relevant cancer types.

3. Use of H1944 cell line:

Since H1944 carries a KEAP1 mutation, it may not be an appropriate reference for comparison in this context, especially for the KEAP1 Western blot in Figure R1. Including a KEAP1 wild-type reference cell line would help clarify the interpretation of these results.

We thank the reviewer for this comment as it shows us that we still have not clarified the role of H1944 in this figure. This cell line serves as a “technical control” which helps to compare bands from different westerns. Although β -Tubulin and GAPDH serve a similar purpose, we felt that including the same cell line on both western blots allows to compare those two westerns even better. We could have used any other cell line expressing KEAP1 protein, but decided to use H1944 for consistency reasons (see also Fig. 1B, S1A, S6C, S11B). Since H1944 serves as a technical control but not a functional reference, we consider the mutation status of KEAP1 in this cell line as irrelevant for the interpretation of the western.

You suggested to use a KEAP1-wild-type cell line as reference to the KEAP1-wild type SCLC lines. We had used Beas-2B and HaCaT cells as references for normal cells, which respond to NRF2 induction. Both cells are KEAP1-wild-type, like the SCLC lines analyzed here.

Figure R1: Protein levels of KEAP1 in total cell extracts from 13 SCLC and non-cancerous cell lines (Beas-2B, HaCaT) were analysed by immunoblotting (representative image, qualitatively similar to 2 biological replicates). The NSCLC cell line H1944 was used as a loading control to compare the Westerns.

4. Figure R2.1:

- (A–B) The rationale for restricting CDDO-Me treatment to 6 hours is not clearly explained. Extending or justifying this time course would improve the rigor of the experiment.

We agree with the reviewer that the experimental design had not been properly explained. First of all, we would like to clarify the purpose of the analysis. We wanted to know, whether SCLCs, like non-cancerous cells (e.g., Beas-2B) have a functional NRF2-regulon, in that KEAP1 inhibition not only leads to an increase of NRF2 protein, but also to the induction of NRF2 response genes. We had chosen the 6-hour CDDO-Me treatment time point because it produced the most consistent biomarker induction across cell lines; longer exposures introduced variability in the expression of some biomarkers. Figure R3.2 shows that while NQO1 is robustly induced by CDDO-Me both at 6 h and 24 h, HMOX1 shows the most robust response at 6 h, while induction is more moderate at 24 h. Accordingly, in order to maximize the resolution of the assessment whether the NRF2-regulon works in SCLC, we chose the 6 h time point.

Figure R3.2: Dose- and time-dependent effects of CDDO-Me on NRF2 target gene expression in Beas-2B, H82 and H526 cell lines. The indicated cell lines were treated with DMSO (0 nM CDDO-Me) or CDDO-Me (50 nM or 100 nM) for 6 h or 24 h. The level of NRF2 target genes were analyzed by qPCR (Control without CDDO-Me was set to 1; relative data represent mean \pm SD of one experiment performed in triplicates; *** p < 0.001, **** p < 0.0001, one-way ANOVA, compared to the control).

- (B) The choice to remove CDDO-Me prior to DKFZ-682 treatment rather than performing co-treatment warrants clarification. A co-treatment strategy could better reflect potential interactions between these compounds.

Your interpretation of the experiment as sequential rather than co-treatment reflects unclear legends of Figure R2.1 and several figures in the manuscript. We confirm that co-treatments were performed in all experiments, as you had suggested; we will revise Figure 3G, S10B, S11D legend in stating "... The cells were treated with CDDO-Me (50 nM) or DMSO (control). After 24 h, a concentration series of DKFZ-682 was added for another 24 h....".

- Including parallel experiments with non-cancerous cell lines would further strengthen the claim that “NRF2 induction does not enhance the ROS-scavenging capacity of SCLC.”

Thank you for this helpful comment. We had interpreted your previous comment (Comment 2) as a suggestion to perform a dose–response analysis in SCLC cells to determine whether the lack of response to CDDO-Me could be attributed to underdosing. As shown in Figure R2.1, this is not the case. We had already conducted several parallel experiments comparing non-cancerous and SCLC cell lines using the maximal non-toxic concentration of CDDO-Me (50 nM) (see Fig. 3, Fig. S8, Fig. S9D–G, Fig. S10). For this reason, a non-cancerous cell line was not initially included in Figure R2.1. Following your suggestion, we have now added data from Beas-2B cells to the revised Figure R2.1 (see below). As expected, we find that CDDO-Me induces dose-dependent resistance in the non-cancerous cell line but not in SCLCs. Using higher doses to induce resistance in SCLCs is not feasible, as such concentrations would be toxic and thereby counteract a potential resistance-induction to DKFZ-682 (Fig. S8D).

Revised Figure R2.1: Dose-dependent effects of CDDO-Me on NRF2 target gene expression and drug sensitivity in SCLC and Beas-2B. (A, B) The indicated cell lines were treated with DMSO (control), CDDO-Me

(25 nM, 50 nM or 100 nM) for 6 h. The level of NRF2 target genes were analyzed by qPCR (Control without CDDO-Me was set to 1; relative data represent mean \pm SD of two independent experiments each performed in triplicates; * $p < 0.05$, ** $p < 0.01$, *** $p < 0.001$, **** $p < 0.0001$, one-way ANOVA, compared to the control). (C) The cells were treated with CDDO-Me (25 nM, 50 nM or 100 nM) or DMSO (control). After 24 h, a concentration series of DKFZ-682 was added for another 24 h. The cell viability was quantified by the CellTiter-Glo assay. The results are representative of two independent experiments, each performed in triplicates.